# Hierarchical sensitivity analysis for a large-scale process-based hydrological model applied to an Amazonian watershed

Haifan Liu[1], Heng Dai[2,3,4*], Jie Niu[4*], Bill X. Hu[4], Dongwei Gui[2,3], Han Qiu[5], Ming Ye[6], Xingyuan Chen[7], Chuanhao Wu[4], Jin Zhang[4], and William Riley[8]

[1]School of Water Resources and Environment, China University of Geosciences, Beijing, 100083, China.
[2]State Key Laboratory of Desert and Oasis Ecology, Xinjiang Institute of Ecology and Geography, Chinese Academy of Sciences, Urumqi 830011, China.
[3]Cele National Station of Observation and Research for Desert-Grassland Ecosystem,Xinjiang Institute of Ecology and Geography, Chinese Academy of Sciences, Cele 848300, China
[4]Institute of Groundwater and Earth Sciences, Jinan University, Guangzhou 510632, China.
[5]Department of Civil and Environmental Engineering, Michigan State University, East Lansing, MI. USA.
[6]Department of Earth, Ocean, and Atmospheric Science, Florida State University, Tallahassee, FL 32306, USA.
[7]Pacific Northwest National Laboratory, Richland, WA 99352, USA.
[8]Earth Sciences Division, Lawrence Berkeley National Laboratory, Berkeley, California, USA.

*Correspondence to*: Heng Dai (heng.dai@jnu.edu.cn) and Jie Niu (jniu@jnu.edu.cn)

**Abstract.** Sensitivity analysis methods have recently received much attention for identifying important uncertainty sources (or uncertain inputs) and improving model calibrations and predictions for hydrological models. However, it is still challenging to apply the quantitative and comprehensive global sensitivity analysis method to complex large-scale process-based hydrological models (PBHMs) because of its variant uncertainty sources and high computational cost. Therefore, a global sensitivity analysis method that is capable of simultaneously analyzing multiple uncertainty sources of PBHMs and providing quantitative sensitivity analysis results is still lacking. In an effort to develop a new tool for overcoming these weaknesses, we improved the hierarchical sensitivity analysis method by defining a new set of sensitivity indices for subdivided parameters. A new binning method and Latin hypercube sampling (LHS) were implemented for estimating these new sensitivity indices. For test and demonstration purposes, this improved global sensitivity analysis method was implemented to quantify three different uncertainty sources (parameters, models, and climate scenarios) of a three-dimensional, large-scale and process-based hydrologic model (PAWS) with an application case in an ~9,000 km$^2$ Amazon catchment. The importance of different uncertainty sources was quantified by sensitivity indices for two hydrologic outputs of interest: evapotranspiration (*ET*) and groundwater contribution to streamflow ($Q_G$). The results show that the parameters, especially the vadose zone parameters, are the most important uncertainty contributors for both outputs. In addition, the influence of climate scenarios on *ET* predictions is also important. Furthermore, the thickness of the aquifers is important for $Q_G$ predictions, especially in main stream areas. These sensitivity analysis results provide useful information for modelers, and our method is mathematically rigorous and can be applied to other large-scale hydrological models.

# 1 Introduction

The rapidly increasing computing power in recent years has accelerated the innovation of hydrological models, and more complex hydrological processes have been included in new models, which are capable of simulating large-scale problems (Freeze and Harlan, 1969; Singh and Woolhiser, 2002). PBHMs are complex hydrological models that link the characteristics of a river basin with hydrological processes (Refsgaard and Knudsen, 1996; Maxwell et al., 2014). The functions of PBHMs include both evaluation of the watershed response to future climate scenarios and simulation of the basin-to-continental scale ecosystem energy balance, biogeochemistry, and ecological functioning (Vertessy et al., 1993; Parkin et al., 1996; Bixio et al., 2002; Oogathoo et al., 2011; Weill et al., 2011; Shen et al., 2013; Maxwell et al., 2014; Riley and Shen, 2014). Specific to the hydrological process, PBHMs are capable of simulating the surface water processes of $ET$, overland flow, channel runoff, and so on (Beven, 2002). For subsurface water, PBHMs can simulate complex hydrological processes in the soil, such as root extraction, infiltration, soil evaporation, and groundwater discharge and recharge in the vadose zone, by solving the Richards equation (Maxwell et al., 2014). However, these complex processes and governing equations embedded in the PBHMs inevitably induce large uncertainties in the modeling predictions (Neuman, 2003; Rojas et al., 2010; Lu et al., 2012; Shen et al., 2014; Razavi and Gupta, 2015, 2016; Qiu et al., 2019). How to efficiently decrease these large uncertainties becomes an essential problem for modelers. Sensitivity analysis aims to identify the most influential sources of uncertainty and is therefore an important tool (Saltelli and Sobol, 1995; Saltelli et al., 2000, 2010; Song et al., 2015). The sensitivity analysis results assist modelers and managers in focusing on observing and calibrating the uncertain inputs that have the greatest influences on model outputs. Thus, the sensitivity analysis process saves resources (e.g., funding and manpower) used for calibration and significantly improves the efficiency of reducing the uncertainty of PBHM predictions.

In general, sensitivity analysis methods can be divided into local and global categories. The main limitation of the local sensitivity analysis is that its results are only valid for a small range of parameter values (Gedeon and Mallants, 2012; King and Perera, 2013; Wainwright et al., 2014; Dai and Ye, 2015). Compared with the local method, global sensitivity analysis can provide sensitivity estimates for the entire range of uncertain parameter values (Saltelli et al., 2000, 2010; Razavi and Gupta, 2015, 2016). Because of this advantage, global sensitivity analysis has gained popularity in recent modeling works despite its high computational cost (Hamby, 1994; van Griensven et al., 2006; Sulis et al., 2011; Baroni and Tarantola, 2014). Common global sensitivity analysis methods include screening methods, regression-based methods, variance-based methods, meta-model methods (Song et al., 2013), and information-entropy-based methods (Zeng et al., 2012). Among the different global sensitivity analysis methods, the variance-based method has been widely accepted and used because of its ability to accurately quantify the importance of uncertain parameters while considering their interactions (Saltelli and Sobol, 1995; Zhang et al., 2013; Dai and Ye, 2015).

To date, considerable research has been conducted to reduce the uncertainties in hydrological models by using local or global sensitivity analysis methods (e.g., Nijssen et al., 2001; Chávarri et al., 2013; de Paiva et al., 2013). However,

conducting a comprehensive global sensitivity analysis, especially variance-based sensitivity analysis on PBHMs, remains a challenge, and there are two main obstacles. The first obstacle is the high computational cost rising from two sources: the high complexity of the model itself and the method requirement of variance-based global sensitivity analysis. A PBHM usually has a very large number of parameters and multiple high-order nonlinear governing equations. These facts combined

with a large-scale model domain cause the running of a PBHM itself to be very computationally expensive. For the sensitivity analysis method, compared with the local sensitivity analysis, which can only provide results valid in a certain range of parameter values (e.g., the derivative of the model prediction with respect to parameter A at a certain value point can be a measurement of A's local sensitivity at this point), the global sensitivity analysis is more comprehensive because its results are valid for the whole range of parameter values. To achieve this goal, the methods of global sensitivity analysis are

all relatively computationally expensive, especially for the variance-based method, which uses complex sampling techniques, and its computational cost grows exponentially with the number of parameters (Saltelli et al., 2000, 2010). Therefore, the implementation of a global sensitivity analysis for a PBHM leads to an extremely high computational cost considering that we have to run a large number of simulations for a complex PBHM using different parameter samples.

The second obstacle of implementing the global sensitivity analysis method in PBHMs is the variant uncertainty sources included in the model. Conventional global sensitivity analysis generally considers only uncertainty from model parameters

and ignores other important hydrological model uncertainties. However, for PBHMs, uncertainties usually arise from three different sources, including parametric uncertainty, model structural uncertainty (induced through multiple different plausible conceptual or mathematical models), and scenario uncertainty (caused by alternative unpredictable future climate conditions) (Ye et al., 2005; Makler-Pick et al., 2011; Neumann, 2012; Dai and Ye, 2015; Song et al., 2015; Dai et al., 2017a,

2017b; Zeng et al., 2018; Pan et al., 2020). To overcome these two obstacles, Dai et al. (2017a) developed a new hierarchical sensitivity analysis method that integrates the variance-based method and hierarchical uncertainty framework. By combining uncertain inputs based on their characteristics and dependencies, hierarchical sensitivity analysis can quantify the sensitivity of different sources of uncertainty involved in hydrological models (e.g., parameters, models, and climate scenarios) and dramatically reduce the computational cost. However, the original hierarchical sensitivity analysis method is limited to

considering parameters as a whole, and the sensitivity indices of different parameters cannot be defined or estimated. This simple strategy may be adequate for a groundwater modeling case, but it cannot provide detailed information for a PBHM that includes multiple hydrological processes.

This research presents a new tool of the improved hierarchical sensitivity analysis method and demonstrates its implementation to a pilot example for comprehensive global sensitivity analysis of large-scale PBHMs. A new set of

subdivided parametric sensitivity indices was defined to quantify the importance of a physical process involving only partial model parameters. A new binning method was implemented with the Latin hypercube sampling (LHS) method to estimate these subdivided parameter sensitivity indices. The LHS method also makes the assessment of hierarchical sensitivity analysis for large-scale PBHMs more computationally affordable compared with the original Monte Carlo method. This new

and flexible hierarchical sensitivity analysis method provides modelers with the novel capability of analyzing sensitivity from the physical process viewpoint and estimating accurate importance for further subdivided parameter groups.

The process-based adaptive watershed simulator (PAWS) model was first developed in Shen and Phanikumar (2010); the PAWS is capable of simulating large catchments and long-term frames by efficiently coupling surface and subsurface hydrological processes. Coupling the PAWS with the CLM (Community Land Model) can enable the model to describe vegetation respiration and evapotranspiration in a physics-based manner (Shen et al., 2014; Niu et al., 2017). The model has been applied extensively in many watersheds, e.g., the large-scale watersheds in Michigan, U.S. (Shen et al., 2013, 2014, 2016; Niu et al., 2014, 2017; Ji et al., 2015; Qiu et al., 2019) and the watershed in the Amazon basin (Niu et al., 2017), and the model has presented good performances in these watersheds. The PAWS can also estimate multiple key variables of hydrological states and fluxes at different spatiotemporal scales. The high efficiency, great performance, and complex variables all make PAWS an excellent model choice for PBHMs to evaluate and demonstrate the sensitivity analysis method.

The PAWS model with the new hierarchical sensitivity analysis method was implemented in a study area of the ~9,000 km$^2$ Amazon catchment located in northern Manaus, Brazil, for the purposes of evaluation and demonstration. Three different types of uncertainty sources (climate scenario, model, and parameters) were all included in this test case. The parameters were further divided into three groups (vadose zone parameters, groundwater parameters, and overland flow parameter) to investigate the detailed importance information of the model parameters through the new subdivided parameter sensitivity indices. By developing the new hierarchical sensitivity analysis method and implementing it in this test case, we aim to (1) provide a new tool and pilot example of comprehensive global sensitivity analysis for the PBHMs; (2) identify the most important uncertainty sources for modeling hydrological processes in the Amazon; and (3) investigate the possible patterns for sensitivity analysis results of PBHMs.

We introduce the study area and the numerical model in Section 2.1. Sections 2.2 and 2.3 present the improved hierarchical sensitivity analysis method and its algorithms in detail. Then, we describe the generation of uncertainty sources based on the study site information in Section 2.4. We present and discuss the results in Section 3. Finally, Section 4 summarizes the key findings of this research.

## 2 Materials and methods

### 2.1 Study site and numerical model

The study site is located in northern Manaus, Brazil (Fig. 1), and the site has a drainage area of ~9,000 km$^2$. Within the central Amazon, the watershed is mostly covered by tropical forest, with ~12% cropland and ~5% wetland (based on CLM land surface data; Niu et al. (2017)). With the relatively high elevation (90 – 210 m) of the upper landscape and relatively low elevation (45 – 55 m) of the swampy valleys, a dense drainage network formed in the region. The watershed has 4 rivers: the Urubu, Preto da Eva, Tarumã-açu, and Tarumã-mirim Rivers. The average precipitation in this region has large seasonal variability. December to May is the wet season, and June to November is the dry season.

The modeling tool used in this study is the PAWS model (Shen and Phanikumar, 2010; Shen et al., 2014; Niu et al., 2017). The main reason for choosing the PAWS as the pilot example of PBHMs is that compared with other PBHMs, the PAWS is a comprehensive and representative large-scale hydrological model that can be applied to large catchments and long-term frames by efficiently coupling both surface and subsurface hydrological processes (Shen and Phanikumar, 2010). The complexity and parameter dimensionality of the PAWS are high enough to test and demonstrate our new global sensitivity analysis method. Furthermore, the PAWS was previously applied to the studied watershed, and it was capable of simulating multiple key variables of hydrological states and fluxes at different spatiotemporal scales and presented good model performance validated by various ground and satellite observation data (Niu et al., 2017). This previous model application provides a solid basis for our uncertainty identification and sensitivity analysis study.

The details of the numerical implementation and the governing equations of the PAWS can be found in Appendix A. Briefly, four flow domains are simulated in the PAWS, including the stream channel, overland flow, vadose zone, and saturated groundwater. The structured grid-based finite-volume method is the main numerical scheme applied to discretize the governing equations of the various hydrologic components. The PAWS also simulates two land surface subdomains, i.e., infiltration and evaporation, which are depicted in the ponding subdomain, while overland flow occurs in the surface flow subdomain. The PAWS considers the horizontal interaction of both surface runoff and groundwater flow between model grids, which represents the actual hydrological processes and is often ignored by other regional and global hydrologic models. The 1-D diffusive wave equation is solved to simulate channel flow, and the 2-D version is used for overland flow. The leakance concept is the concept applied to explicitly simulate the exchange between the channel and groundwater. The PAWS has been coupled with the CLM (Shen et al., 2014), which calculates the surface energy balance and soil and plant carbon and nitrogen cycles. Canopy interception and *ET* demand (both transpiration and soil evaporation) are also computed in the CLM at each time step.

For the numerical model case in this study, a 1 km × 1 km grid is used for horizontal discretization, resulting in 118 × 122 grid cells for the study site. In this model, 20 vertical layers are defined to discretize the vadose zone, and for the fully saturated groundwater, there are two layers: the unconfined aquifer at the top and the confined aquifer at the bottom. In this study, the 90 m resolution NASA Shuttle Radar Topography Mission (SRTM) (U.S. Geological Survey; http://eros.usgs.gov) data are applied as DEM input, but for the channel network and watershed boundary delineation, the Advanced Spaceborne Thermal Emission and Reflection Radiometer (ASTER) provides the 30 m resolution Global Digital Elevation Model Version 2 (GDEM V2). CLM land surface data are applied as land use and land cover (LULC) inputs. Details regarding these data can be found in Niu et al. (2017). More information on the governing equations of the PAWS can be found in Shen and Phanikumar (2010) and Niu et al. (2014).

## 2.2 Hierarchical sensitivity analysis method with subdivided parametric sensitivity

The essential concept of the hierarchical sensitivity analysis method involves categorizing and quantifying different complex uncertainties of certain model systems while considering their dependence relationships. Different uncertainty sources (or

uncertain inputs) are placed in different layers of a hierarchical uncertainty framework, which is then integrated with the variance-based global sensitivity analysis method to form a new set of sensitivity indices to accurately quantify the importance of different uncertainty sources.

In this study, climate scenarios, different aquifer thicknesses, and parameters are treated as random uncertain inputs, and they represent the climate scenario uncertainty, model uncertainty, and parameter uncertainty, respectively. Notably, the thicknesses of aquifers here represent the model uncertainty because the different thicknesses of distinct types of aquifers lead to different conceptual hydrological models, and a similar concept (different thicknesses were used for two underground geological formations) for the model uncertainty was used in previous work (Dai et al., 2017a). Six model parameters are included in this test case, and they are divided into three groups. The first group includes vadose zone parameters ($\mathbf{PR}_{VDZ}$): soil saturated hydraulic conductivity, $K_s$ (m day$^{-1}$), the van Genuchten equation parameters $\alpha$ (m$^{-1}$) and $N$ (unitless) (van Genuchten, 1980). The second group is composed of groundwater parameters ($\mathbf{PR}_{GW}$): unconfined aquifer hydraulic conductivity, $K_1$ (m day$^{-1}$), and confined aquifer hydraulic conductivity, $K_2$ (m day$^{-1}$). The third group is the overland flow parameter ($\mathbf{PR}_{OVN}$): the length of the flow path for runoff contribution to the overland flow domain, $L$ (m). Here, we consider the van Genuchten parameters $\alpha$ and $N$ because the correlation between $\alpha$ and $N$ can largely affect the soil water release and infiltration processes in the vadose zone (Pan et al., 2011). In the hierarchical uncertainty framework, all these uncertainties are placed into the proper levels based on their dependence relationships. The climate scenario uncertainty is at the top layer, and the model uncertainty and parameter uncertainty are at the middle and bottom layers, respectively (Fig. 2), which is because the CS are the driving forces of the hydrological model system, and multiple models can be built under a single scenario. Similar to the model and parameters, each model can contain a different set of parameters (Meyer et al., 2007). According to the hierarchical sensitivity analysis method, the partial variances contributed by the climate scenario uncertainty, model uncertainty, and parameter uncertainty can be expressed as Eqs. (1)-(3), respectively (see Appendix B for more details).

$$V\left(\mathbf{CS}\right) = V_{\mathbf{CS}} E_{\mathbf{NM|CS}} E_{\mathbf{PR|NM,CS}} \left(\Delta \mid \mathbf{NM, CS}\right) \tag{1}$$

$$V\left(\mathbf{ST}\right) = E_{\mathbf{CS}} V_{\mathbf{NM|CS}} E_{\mathbf{PR|NM,CS}} \left(\Delta \mid \mathbf{NM, CS}\right) \tag{2}$$

$$V\left(\mathbf{PR}\right) = E_{\mathbf{CS}} E_{\mathbf{NM|CS}} V_{\mathbf{PR|NM,CS}} \left(\Delta \mid \mathbf{NM, CS}\right) \tag{3}$$

where $\Delta$ is the model output, **CS** represents the set of alternative climate scenarios, **NM** represents the multiple plausible models with different aquifer thicknesses, and **PR** represents the multiple parameter sets under a certain model. The notations of **NM|CS** and **PR|NM, CS** indicate the hierarchical relationships that models are conditioned on climate scenarios and parameters are conditioned on models and climate scenarios. The term $\Delta$|**NM, CS** indicates that the output is calculated using the parameter sets that are conditioned on climate scenarios and models.

The sensitivity indices for the climate scenarios $S_{CS}$, models $S_{NM}$, and parameters $S_{PR}$ are expressed as Eqs. (4)-(6), following the hierarchical sensitivity analysis method:

$$S_{CS} = \frac{V_{CS}E_{NM|CS}E_{PR|NM,CS}(\Delta|NM,CS)}{V(\Delta)} = \frac{V(CS)}{V(\Delta)}, \tag{4}$$

$$S_{NM} = \frac{E_{CS}V_{NM|CS}E_{PR|NM,CS}(\Delta|NM,CS)}{V(\Delta)} = \frac{V(NM)}{V(\Delta)}, \tag{5}$$

$$S_{PR} = \frac{E_{CS}E_{NM|CS}V_{PR|NM,CS}(\Delta|NM,CS)}{V(\Delta)} = \frac{V(PR)}{V(\Delta)}. \tag{6}$$

where $V(\Delta)$ is the total variance in the model output (Eq. (B5)). The above equations are directly derived based on the hierarchical sensitivity analysis method. Notably, the parameter sensitivity index in Eq. (6) includes the influence of all parameters. However, to explore the detailed parameter sensitivity, the total parameter uncertainty is further decomposed into three components, representing the uncertainties contributed from vadose zone parameters ($PR_{VDZ}$), groundwater parameters ($PR_{GW}$) and the overland flow parameter ($PR_{OVN}$). Using the variance decomposition method (Eq. (B1)), the partial variance in the parameters can be further decomposed as follows:

$$
\begin{aligned}
V(PR) &= E_{CS}E_{NM|CS}V_{PR|NM,CS}(\Delta|NM,CS) \\
&= E_{CS}E_{NM|CS}\left( \begin{array}{l} V_{PR_{VDZ}|NM,CS}\left(E_{PR_{\sim VDZ}|PR_{VDZ},NM,CS}(\Delta|PR_{VDZ},NM,CS)\right)+ \\ E_{PR_{VDZ}|NM,CS}\left(V_{PR_{\sim VDZ}|PR_{VDZ},NM,CS}(\Delta|PR_{VDZ},NM,CS)\right) \end{array} \right). \\
&= E_{CS}E_{NM|CS}V_{PR_{VDZ}|NM,CS}E_{PR_{GW},PR_{OVN}|PR_{VDZ},NM,CS}(\Delta|PR_{VDZ},NM,CS) \\
&\quad + E_{CS}E_{NM|CS}E_{PR_{VDZ}|NM,CS}V_{PR_{GW},PR_{OVN}|PR_{VDZ},NM,CS}(\Delta|PR_{VDZ},NM,CS)
\end{aligned}
\tag{7}
$$

where the notation $PR_{\sim VDZ}$ refers to other uncertain parameters excluding vadose zone parameters, which are groundwater parameters and the overland flow parameter. The first term of Eq. (7) on the right-hand side is the partial variance contributed by $PR_{VDZ}$, and the second term represents the partial variance in the other parameters, which are groundwater parameters and the overland flow parameter. Note that Eq. (7) is decomposed based on the vadose zone parameters; when we decompose the partial variance in parameters based on the groundwater parameters or the overland flow parameter, the partial variance in the parameters can be further decomposed as Eqs. (8) and (9):

$$V(\mathbf{PR}) = E_{\mathbf{CS}} E_{\mathbf{NM}|\mathbf{CS}} V_{\mathbf{PR}|\mathbf{NM},\mathbf{CS}} (\Delta \mid \mathbf{NM}, \mathbf{CS})$$

$$= E_{\mathbf{CS}} E_{\mathbf{NM}|\mathbf{CS}} \begin{pmatrix} V_{\mathbf{PR}_{\mathbf{GW}}|\mathbf{NM},\mathbf{CS}} \left( E_{\sim \mathbf{PR}_{\mathbf{GW}}|\mathbf{PR}_{\mathbf{GW}},\mathbf{NM},\mathbf{CS}} (\Delta \mid \mathbf{PR}_{\mathbf{GW}}, \mathbf{NM}, \mathbf{CS}) \right) + \\ E_{\mathbf{PR}_{\mathbf{GW}}|\mathbf{NM},\mathbf{CS}} \left( V_{\sim \mathbf{PR}_{\mathbf{GW}}|\mathbf{PR}_{\mathbf{GW}},\mathbf{NM},\mathbf{CS}} (\Delta \mid \mathbf{PR}_{\mathbf{GW}}, \mathbf{NM}, \mathbf{CS}) \right) \end{pmatrix},$$

$$= E_{\mathbf{CS}} E_{\mathbf{NM}|\mathbf{CS}} V_{\mathbf{PR}_{\mathbf{GW}}|\mathbf{NM},\mathbf{CS}} E_{\mathbf{PR}_{\mathbf{VDZ}},\mathbf{PR}_{\mathbf{OVN}}|\mathbf{PR}_{\mathbf{GW}},\mathbf{NM},\mathbf{CS}} (\Delta \mid \mathbf{PR}_{\mathbf{GW}}, \mathbf{NM}, \mathbf{CS})$$

$$+ E_{\mathbf{CS}} E_{\mathbf{NM}|\mathbf{CS}} E_{\mathbf{PR}_{\mathbf{GW}}|\mathbf{NM},\mathbf{CS}} V_{\mathbf{PR}_{\mathbf{VDZ}},\mathbf{PR}_{\mathbf{OVN}}|\mathbf{PR}_{\mathbf{GW}},\mathbf{NM},\mathbf{CS}} (\Delta \mid \mathbf{PR}_{\mathbf{GW}}, \mathbf{NM}, \mathbf{CS}) \tag{8}$$

$$V(\mathbf{PR}) = E_{\mathbf{CS}} E_{\mathbf{NM}|\mathbf{CS}} V_{\mathbf{PR}|\mathbf{NM},\mathbf{CS}} (\Delta \mid \mathbf{NM}, \mathbf{CS})$$

$$= E_{\mathbf{CS}} E_{\mathbf{NM}|\mathbf{CS}} \begin{pmatrix} V_{\mathbf{PR}_{\mathbf{OVN}}|\mathbf{NM},\mathbf{CS}} \left( E_{\sim \mathbf{PR}_{\mathbf{OVN}}|\mathbf{PR}_{\mathbf{OVN}},\mathbf{NM},\mathbf{CS}} (\Delta \mid \mathbf{PR}_{\mathbf{OVN}}, \mathbf{NM}, \mathbf{CS}) \right) + \\ E_{\mathbf{PR}_{\mathbf{OVN}}|\mathbf{NM},\mathbf{CS}} \left( V_{\sim \mathbf{PR}_{\mathbf{OVN}}|\mathbf{PR}_{\mathbf{OVN}},\mathbf{NM},\mathbf{CS}} (\Delta \mid \mathbf{PR}_{\mathbf{OVN}}, \mathbf{NM}, \mathbf{CS}) \right) \end{pmatrix}.$$

$$= E_{\mathbf{CS}} E_{\mathbf{NM}|\mathbf{CS}} V_{\mathbf{PR}_{\mathbf{OVN}}|\mathbf{NM},\mathbf{CS}} E_{\mathbf{PR}_{\mathbf{VDZ}},\mathbf{PR}_{\mathbf{GW}}|\mathbf{PR}_{\mathbf{OVN}},\mathbf{NM},\mathbf{CS}} (\Delta \mid \mathbf{PR}_{\mathbf{OVN}}, \mathbf{NM}, \mathbf{CS})$$

$$+ E_{\mathbf{CS}} E_{\mathbf{NM}|\mathbf{CS}} E_{\mathbf{PR}_{\mathbf{OVN}}|\mathbf{NM},\mathbf{CS}} V_{\mathbf{PR}_{\mathbf{VDZ}},\mathbf{PR}_{\mathbf{GW}}|\mathbf{PR}_{\mathbf{OVN}},\mathbf{NM},\mathbf{CS}} (\Delta \mid \mathbf{PR}_{\mathbf{OVN}}, \mathbf{NM}, \mathbf{CS}) \tag{9}$$

The first terms in Eqs. (8) and (9) represent the partial variances contributed by the groundwater and overland flow parameters, respectively. Then, we can define a new set of subdivided parameter sensitivity indices for the $\mathbf{PR}_{\mathbf{VDZ}}$, $\mathbf{PR}_{\mathbf{GW}}$ and $\mathbf{PR}_{\mathbf{OVN}}$ following the first-order sensitivity index definition (Eq. (B2)):

$$S_{\mathbf{PR}_{\mathbf{VDZ}}} = \frac{E_{\mathbf{CS}} E_{\mathbf{NM}|\mathbf{CS}} V_{\mathbf{PR}_{\mathbf{VDZ}}|\mathbf{NM},\mathbf{CS}} E_{\mathbf{PR}_{\mathbf{GW}},\mathbf{PR}_{\mathbf{OVN}}|\mathbf{PR}_{\mathbf{VDZ}},\mathbf{NM},\mathbf{CS}} (\Delta \mid \mathbf{PR}_{\mathbf{VDZ}}, \mathbf{NM}, \mathbf{CS})}{V(\Delta)} = \frac{V(\mathbf{PR}_{\mathbf{VDZ}})}{V(\Delta)}, \tag{10}$$

$$S_{\mathbf{PR}_{\mathbf{GW}}} = \frac{E_{\mathbf{CS}} E_{\mathbf{NM}|\mathbf{CS}} V_{\mathbf{PR}_{\mathbf{GW}}|\mathbf{NM},\mathbf{CS}} E_{\mathbf{PR}_{\mathbf{VDZ}},\mathbf{PR}_{\mathbf{OVN}}|\mathbf{PR}_{\mathbf{GW}},\mathbf{NM},\mathbf{CS}} (\Delta \mid \mathbf{PR}_{\mathbf{GW}}, \mathbf{NM}, \mathbf{CS})}{V(\Delta)} = \frac{V(\mathbf{PR}_{\mathbf{GW}})}{V(\Delta)}, \tag{11}$$

$$S_{\mathbf{PR}_{\mathbf{OVN}}} = \frac{E_{\mathbf{CS}} E_{\mathbf{NM}|\mathbf{CS}} V_{\mathbf{PR}_{\mathbf{OVN}}|\mathbf{NM},\mathbf{CS}} E_{\mathbf{PR}_{\mathbf{VDZ}},\mathbf{PR}_{\mathbf{GW}}|\mathbf{PR}_{\mathbf{OVN}},\mathbf{NM},\mathbf{CS}} (\Delta \mid \mathbf{PR}_{\mathbf{OVN}}, \mathbf{NM\ CS})}{V(\Delta)} = \frac{V(\mathbf{PR}_{\mathbf{OVN}})}{V(\Delta)}. \tag{12}$$

## 2.3 Sensitivity index estimation using the LHS and binning method

The hierarchical sensitivity analysis method proposed by Dai et al. (2017a) was sampled using the conventional Monte Carlo random sampling method, which is computationally expensive for the sensitivity analysis of large-scale PBHMs. In this study, the different parameters were simultaneously sampled by the LHS method (Zhang and Pinder, 2003; Kanso et al., 2006). Compared with the conventional Monte Carlo method, the LHS method can guarantee space-filling and noncollapsing of parameter samples (Grosso et al., 2009; Crombecq et al., 2011; Husslage et al., 2011; Damblin et al., 2013; Ba et al., 2015; Qian, 2012), which means that the sampling points can be evenly distributed throughout the sampling region and that there are no two sampling points with the same value. Thus, LHS is a sampling method with higher sampling efficiency (Helton and Davis, 2003). The convergence rate of the conventional Monte Carlo method is $O(N^{-1/2})$, where $N$ is

the sample size (Caflisch, 1998). However, for a system where the parameters are simply distributed (e.g., uniformly distributed), the convergence rate of LHS can reach $O(N^{-3})$ (Iman and Conover, 1980). The LHS method has been one popular sampling technique used to reduce computational cost.

For the function $Y = f(X)$, the input vector $X$ consists of $k$ parameters (i.e., $X = (X_1, X_2, \ldots, X_k)$). By using the LHS method, the range of $X_i$, $i = 1, 2, \ldots, k$ can be divided into $n$ nonoverlapping intervals with equal probabilities. The $n$ values obtained from $X_1$ are randomly paired with $n$ values obtained from $X_2$; these $n$ paired values are then combined with those $n$ values from $X_3$. We repeat this process until the new $n \times k$ sample matrix $A$ is developed. This sample matrix $A$ can be used to calculate the sensitivity index for the model output. More details regarding LHS are described in previous studies (McKay et al., 1979; Owen, 1998; Helton and Davis, 2003).

Using the variance definition, the partial variance in $V(\mathbf{PR})$ can be first expressed as follows:

$$
\begin{aligned}
V(\mathbf{PR}) &= E_{\mathbf{CS}} E_{\mathbf{NM}|\mathbf{CS}} V_{\mathbf{PR}|\mathbf{NM},\mathbf{CS}} \left( \Delta \mid \mathbf{NM}, \mathbf{CS} \right) \\
&= E_{\mathbf{CS}} E_{\mathbf{NM}|\mathbf{CS}} \left( E_{\mathbf{PR}|\mathbf{NM},\mathbf{CS}} \left( \Delta \mid \mathbf{NM}, \mathbf{CS} \right)^2 - \left( E_{\mathbf{PR}|\mathbf{NM},\mathbf{CS}} \left( \Delta \mid \mathbf{NM}, \mathbf{CS} \right) \right)^2 \right)
\end{aligned}
\tag{13}
$$

After applying the formula of expectation and the LHS method, the terms $V(\mathbf{PR})$, $V(\mathbf{NM})$ and $V(\mathbf{CS})$ can be expressed as follows:

$$
\begin{aligned}
V(\mathbf{PR}) &= E_{\mathbf{CS}} E_{\mathbf{NM}|\mathbf{CS}} \left( E_{\mathbf{PR}|\mathbf{NM},\mathbf{CS}} \left( \Delta \mid \mathbf{NM}, \mathbf{CS} \right)^2 - \left( E_{\mathbf{PR}|\mathbf{ST},\mathbf{CS}} \left( \Delta \mid \mathbf{NM}, \mathbf{CS} \right) \right)^2 \right) \\
&= E_{\mathbf{CS}} E_{\mathbf{NM}|\mathbf{CS}} \left( \frac{1}{n} \sum_{j=1}^{n} \Delta^2 \left( PR_j \mid NM_k, CS_l \right) - \left( \frac{1}{n} \sum_{j=1}^{n} \Delta \left( PR_j \mid NM_k, CS_l \right) \right)^2 \right) \\
&= \sum_l \sum_k \left( \frac{1}{n} \sum_{j=1}^{n} \Delta^2 \left( PR_j \mid NM_k, CS_l \right) - \left( \frac{1}{n} \sum_{j=1}^{n} \Delta \left( PR_j \mid NM_k, CS_l \right) \right)^2 \right) P(NM_k \mid CS_l) P(CS_l)
\end{aligned}
\tag{14}
$$

$$
\begin{aligned}
V(\mathbf{NM}) &= E_{\mathbf{CS}} V_{\mathbf{NM}|\mathbf{CS}} E_{\mathbf{PR}|\mathbf{NM},\mathbf{CS}} \left( \Delta \mid \mathbf{NM}, \mathbf{CS} \right) \\
&= \sum_l P(CS_l) \left( \sum_k \left( \frac{1}{n} \sum_{j=1}^{n} \Delta \left( PR_j \mid NM_k, CS_l \right) \right)^2 P(NM_k \mid CS_l) - \left( \sum_k \left( \frac{1}{n} \sum_{j=1}^{n} \Delta \left( PR_j \mid NM_k, CS_l \right) P(NM_k \mid CS_l) \right) \right)^2 \right)
\end{aligned}
\tag{15}
$$

$$V(\mathbf{CS}) = V_{\mathbf{CS}} E_{\mathbf{NM}|\mathbf{CS}} E_{\mathbf{PR}|\mathbf{NM},\mathbf{CS}}(\Delta \mid \mathbf{NM}, \mathbf{CS})$$

$$= \sum_l P(CS_l) \left( P(NM_k \mid CS_l) \left( \frac{1}{n} \sum_{j=1}^n \Delta_k (PR_j \mid NM_k, CS_l) \right) \right)^2 ,$$

$$- \left( \sum_l \sum_k P(CS_l) P(NM_k \mid CS_l) \left( \frac{1}{n} \sum_{j=1}^n \Delta_k (PR_j \mid NM_k, CS_l) \right) \right)^2 \tag{16}$$

where $n$ and $j$ represent the total sample number of LHS and the index of LHS samples, respectively, $P(NM_k \mid CS_l)$ represents the prior weight of model $NM_k$ under climate scenario $CS_l$ with $\sum_k P(NM_k \mid CS_l) = 1$ and $P(CS_l)$ is the prior weights of different CS satisfying $\sum_l P(CS_l) = 1$. The values of the weights for alternative models or CS could be selected using prior knowledge or objective criteria, e.g., posterior probabilities of the Bayesian theorem (Neumann, 2012; Schöniger et al., 2014).

To calculate the subdivided parametric sensitivity indices, i.e., the sensitivity indices for vadose zone parameters, groundwater parameters, and overland flow parameter, a binning method was implemented in this study. This binning method was designed to estimate the partial variance terms of subdivided parameter groups with paired LHS samples of parameters. Using the sensitivity index of vadose zone parameters as an example, the range of vadose zone parameters was divided into multiple equal bins, and the partial variance term $V_{\mathbf{PR}_{\mathbf{VDZ}}|\mathbf{NM},\mathbf{CS}} E_{\mathbf{PR}_{\mathbf{GW}},\mathbf{PR}_{\mathbf{OVN}}|\mathbf{PR}_{\mathbf{VDZ}},\mathbf{NM},\mathbf{CS}}(\Delta \mid \mathbf{PR}_{\mathbf{VDZ}}, \mathbf{NM}, \mathbf{CS})$ was approximated by $V_{\mathbf{PR}_{\mathbf{VDZ}}^{bin}|\mathbf{NM},\mathbf{CS}} E_{\mathbf{PR}_{\mathbf{GW}},\mathbf{PR}_{\mathbf{OVN}}|\mathbf{PR}_{\mathbf{VDZ}}^{bin},\mathbf{NM},\mathbf{CS}}(\Delta \mid \mathbf{PR}_{\mathbf{VDZ}}^{bin}, \mathbf{NM}, \mathbf{CS})$ using the model outputs calculated by those parameter sample pairs that contain vadose zone parameters in the same bin (noted as $\mathbf{PR}_{\mathbf{VDZ}}^{bin}$). Then, the partial variance term in Eq. (10) can be computed as follows:

$$V_{\mathbf{PR}_{\mathbf{VDZ}}|\mathbf{NM},\mathbf{CS}} E_{\mathbf{PR}_{\mathbf{GW}},\mathbf{PR}_{\mathbf{OVN}}|\mathbf{PR}_{\mathbf{VDZ}},\mathbf{NM},\mathbf{CS}}(\Delta \mid \mathbf{PR}_{\mathbf{VDZ}}, \mathbf{NM}, \mathbf{CS})$$

$$= V_{\mathbf{PR}_{\mathbf{VDZ}}^{bin}|\mathbf{NM},\mathbf{CS}} E_{\mathbf{PR}_{\mathbf{GW}},\mathbf{PR}_{\mathbf{OVN}}|\mathbf{PR}_{\mathbf{VDZ}}^{bin},\mathbf{NM},\mathbf{CS}}(\Delta \mid \mathbf{PR}_{\mathbf{VDZ}}^{bin}, \mathbf{NM}, \mathbf{CS})$$

$$= E_{\mathbf{PR}_{\mathbf{VDZ}}^{bin}|\mathbf{NM},\mathbf{CS}} \left( E_{\mathbf{PR}_{\mathbf{GW}},\mathbf{PR}_{\mathbf{OVN}}|\mathbf{PR}_{\mathbf{VDZ}}^{bin},\mathbf{NM},\mathbf{CS}}(\Delta \mid \mathbf{PR}_{\mathbf{VDZ}}^{bin}, \mathbf{NM}, \mathbf{CS}) \right)^2$$

$$- \left( E_{\mathbf{PR}_{\mathbf{VDZ}}^{bin}|\mathbf{NM},\mathbf{CS}} \left( E_{\mathbf{PR}_{\mathbf{GW}},\mathbf{PR}_{\mathbf{OVN}}|\mathbf{PR}_{\mathbf{VDZ}}^{bin},\mathbf{NM},\mathbf{CS}}(\Delta \mid \mathbf{PR}_{\mathbf{VDZ}}^{bin}, \mathbf{NM}, \mathbf{CS}) \right) \right)^2 \tag{17}$$

The subscript $\mathbf{PR}_{\mathbf{VDZ}}^{bin}|\mathbf{NM}, \mathbf{CS}$ represents the vadose zone parameters in the same bins under the fixed model and fixed climate scenario. The subscript $\mathbf{PR}_{\mathbf{GW}}, \mathbf{PR}_{\mathbf{OVN}}|\mathbf{PR}_{\mathbf{VDZ}}^{bin}, \mathbf{NM}, \mathbf{CS}$ represents the change in the combination of $\mathbf{PR}_{\mathbf{GW}}$ and $\mathbf{PR}_{\mathbf{OVN}}$ sets belonging to a specific $\mathbf{PR}_{\mathbf{VDZ}}$ bin under a fixed model and fixed climate scenario. The term $\Delta|\mathbf{PR}_{\mathbf{VDZ}}^{bin}, \mathbf{NM}, \mathbf{CS}$ represents the output under the fixed vadose zone parameter, subsurface stratigraphy model, and climate scenario. $P(\mathbf{PR}_{\mathbf{VDZ}}^{bin}|\mathbf{NM}, \mathbf{CS})$ refers to the weights of different bins for $\mathbf{PR}_{\mathbf{VDZ}}$ under the fixed model and fixed climate scenario.

The procedures for calculating the subdivided parametric sensitivity indices for $\mathbf{PR}_{\text{VDZ}}$ using the combined LHS and binning methods are listed as follows: (1) simulate $\Delta$ for all $\mathbf{CS}$, models, and parameter realizations, (2) divide the $\mathbf{PR}_{\text{VDZ}}$ realizations into bins, and (3) calculate $E_{\mathbf{PR}_{\text{GW}}, \mathbf{PR}_{\text{OVN}}|\mathbf{PR}_{\text{VDZ}}, \mathbf{NM}, \mathbf{CS}}(\Delta|\mathbf{PR}_{\text{VDZ}}, \mathbf{NM}, \mathbf{CS})$ by replacing it with $E_{\mathbf{PR}_{\text{GW}}, \mathbf{PR}_{\text{OVN}}|\mathbf{PR}_{\text{VDZ}}^{\text{bin}}, \mathbf{NM}, \mathbf{CS}}(\Delta|\mathbf{PR}_{\text{VDZ}}^{\text{bin}}, \mathbf{NM}, \mathbf{CS})$. After $E_{\mathbf{PR}_{\text{GW}}, \mathbf{PR}_{\text{OVN}}|\mathbf{PR}_{\text{VDZ}}^{\text{bin}}, \mathbf{NM}, \mathbf{CS}}(\Delta|\mathbf{PR}_{\text{VDZ}}^{\text{bin}}, \mathbf{NM}, \mathbf{CS})$ is calculated for each bin of $\mathbf{PR}_{\text{VDZ}}$, the partial variance for $\mathbf{PR}_{\text{VDZ}}$, i.e., the molecule of Eq. (10) can be expressed as follows:

$$
\begin{aligned}
V\left(\mathbf{PR}_{\text{VDZ}}\right) &= E_{\mathbf{CS}} E_{\mathbf{NM}|\mathbf{CS}} V_{\mathbf{PR}_{\text{VDZ}}|\mathbf{NM}, \mathbf{CS}} E_{\mathbf{PR}_{\text{GW}}, \mathbf{PR}_{\text{OVN}}|\mathbf{PR}_{\text{VDZ}}, \mathbf{NM}, \mathbf{CS}} \left(\Delta \mid \mathbf{PR}_{\text{VDZ}}, \mathbf{NM}, \mathbf{CS}\right) \\
&= E_{\mathbf{CS}} E_{\mathbf{NM}|\mathbf{CS}} \left\{
\begin{array}{l}
E_{\mathbf{PR}_{\text{VDZ}}^{\text{bin}}|\mathbf{NM}, \mathbf{CS}} \left( E_{\mathbf{PR}_{\text{GW}}, \mathbf{PR}_{\text{OVN}}|\mathbf{PR}_{\text{VDZ}}^{\text{bin}}, \mathbf{NM}, \mathbf{CS}} \left(\Delta \mid \mathbf{PR}_{\text{VDZ}}^{\text{bin}}, \mathbf{NM}, \mathbf{CS}\right)\right)^2 \\
- \left( E_{\mathbf{PR}_{\text{VDZ}}^{\text{bin}}|\mathbf{NM}, \mathbf{CS}} \left( E_{\mathbf{PR}_{\text{GW}}, \mathbf{PR}_{\text{OVN}}|\mathbf{PR}_{\text{VDZ}}^{\text{bin}}, \mathbf{NM}, \mathbf{CS}} \left(\Delta \mid \mathbf{PR}_{\text{VDZ}}^{\text{bin}}, \mathbf{NM}, \mathbf{CS}\right)\right)\right)^2
\end{array}
\right\} \\
&= \sum_l \sum_k \left\{
\begin{array}{l}
\dfrac{1}{W} \sum_w \left( \dfrac{1}{U} \sum_u \Delta\left(\left(\mathbf{PR}_{\text{GW}}, \mathbf{PR}_{\text{OVN}}\right)_u \mid \mathbf{PR}_{\text{VDZ}}^{\text{bin}_w}, NM_k, CS_l\right)\right)^2 \\
- \left( \dfrac{1}{WU} \sum_w \sum_u \Delta\left(\left(\mathbf{PR}_{\text{GW}}, \mathbf{PR}_{\text{OVN}}\right)_u \mid \mathbf{PR}_{\text{VDZ}}^{\text{bin}_w}, NM_k, CS_l\right)\right)^2
\end{array}
\right\} P\left(NM_k \mid CS_l\right) P\left(CS_l\right)
\end{aligned}
\tag{18}
$$

where the symbol $U$ refers to the number of combinations of $\mathbf{PR}_{\text{GW}}$ and $\mathbf{PR}_{\text{OVN}}$ in bin $\mathbf{PR}_{\text{VDZ}}^{\text{bin}_w}$, i.e., the size of the parameter set in bin $\mathbf{PR}_{\text{VDZ}}^{\text{bin}_w}$, and the symbol $u$ is the index for these combinations. $w$ represents the index for the bins of vadose zone parameters, and $W$ is the total number of bins. After applying the LHS sampling method and the same binning method, the partial variance for $\mathbf{PR}_{\text{GW}}$ and $\mathbf{PR}_{\text{OVN}}$ can be estimated as follows:

$$
\begin{aligned}
V\left(\mathbf{PR}_{\text{GW}}\right) &= E_{\mathbf{CS}} E_{\mathbf{NM}|\mathbf{CS}} V_{\mathbf{PR}_{\text{GW}}|\mathbf{NM}, \mathbf{CS}} E_{\mathbf{PR}_{\text{VDZ}}, \mathbf{PR}_{\text{OVN}}|\mathbf{PR}_{\text{GW}}, \mathbf{NM}, \mathbf{CS}} \left(\Delta \mid \mathbf{PR}_{\text{GW}}, \mathbf{NM}, \mathbf{CS}\right) \\
&= \sum_l \sum_k \left\{
\begin{array}{l}
\dfrac{1}{W} \sum_w \left( \dfrac{1}{U} \sum_u \Delta\left(\left(\mathbf{PR}_{\text{VDZ}}, \mathbf{PR}_{\text{OVN}}\right)_u \mid \mathbf{PR}_{\text{GW}}^{\text{bin}_w}, NM_k, CS_l\right)\right)^2 \\
- \left( \dfrac{1}{WU} \sum_w \sum_u \Delta\left(\left(\mathbf{PR}_{\text{VDZ}}, \mathbf{PR}_{\text{OVN}}\right)_u \mid \mathbf{PR}_{\text{GW}}^{\text{bin}_w}, NM_k, CS_l\right)\right)^2
\end{array}
\right\} P\left(NM_k \mid CS_l\right) P\left(CS_l\right),
\end{aligned}
\tag{19}
$$

$$
\begin{aligned}
V\left(\mathbf{PR}_{\text{OVN}}\right) &= E_{\mathbf{CS}} E_{\mathbf{NM}|\mathbf{CS}} V_{\mathbf{PR}_{\text{OVN}}|\mathbf{NM}, \mathbf{CS}} E_{\mathbf{PR}_{\text{VDZ}}, \mathbf{PR}_{\text{GW}}|\mathbf{PR}_{\text{OVN}}, \mathbf{NM}, \mathbf{CS}} \left(\Delta \mid \mathbf{PR}_{\text{OVN}}, \mathbf{NM}, \mathbf{CS}\right) \\
&= \sum_l \sum_k \left\{
\begin{array}{l}
\dfrac{1}{W} \sum_w \left( \dfrac{1}{U} \sum_u \Delta\left(\left(\mathbf{PR}_{\text{VDZ}}, \mathbf{PR}_{\text{GW}}\right)_u \mid \mathbf{PR}_{\text{OVN}}^{\text{bin}_w}, NM_k, CS_l\right)\right)^2 \\
- \left( \dfrac{1}{WU} \sum_w \sum_u \Delta\left(\left(\mathbf{PR}_{\text{VDZ}}, \mathbf{PR}_{\text{GW}}\right)_u \mid \mathbf{PR}_{\text{OVN}}^{\text{bin}_w}, NM_k, CS_l\right)\right)^2
\end{array}
\right\} P\left(NM_k \mid CS_l\right) P\left(CS_l\right)
\end{aligned}
\tag{20}
$$

The binning method is a rigorously derived mathematical technique designed to separate and estimate the partial variances contributed from different parameters of one LHS method sampled parameter set. Because the mathematical equations are general and rigorous, this method can be applied to any modeling case with LHS parameter samplings. However, when the samplings for different parameters are totally random and unrelated, such as the conventional Monte Carlo simulation, the

280 binning method is not applicable. Using LHS and the binning method, the number of realizations is reduced to the size of the parameter sets obtained from the LHS method. Thus, the computation cost for estimating the subdivided parametric indices can be highly reduced. Dai et al. (2017b) confirmed a similar accuracy of 36,000,000 Monte Carlo realization results with 16,000 realizations when applying only the binning method for a synthetic example. The combination of the LHS method with the binning method makes it computationally affordable to analyze the detailed parametric sensitivity for such a large-

285 scale, complex hydrologic model.

### 2.4 The generation of uncertain inputs

For the CS, we generated six typical and alternative scenarios based on NASA's Tropical Measuring Mission (TRMM) data (http://trmm.gsfc.nasa.gov/) and the default CLM CRU-NCEP (CRUNCEP) dataset (Piao et al., 2012) from 1998 to 2013. We considered five climate variables: daily precipitation, temperature, solar radiation, humidity, and wind speed. The

290 precipitation data were obtained from the TRMM, while the temperature, solar radiation, humidity and wind speed data are based on the CRUNCEP because the model fails to capture the peak stream discharges using the CRUNCEP rainfall data (Niu et al., 2017). We first divided the annual climate dataset into dry and wet seasons according to the precipitation values (six months for each season). Then, we sorted the wet and dry seasons according to their total precipitation values during the whole season. Next, we divided these wet and dry seasons into three different groups representing six climate scenarios from

295 wet to dry (Fig. 3). The mean and standard deviation of the values of the different climate variables (e.g., precipitation, maximum temperature) for each group were calculated using the daily data (Table 1). Finally, we generated random daily weather data for each climate scenario based on these mean and standard deviation data using a normal distribution. The mean and standard deviation for each climate scenario's daily data are listed in Table 1, and Fig. 3 displays a box plot of the precipitation data for the six climate scenarios ($CS_1$, $CS_2$, $CS_3$, $CS_4$, $CS_5$, $CS_6$).

**Table 1. Statistical information for the daily data for the six CS. Here, $\mu$ represents the mean value and $\sigma$ represents the standard deviation.**

| climate scenarios | Wet season | | | Dry season | | |
|---|---|---|---|---|---|---|
| | $CS_1$ | $CS_2$ | $CS_3$ | $CS_4$ | $CS_5$ | $CS_6$ |
| precipitation ($\mu$ [mm], $\sigma$) | (10.96, 2.78) | (9.49, 2.8) | (7.87, 2.91) | (4.84, 1.81) | (3.99, 1.62) | (3.38, 1.35) |
| maximum temperature ($\mu$ [°C], $\sigma$) | (29.33, 0.66) | (29.94, 0.60) | (30.03, 0.62) | (30.80, 0.65) | (30.80, 1.03) | (31.50, 1.04) |

| | | | | | |
|---|---|---|---|---|---|
| minimum temperature ($\mu$ [°C], $\sigma$) | (25.13, 0.54) | (25.63, 0.55) | (25.74, 0.48) | (25,59, 0.77) | (25.47, 0.81) | (26.02, 0.82) |
| radiation intensity ($\mu$ [MJ m$^{-2}$], $\sigma$) | (3973.5, 129.6) | (3975.1, 122.9) | (3982.4, 113.6) | (4285.5, 199.1) | (4299.5, 195.6) | (4312.1, 215.8) |
| relative humidity ($\mu$ [unitless], $\sigma$) | (0.0188, 4.65e-4) | (0.0191, 3.54e-4) | (0.0192, 4.72e-4) | (0.0186, 5.44e-4) | (0.0185, 5.76e-4) | (0.0188, 5.75e-4) |
| average wind speed ($\mu$ [m s$^{-1}$], $\sigma$) | (0.595, 0.122) | (0.648, 0.141) | (0.642, 0.148) | (0.549, 0.073) | (0.518, 0.061) | (0.552, 0.081) |

For the model uncertainty, the research of Brunke et al. (2016) shows that the shallow bedrock depth or deep bedrock depth has a great influence on surface runoff and base flow in CLM. Therefore, in this study, we will consider the effects of different aquifer models. Niu et al. (2017) simulated an unconfined aquifer with 100 m depth and 200 m thickness for the confined aquifer. Considering that (1) the stratification of the soil and aquifer is relatively stable, and the thickness does not change much (do Rosário et al., 2016), (2) there is a lack of actual measurements in this area to determine the stratification of unconfined aquifers and confined aquifers, and (3) according to Pelletier et al. (2016), the thickness of the unconfined aquifer in the central Amazon is larger than 50 m, and the depth of the bedrock is very deep, as three aquifer models involving different thicknesses of the unconfined and confined aquifers were generated to investigate the sensitivity of the model outputs to aquifer thickness. These three aquifer models involving different thicknesses of the unconfined and confined aquifers are (1) 100 m and 200 m ($NM_1$), (2) 50 m and 250 m ($NM_2$), and (3) 250 m and 50 m ($NM_3$), respectively. These three models represent the situations of (i) similar thickness of the unconfined and confined aquifer, medium bedrock depth, (ii) thick confined aquifer, low bedrock depth, and (iii) thick unconfined aquifer, large bedrock depth.

The six different model parameters were sampled by LHS within the feasible range (Table 2), and 600 samples of the parameter set were generated. The reasons for using 600 parameter samples in this study are because, first, based on the experiences of previous research cases (Emery et al., 2016; Dai et al., 2019), 600 is an adequate parameter sample size for this research considering the model domain and number of uncertain parameters; and second, considering the computational cost, 600 parameter samples are an appropriate sample size for this study. By combining model uncertainty and climate scenario uncertainty, there are $600 \times 3 \times 6 = 10{,}800$ simulations in total. The pure simulation time without analyzing data is already very time consuming even when using the best high-performance computing (HPC) platform we have.

**Table 2. Six chosen parameters to be included in parameter uncertainty**

| Group | Parameter | Unit | Description | Allowable Range |
|---|---|---|---|---|
| vadose zone (**PR**$_{VDZ}$) | $K_s$ | m day$^{-1}$ | soil saturated hydraulic conductivity | 0.0-10.0 |
| | $\alpha$ | m$^{-1}$ | Van Genuchten parameter | 0.1-4.0 |
| | $N$ | | Van Genuchten parameter | 1.03-5.0 |
| groundwater (**PR**$_{GW}$) | $K_1$ | m day$^{-1}$ | unconfined aquifer hydraulic conductivity | 0.0-60.0 |
| | $K_2$ | m day$^{-1}$ | confined aquifer hydraulic conductivity | 0.0-10.0 |
| overland flow (**PR**$_{OVN}$) | $L$ | m | length of flow path for runoff contribution to the overland flow domain | 20.0-700.0 |

According to the study of Cuartas et al. (2012), the clay content in the northwestern part of Manaus is very high (65-90%). Considering the difference in regional soil texture (Fisher et al., 2008; Teixeira et al., 2014), the allowable range of soil saturated conductivity $K$s selected in this study is between 0-10 m day$^{-1}$. The ranges of unconfined aquifer conductivity, $K_1$,

and confined aquifer conductivity, $K_2$, are chosen as 0-10 m day$^{-1}$ and 0-60 m day$^{-1}$, respectively. The results of the model calibration in Niu et al. (2017), which are related to the characteristics of the soil and groundwater layers in the watershed (Oleson et al., 2008; Christoffersen et al., 2014), are used to define the mean values of distributions used for these uncertain parameters. The soil saturated conductivity, $K_s$, unconfined aquifer conductivity, $K_1$, and confined aquifer conductivity, $K_2$, were assumed to follow lognormal distributions (log-N (1.6094, 0.4214$^2$), log-N (3.4012, 0.4214$^2$), and log-N (1.6094,

0.4214$^2$), respectively). The remaining three parameters ($\alpha$, $N$, and $L$) were assumed to follow a uniform distribution: U (0.1, 4), U (1.03, 5), and U (20, 700). The allowable ranges of these six parameters are listed in Table 2.

From Section 3.1 to Section 3.4, we assumed that the different scenarios have equal probability. Moreover, three models under each climate scenario were also assumed to have equal weights, i.e., $P(CS_l) = 1/6$, and $P(NM_k|CS_l) = 1/3$. However, the weights for models and scenarios may affect the output results. We investigated the variability in the results to the

changing weights for $NM_1$, $CS_1$ (the wettest climate scenario), and $CS_6$ (the driest climate scenario) in Section 3.5. This experiment is helpful for improving our understanding of sensitivity analysis results.

## 3 Results and discussion

### 3.1 Model predictions

As mentioned in the above section, the total number of PAWS+CLM simulations considering all possible combinations of

the three uncertain factors is 6×3×600=10,800, which represents six climate scenarios, three model conceptualizations of

aquifer thickness, and 600 sampled parameter sets. We used the parallel computing technique for running these simulations through the HPC platform (13 cores of Xeon 2.8G CPU). The average time spent on a single simulation was 2.8 minutes, and a total of 10,800 simulations were run for 3 weeks. The simulation time for all the simulations was six months (180 days, 4320 hours), which is the length of the dry or wet season in the central Amazon region. The results given by the PAWS were represented in two forms: (1) space-accumulative output values over the whole grid at each time step and (2) time-accumulative output values over the whole simulated period for each grid. In this study, the time step is one hour. Figure 4 depicts the space-accumulative model predictions for the two outputs of interest, $ET$ and $Q_G$, using different inputs of scenarios, models, and parameter sets. All prediction results are grouped into 24 groups based on local time, which represent 1:00 to 24:00 throughout the day (Fig. 4). Each box in Fig. 4 describes the prediction results estimated using all the combinations of 600 parameter sets, three models and six CS at the same local times. Figure 4(a) shows that the $ET$ predictions throughout a day have a time-varying pattern, and their values are significantly larger during the daytime and smaller at night. This pattern coincides with the physical fact that sunlight leads to higher temperature and more plant transpiration. The uncertainty of $ET$ predictions during the daytime is also larger than that during the night. Figure 4(b) shows that the predictions of $Q_G$ have no significant time-varying pattern throughout the day. However, the prediction results of $ET$ and $Q_G$ both demonstrate great variability or uncertainty for each time group. Further quantitative sensitivity analysis is necessary to identify the most important sources of uncertainty for these predictions.

### 3.2 Sensitivity indices for evapotranspiration

First, we calculated the sensitivity indices for the space-accumulative $ET$ over the whole watershed at all time steps using Eqs. (4)-(6). Figure 5(a) shows the sensitivity indices for the whole simulation period of 4320 time steps. All the sensitivity indices fluctuate strongly with time, except for the sensitivity indices of the models. The sensitivity indices for the models ($S_{NM}$) are always close to zero at every time step, indicating that aquifer thickness has little influence on space-accumulative $ET$. Figure 5(b)-(g) plot the sensitivity indices across six periods, exhibiting the details at each time step. Every period lasts for three days. The patterns of the sensitivity indices have a daily cycle, but specific values of the sensitivity indices at the same wall-clock time on different days are distinguished. Figure 5 indicates that the sensitivity to various factors is strongly time dependent. Notably, at 12:00-13:00, the CS are always the most important factors affecting the sensitivity of $ET$ (Fig. 5(b)-(g)), which may be because $ET$ is directly influenced by solar radiation values, and the radiation forcing used in this study reaches its maximum value at approximately 12:00. Therefore, the CS dominate the uncertainties at 12:00-13:00. Another finding is that at 24:00-1:00, the sensitivity indices for the parameters ($S_{PR}$) show absolute dominance, but the sensitivity indices for the climate scenarios ($S_{CS}$) are decreased. A possible explanation for this result might be that precipitation and radiation forcing all decrease to zero during this period, leading to a decrease in the sensitivity indices for the climate scenarios ($S_{CS}$). In contrast, the importance of parameters is greatly increased. Six time points (simulation times = 1428 hours, 1440 hours, 2868 hours, 2880 hours, 4308 hours, and 4320 hours) were chosen as examples to show the specific sensitivity indices (Fig. 6). Simulation times of 1428 hours, 2868 hours, and 4320 hours belonged to different days,

but all corresponded to 12:00 local time. At these time points, the climate scenario uncertainty ($S_{CS}$) is the most important contributor to the total $ET$ prediction uncertainty, accounting for 54-77% of the total uncertainty, and parameters ($S_{PR}$) contribute the second most to uncertainty. However, at different time points (1440 hours, 2880 hours, and 4320 hours, corresponding to 24:00 local time), the parameters are the dominant uncertainty contributor, with $S_{PR}$ ranging from 89 to 92%.

We also calculated the sensitivity indices for every grid cell within the model domain using the time-accumulative $ET$ predictions over all simulation periods (4320 hours). Figure 7 shows the spatial variability in the sensitivity indices for the temporal mean $ET$ predictions. The maps demonstrate that for most grids, parameters are the most important uncertainty contributor to $ET$ predictions ($S_{PR}>0.50$), and CS are the second most important contributor to uncertainty. However, for stream grid cells, the importance of aquifer thicknesses increases. Therefore, the parameters and aquifer thicknesses are both important. Here, aquifer thicknesses refer to the average aquifer thickness for the whole watershed. The increase in the model sensitivity indices indicates that the structure of the aquifer significantly affects the baseflow and then influences the river evaporation predictions. Figure 7 shows that the parameter uncertainty within the overall watershed is important for $ET$, and for river evaporation, the aquifer thicknesses are also important.

### 3.3 Sensitivity indices for groundwater contribution to streamflow

Groundwater has been demonstrated to be crucial for soil moisture in the Amazon region by previous research) (Miguez-Macho and Fan, 2012b). Meanwhile, it also exerts a significant buffering effect on maintaining evapotranspiration during dry seasons (Miguez-Macho and Fan, 2012a; Pokhrel et al., 2014). The model PAWS uses the output of $Q_G$ to quantify the variation in groundwater volumes and measure the interaction process between groundwater and rivers. It is essential to implement the sensitivity analysis to investigate which factor is most influential on this groundwater exchange process. The same sensitivity analysis procedures were also conducted for the model predictions of $Q_G$.

Figure 8(a) shows the sensitivity indices for the whole simulation period of 4320 time steps for $Q_G$ predictions. This figure indicates that regardless of the time steps, parameters are always the dominant contributor to the total $Q_G$ prediction uncertainty. This result may be explained by the fact that soil parameters strongly affect the soil water redistribution process, including infiltration into groundwater. We selected the same period as Fig. 5(b)-(g) to display the more detailed results for $Q_G$ predictions in Fig. 8(b)-(g). As shown in these figures, the sensitivity indices of the models ($S_{NM}$) and climate scenarios ($S_{CS}$) reach peak values at approximately 1:00. In terms of $S_{CS}$, this may be because the exchange between groundwater and river flow occurs hours later than the rainfall process, and the amount of water during the exchange process always reaches its peak at night, at approximately 1:00. The $S_{NM}$ might be because the thickness of aquifers will greatly influence the water redistribution process in the aquifer. Another pattern demonstrated in Fig. 8 is that the values of $S_{CS}$ generally increase with time. This trend may be caused by the seasonality effect of CS or the long-term cumulative influence of CS on the groundwater flow.

Because groundwater exchange with stream flow occurs only at grid cells along the streams, the sensitivity indices only have valid values in those stream grid cells (Fig. 9). Our results indicate that considering most grid cells, the parameters are the most important contributor to the uncertainty of time-accumulative $Q_G$ predictions, and the second most important factor is aquifer thickness. However, if we divide the grid cells into groundwater and stem river grid cells based on their location relative to the river and aquifer type, the sensitivity analysis results are totally different in these two types of grid cells. The model parameter uncertainty is usually the most important in stem river grid cells; in contrast, the aquifer thicknesses contribute the largest portion of the uncertainty in groundwater grid cells. This pattern of results may be caused by the unconfined aquifer and river being unconnected in the stem river grid cells, and there is an unsaturated zone between two of them. Therefore, the soil parameters affect $Q_G$ predictions in stem river grid cells. Moreover, the groundwater table is relatively high, and the groundwater is directly connected with rivers in the groundwater grid cells. Thus, the aquifer thicknesses are more important under this condition.

### 3.4 Sensitivity indices for subdivided parameters

Based on the sensitivity analysis for $ET$ and $Q_G$ predictions, the results show that parameters are important uncertain inputs for both the space-accumulative and time-accumulative uncertainties. In this study, we used Eqs. (10)-(12) to further calculate the subdivided parametric sensitivity indices, which can provide a more detailed sensitivity analysis for model simulation. Through this investigation, the parametric sensitivity was subdivided into three groups: (1) the sensitivity for vadose zone parameters ($S_{PR_{VDZ}}$), (2) the sensitivity for groundwater parameters ($S_{PR_{GW}}$), and (3) the sensitivity for the overland flow parameter ($S_{PR_{OVN}}$). Using the binning method, we calculated the space-accumulative and time-accumulative subdivided parametric sensitivity indices for $ET$ and $Q_G$. We plotted frequency histograms of the subdivided parametric sensitivity indices over 4320 hours in Fig. 10.

Figure 10(a) depicts the results for $ET$. The value of $S_{PR_{VDZ}}$ is concentrated in the range of 0.1-0.9, and $S_{PR_{GW}}$ is concentrated in the range of 0.003-0.032. The value of $S_{PR_{OVN}}$ is so small that the influence of the overland flow parameter can be ignored. This indicates that vadose zone parameters ($PR_{VDZ}$) dominate the total parametric uncertainties for $ET$. Figure 10(b) shows the frequency histogram of space-accumulative subdivided parametric sensitivity results for $Q_G$. $S_{PR_{VDZ}}$ is still concentrated in the larger number range (0.2-0.53), and the value of $S_{PR_{GW}}$ changes from 0.04 to 0.3. The number of $S_{PR_{OVN}}$ is also the lowest, indicating that the overland flow parameter has little effect on $Q_G$. The order of importance of the uncertain inputs is the same for both $ET$ and $Q_G$ predictions. However, it is significantly different from $ET$ in that although $PR_{GW}$ is the second most important parameter group, the value of $S_{PR_{GW}}$ in the $Q_G$ results is an order of magnitude higher than that in the $ET$ results. In the $Q_G$ results, the range of $S_{PR_{GW}}$ is concentrated in the range of 0.05-0.2, while in the $ET$ results, the value of $S_{PR_{GW}}$ is concentrated in the range of 0.003-0.032.

We plotted the time-accumulative subdivided parametric sensitivity indices for $ET$ in Fig. 11(a) and for $Q_G$ in Fig. 11(b). Considering $ET$ as our output, for most grids, the vadose zone parameters are the most important contributor to parametric

uncertainties. Compared with that on other grids, the influence of groundwater parameters on the river grids is more significant (Fig. 11(a)). For the $Q_G$ results, the vadose zone parameters generally dominate the parametric sensitivities for

most grids (Fig. 11(b)). However, if considering different types of grid cells, we find that the vadose zone parameters mainly affect the stem river grid cells and have a relatively small influence on the groundwater grid cells. This pattern coincides with our hypotheses that there is an unsaturated zone between the stem rivers and groundwater. More detailed sensitivity indices for all six parameters are demonstrated in Appendix C.

### 3.5 Effects of prior weights on sensitivity indices

In this section, we changed the prior weights of the CS and numerical models to investigate their influences on the space-accumulative sensitivity indices. Because the number of space-accumulative results for $ET$ and $Q_G$ is too large to be well exhibited, we chose one time step (4308 hours, 12:00 wall-clock time) to show the trend. We randomly changed the values of the weights for $NM_1$ (the thickness of the unconfined aquifer is 50 m and that of the confined aquifer is 250 m), $CS_1$ (the wettest climate scenario), and $CS_6$ (the driest climate scenario) to between 0 and 1. If the weight for $NM_1$, $CS_1$, or $CS_6$ is $p$,

then the weight of the remaining climate scenarios or models will be assumed to be $(1-p)/n$, where $n$ is the number of the remaining climate scenarios or models. Figure 12(a) indicates that when we consider $ET$ as our output, with the increase in the prior weight of $NM_1$, the uncertainty of the CS will decrease to 50%, while the uncertainty of the parameters will increase to 50%. Both parameters and CS have important effects on $ET$. Different from the results for $ET$, with the increase in the prior weights of $NM_1$, the sensitivity index of the numerical models for $Q_G$ decreases to 0 (because only one model exists

under this condition), and the scenario uncertainty changes only slightly. Moreover, the uncertainty of parameters always dominates the total uncertainty for $Q_G$ (Fig. 12(b)) regardless of the prior weight value. In general, the different prior weight values for the numerical models only slightly change the sensitivity analysis results.

Figure 12(c)-(f) exhibit the influences of prior weights for the wettest and driest CS on $ET$. These figures first demonstrate that changing the values of the prior weights of $CS_1$ and $CS_6$ has larger impacts on $ET$ predictions than on $Q_G$ predictions.

This pattern coincides with the fact that the parameter uncertainty dominates the total predictive uncertainty of $Q_G$ and that the scenario uncertainty is relatively small. Therefore, the selection of prior weight values for the scenarios does not have a significant effect on the sensitivity analysis results for the $Q_G$ predictions, and the parameter sensitivity index is always the largest (Fig. 12(d) and (f)). For the sensitivity analysis results pertaining to $ET$ predictions, changing the values of the weights for $CS_1$ and $CS_6$ has different effects. The sensitivity index values of the climate scenarios for $ET$ predictions

monotonically decrease, while the importance of parameters continues to increase as the prior weight of $CS_1$ is larger than 40%, which reflects that when the probability of extreme humid seasons in the Amazon is greater than 40%, the importance of parameters takes precedence over the importance of climate scenarios for $ET$. However, the value of $S_{CS}$ for $ET$ predictions first increases when the prior weight of $CS_6$ approaches 10% and then decreases after the prior weight of $CS_6$ approaches 90%, and $S_{PR}$ shows the opposite trend (Fig. 12(e)). This shows that when the probability of occurrence of

extreme dry seasons is between 10% and 90%, the climate scenario is always the most important uncertain input unless the occurrence probability of the extreme dry season is greater than 90%.

### 3.6 Discussion

The results from this case study exhibit the importance of parameters, especially the vadose zone parameters, for $ET$ and $Q_G$ predictions. Furthermore, according to the space-accumulative results, the climate scenario is also an important uncertainty source for $ET$ predictions, especially at 12:00. Meanwhile, the thickness of the aquifer has a nonignorable influence on the $Q_G$ predictions on the groundwater grid cells. Moreover, according to the results of adjusting the climate scenario and model weights, the change in model (aquifer thickness) weights only has a small impact on the importance of different uncertainties. When the probability of occurrence of the extreme humid season is high, the importance of the parameters increases significantly. However, when the probability of occurrence of the extreme dry season is high, the main factors affecting $ET$ predation are still the climate scenario unless the probability of occurrence of CS is greater than 90%. Although these patterns of sensitivity analysis results may not be universally correct, they can still provide useful insights for other modelers with similar cases and models.

In addition to the specific results, we also have some new insights into the general patterns of sensitivity analysis for the PBHMs provided by this pilot case. For instance, first, the ranks of importance of uncertain inputs are totally different for different model outputs, e.g., CS have a large impact on $ET$ predictions but a small impact on $Q_G$ predictions. There is no one set of results that are valid for all different model outputs. Second, the sensitivity analysis results of $ET$ and $Q_G$ predictions show that the uncertainty has high temporal and spatial variability, which reflects that for very complex hydrological models, such as PBHMs, it is incorrect to generalize the sensitivity analysis results of a grid or a timestep to the entire watershed or the entire simulation cycle. Third, it is necessary to implement such a comprehensive global sensitivity analysis method that considers more than parametric uncertainty for the large-scale PBHMs since the sensitivity analysis results showed that other sources of uncertainty (e.g., climate scenario and model uncertainties) are essential as well for model predictions. Finally, evaluating the sensitivity of the parameters in detail is essential for PBHMs. For such a complex surface-subsurface coupling model, the new sensitivity analysis method can efficiently identify the uncertain inputs that have the greatest impact on the model outputs. This process can greatly improve our understanding of the complex model system and save time that is normally spent calibrating the model.

### 4 Conclusions

This research presented an improved hierarchical sensitivity analysis method for comprehensive global sensitivity analysis of large-scale complex PBHMs. Developed based on the previous hierarchical framework of Dai et al., (2017a), this new methodology can simultaneously consider various types of uncertainty sources and estimate the importance of different processes involved in the modeling work. A new set of sensitivity indices of subdivided parameters was defined to quantify

the importance of processes that only involve partial parameters. The highly efficient sampling algorithm of the LHS and binning method were implemented for the estimation of sensitivity indices to reduce computational cost. For evaluation and demonstration purposes, we implemented the new sensitivity analysis method into a real-world case of large-scale complex PBHM (PAWS), which was applied to a large Amazon catchment. Three common groups of uncertainty sources or uncertain inputs were considered in this study, including six CS, three plausible aquifer models, and six uncertain parameters (i.e., soil saturated conductivity, van Genuchten $\alpha$ and $N$, unconfined aquifer conductivity, confined aquifer conductivity, and the length of the flow path for runoff contribution to the overland flow domain). A new set of subdivided parametric sensitivity indices was defined for three groups of parameters (i.e., vadose zone, groundwater, and overland flow parameters).

The sensitivity analysis results in this study first demonstrate the necessity of implementing such a comprehensive global sensitivity analysis for PBHMs because uncertainty sources other than parameters (e.g., CS and models) are also important for model predictions. Furthermore, the values of model weights have a small impact on the sensitivity analysis results, but the selections of weights for extreme CS may change the ranks of importance for uncertain inputs. Moreover, the sensitivity analysis results are both temporally and spatially dependent and have distinct patterns for different model outputs. Therefore, there is no single conclusion for all model outputs considering different times and locations. In general, the parameter uncertainty is important for both $ET$ and $Q_G$ predictions. Among all the parameters, the vadose zone parameters are the most important, and the parameter of overland flow is negligible. The CS are also important uncertainties for $ET$ predictions, especially at 12:00. Along the river grid cells, the thickness of the aquifer has a significant influence on both $ET$ and $Q_G$ predictions. Although the patterns of sensitivity analysis results found in this study may not be universally valid, they can still provide useful insights for modelers with similar problems. For instance, we can suggest that when modelers apply sophisticated hydrological simulators, such as the PAWS, they should pay more attention to the values of weather variables at approximately 12:00 (the daily peak values) and focus more on estimating the thicknesses of groundwater aquifers near rivers and adjusting the vadose zone parameters.

Through this pilot example of comprehensive global sensitivity analysis, this study proves that using the new improved hierarchical sensitivity analysis method is a computationally affordable and useful way to identify the most important uncertain inputs for large-scale complex PBHMs. The sensitivity analysis results can provide key information on uncertainty sources for modelers and greatly improve the model calibration and uncertainty analysis processes. The proposed method is mathematically rigorous and general and can be applied to extensive, large-scale hydrological or environmental models with different sources of uncertainty.

**Appendix A**

The governing equations of the PAWS are presented in detail in Shen and Phanikumar (2010) and Shen et al. (2013). Here, we will mainly introduce the equations describing the processes involved in this article.

In the PAWS, the soil moisture in the vadose zone is calculated according to the Richards equation. The vertical movement of fluid between saturated and unsaturated soil is calculated based on the mixed form of the Richards equation (Celia et al., 1990; van Dam and Feddes, 2000):

$$535 \quad C(h)\frac{\partial h}{\partial t} = \frac{\partial}{\partial z}\left[K(h)\left(\frac{\partial h}{\partial z}+1\right)\right]+W(h). \tag{A1}$$

where $h$ represents the soil water pressure head, $z$ is the elevation (positive upward), $K(h)$ represents the soil unsaturated conductivity and $W(h)$ is the source or sink term, including the influence of evaporation, root extraction and lateral flow. The differential water capacity can be described as $C(h)=\partial\theta/\partial h$, where $h$ is the soil pressure head and $\theta$ is the water content. The pressure head, $h$, is related to the unsaturated hydraulic conductivity, $K(h)$. According to the Mualem-van Genuchten formula
(Mualem, 1976; van Genuchten, 1980), the soil saturated hydraulic conductivity, $K_s$, van Genuchten $\alpha$ and $N$ will influence the unsaturated conductivity, $K(h)$:

$$S = \frac{\theta(h)-\theta_r}{\theta_s-\theta_r} = \left(1+|\alpha h|^N\right)^{-(N-1)/N}, \tag{A2}$$

$$K(h) = K_s S^\lambda \left[1-\left(1-S^{N/(N-1)}\right)^{(N-1)/N}\right]^2, \tag{A3}$$

where $S$ is the relative saturation, $\theta_s$ is the saturated water content, $\theta_r$ is the residual water content, $N$ is related to the pore-
size distribution, $\alpha$ indicates the reciprocal of air suction and $\lambda$ is a parameter measuring pore connectivity.

The aquifers in the PAWS are depicted as a series of 2-D layers (Shen et al., 2014). In each layer, the 2-D groundwater equation is used to describe the water movement:

$$S\frac{\partial H}{\partial t} = \frac{\partial}{\partial x}\left[T\left(\frac{\partial H}{\partial x}\right)\right]+\frac{\partial}{\partial y}\left[T\left(\frac{\partial H}{\partial y}\right)\right]+R+W-Dp, \tag{A4}$$

where $S$ is the storability; $T$ is the transmissivity of the aquifer; $T=Kb$, where $K$ is the aquifer conductivity and $b$ is the
saturated thickness of the aquifer; $H$ is the aquifer hydraulic head; $R$ is recharge or discharge; $W$ is the source and sink term; and $Dp$ is percolation into deeper aquifers.

The PAWS applies one-dimensional diffusive wave equations to portray the channel flow model (Shen and Phanikumar, 2010; Shen et al., 2014). After calculating the channel flow, the exchange between groundwater and channel flow ($Q_G$) is immediately computed. The calculation of $Q_G$ is based on the leakance concept (Shen and Phanikumar, 2010):

$$555 \quad \frac{h_r^{n+1}-h_r^*}{\Delta t} = K_r\frac{H^*-(Z_b+h_r^{n+1})}{\Delta Z_b}, \tag{A5}$$

where $h_r^*$ is the river level calculated from the channel flow model, $K_r$ is the riverbed conductivity, $Z_b$ is the elevation of the riverbed, $\Delta Z_b$ is the thickness of the riverbed and $H^*$ is the groundwater table. Note that $H^*$ can also be described as Eq. (A5). By solving these two equations together, we can obtain $H^*$ and $h_r^{n+1}$. Then, the value of $Q_G$ can be calculated as follows (Shen and Phanikumar, 2010):

$$Q_G = w\left(h_r^{n+1} - h_r^*\right), \tag{A6}$$

where $w$ is the wetted perimeter. If the river width is greater than 10 m, $w$ can be approximated as the river width.

The PAWS retains its own flow scheme, but the surface processes use the CLM 4.0 model, which enables the simulation of detailed surface processes, such as surface heat flux, water vapor flux, surface radiation balance, crop growth, and plant photosynthesis. The calculation of $ET$ demand is performed in the CLM model based on the climate data, and then, $ET$ demand will be transferred to PAWS as a source term for the vadose zone. More details about the calculation of $ET$ (both evaporation and transpiration information can be found in the technical note of CLM 4.0, http://www.cesm.ucar.edu/models/cesm1.1/clm/CLM4_Tech_Note.pdf). The coupling with the CLM makes the PAWS a more comprehensive and robust surface-subsurface hydrological model.

## Appendix B

For the model: $\Delta = f(X) = f(X_1, ..., X_m)$, where $\Delta$ is the model output and $X = \{X_1, ..., X_m\}$ is a group of uncertainty inputs, using the law of total variance, the total variance in $\Delta$ can be decomposed as follows (Dai et al., 2017a):

$$V(\Delta) = V_{X_i}\left(E_{\mathbf{X}_{\sim i}}\left(\Delta | X_i\right)\right) + E_{X_i}\left(V_{\mathbf{X}_{\sim i}}\left(\Delta | X_i\right)\right), \tag{B1}$$

where the first term of partial variance on the right-hand side is the within-$X_i$ partial variance and represents the variance contribution by $X_i$ and $\mathbf{X}_{\sim i}$ represents all the inputs except $X_i$. The second term on the right-hand side represents the variance contributed by the model inputs excluding $X_i$ as well as the interactions of all the inputs. Based on the definition of the first-order sensitivity index (Saltelli and Sobol, 1995),

$$S_i = \frac{V_{X_i}(E_{\mathbf{X}_{\sim i}}(\Delta | X_i))}{V(\Delta)}, \tag{B2}$$

The percentage of uncertainty contributed by input $X_i$ can be accurately quantified.

For the hierarchical framework in Fig. 2, the variance-based sensitivity analysis method enables decomposition of the total variance into individual contributors as follows:

$$V(\mathbf{\Delta}) = V_{\mathbf{CS}}\left(E_{\sim\mathbf{CS}|\mathbf{CS}}\left(\mathbf{\Delta}|\mathbf{CS}\right)\right) + E_{\mathbf{CS}}\left(V_{\sim\mathbf{CS}|\mathbf{CS}}\left(\mathbf{\Delta}|\mathbf{CS}\right)\right)$$
$$= V_{\mathbf{CS}}\left(E_{\mathbf{NM},\mathbf{PR}|\mathbf{CS}}\left(\mathbf{\Delta}|\mathbf{CS}\right)\right) + E_{\mathbf{CS}}\left(V_{\mathbf{NM},\mathbf{PR}|\mathbf{CS}}\left(\mathbf{\Delta}|\mathbf{CS}\right)\right) \qquad \text{(B3)}$$

The first term of partial variance on the right-hand side of this equation represents the variance caused by multiple CS. The second term on the right-hand side is the partial variance caused by other uncertain inputs and can be further decomposed as follows:

$\qquad V_{\mathbf{NM},\mathbf{PR}|\mathbf{CS}}\left(\mathbf{\Delta}|\mathbf{CS}\right) = V_{\mathbf{NM}|\mathbf{CS}}\left(E_{\mathbf{PR}|\mathbf{NM},\mathbf{CS}}\left(\mathbf{\Delta}|\mathbf{NM},\mathbf{CS}\right)\right) + E_{\mathbf{NM}|\mathbf{CS}}\left(V_{\mathbf{PR}|\mathbf{NM},\mathbf{CS}}\left(\mathbf{\Delta}|\mathbf{NM},\mathbf{CS}\right)\right),$ (B4)

where the first partial variance term on the right-hand side of this equation represents the uncertainty contributed by multiple plausible models. The second term represents the within-model partial variance caused by the uncertain parameters. By substituting Eq. (B4) back into Eq. (B3), we can obtain the following equation:

$$V(\mathbf{\Delta}) = E_{\mathbf{CS}}\left(E_{\mathbf{NM}|\mathbf{CS}}V_{\mathbf{PR}|\mathbf{NM},\mathbf{CS}}\left(\mathbf{\Delta}\mid\mathbf{NM},\mathbf{CS}\right) + V_{\mathbf{NM}|\mathbf{CS}}E_{\mathbf{PR}|\mathbf{NM},\mathbf{CS}}\left(\mathbf{\Delta}\mid\mathbf{NM},\mathbf{CS}\right)\right)$$
$$+ V_{\mathbf{CS}}E_{\mathbf{NM}|\mathbf{CS}}E_{\mathbf{PR}|\mathbf{NM},\mathbf{CS}}\left(\mathbf{\Delta}\mid\mathbf{NM},\mathbf{CS}\right)$$
$$= E_{\mathbf{CS}}E_{\mathbf{NM}|\mathbf{CS}}V_{\mathbf{PR}|\mathbf{NM},\mathbf{CS}}\left(\mathbf{\Delta}\mid\mathbf{NM},\mathbf{CS}\right) + E_{\mathbf{CS}}V_{\mathbf{NM}|\mathbf{CS}}E_{\mathbf{PR}|\mathbf{NM},\mathbf{CS}}\left(\mathbf{\Delta}\mid\mathbf{NM},\mathbf{CS}\right) \qquad \text{(B5)}$$
$$+ V_{\mathbf{CS}}E_{\mathbf{NM}|\mathbf{CS}}E_{\mathbf{PR}|\mathbf{NM},\mathbf{CS}}\left(\mathbf{\Delta}\mid\mathbf{NM},\mathbf{CS}\right)$$
$$= V(\mathbf{PR}) + V(\mathbf{NM}) + V(\mathbf{CS})$$

The three terms on the right-hand side of Eq. (B5) represent the partial variances contributed by the parameters, models and CS, respectively. The equation indicates that the total variance can be decomposed into the variances contributed by the alternative climate scenarios, **CS**, plausible numerical models, **NM**, and uncertain parameters, **PR**. Then, following the first-order sensitivity index definition (Eq. (B2)), the hierarchical sensitivity analysis method defines the indices for **PR**, **NM**, and **CS**, respectively, as shown in Eqs. (4)-(6).

**Appendix C**

To conduct a more comprehensive analysis of all parameters and to compare the impact of two aquifers on $Q_{\mathrm{G}}$, we estimated the sensitivity indices of the six parameters according to Eq. (C1). The difference between this equation and the previous ones is that Eq (C1) no longer groups the parameters, and it calculates the sensitivity indices individually for six parameters.

$$S_{\theta} = \frac{E_{\mathbf{CS}}E_{\mathbf{NM}|\mathbf{CS}}V_{\theta|\mathbf{NM},\mathbf{CS}}E_{\sim\theta|\theta,\mathbf{NM},\mathbf{CS}}\left(\mathbf{\Delta}\mid\theta,\mathbf{NM},\mathbf{CS}\right)}{V(\mathbf{\Delta})} = \frac{V(\theta)}{V(\mathbf{\Delta})}, \qquad \text{(C1)}$$

In this equation, $\theta$ refers to one of the six parameters, i.e., $K_{\mathrm{s}}$, $\alpha$, $N$, $K_1$, $K_2$ and $L$. The subscript $\theta|\mathbf{NM},\mathbf{CS}$ represents the change in one parameter under a fixed model and a climate scenario. The subscript $\sim\theta|\mathbf{PR}_{\mathrm{VDZ}},\mathbf{NM},\mathbf{CS}$ refers to the other

five uncertain parameter inputs excluding $\theta$ parameter. The term, $\Delta|\theta$, **NM, CS** represents the output under the fixed $\theta$, model, and climate scenario.

The spatial distribution of the sensitivity indices of six parameters for $Q_G$ is shown in Figure (C1). According to Figure (C1), the importance of the van Genuchten parameter, $N$, in the stem grid cells is significant. The conductivity of unconfined aquifer $K_1$ has a certain impact on $Q_G$ in most river grid cells. Additionally, it can also be seen from Figure (C1) that for most grids, the influence of $K_1$ is greater than $K_2$, which implies that the unconfined aquifer has a greater influence on baseflow.

**Data availability**

The weather dataset is available from default CLM CRU–NCEP (CRUNCEP) dataset and http:// trmm.gsfc.nasa.gov/. The sensitivity data and the source code used for analysing the sensitivity data in this study is freely available upon request.

**Author contribution**

HD and JN were involved in the project conceptualization and formulating the methodology. HL performed the experiments, analyzed and visualize the data with guidance from HD and JN. HL prepared the original paper, with contributions from HD, JN, HQ and WR. DG, BXH, MY, JZ, CW and XC helped shape the initial ideas for this research.

**Competing interests**

The authors declare that they have no conflict of interest.

**Acknowledgements**

This work was supported by the National Natural Science Foundation of China (Grant No. 41807182), Western Light—western region leading scientists supporting project, Chinese Academy of Sciences (2018-XBYJRC-002).

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

**Figure captions**

**Figure 1.** Two-dimensional map of the watershed used in this study, showing the elevation, channels and watershed boundary. The study area extends from 1°57′36″S to 2°56′0″S and 59°14′48″W to 60°20′0″W.

**Figure 2.** The framework of the hierarchical sensitivity analysis developed for the PAWS and applied to the central Amazon basin. The three uncertainty source types are placed into the appropriate hierarchical level according to their dependence relationships. The left part of this figure shows the sources of these uncertainties, and the right side shows the abbreviations and the structural relationships among the various uncertainties. The number of CS in this study is six; the number of plausible numerical models under each climate scenario is three; and the number of parameter sets under each numerical model is 600. Notably, the parameter uncertainty sources are further divided into three parts: vadose zone parameters, groundwater parameters and the overland flow parameter.

**Figure 3.** We identified six CS based on precipitation data for 1998-2013 from NASA's TRMM data (http://trmm.gsfc.nasa.gov/). The first climate scenario ($CS_1$) is the wettest one, and the sixth climate scenario ($CS_6$) is the driest one.

**Figure 4.** The spatial-accumulative outputs for evapotranspiration (*ET*) (a) and groundwater contribution to stream flow ($Q_G$) (b) at all time steps and considering all three uncertainties. Each time step is divided into different groups based on local time. Different groups represent different hours of the day.

**Figure 5.** Estimated sensitivities for the spatially averaged evapotranspiration (*ET*) at whole time steps (a). We chose six periods at three-day intervals to display the sensitivity index values in detail. The bottom six figures exhibit the sensitivity index results for 241-312 hours (b), 961-1032 hours (c), 1681-1752 hours (d), 2401-2472 hours (e), 3121-3192 hours (f), and 3841-3912 hours (g). $S_{PR}$ is the sensitivity index for parameters. $S_{NM}$ is the sensitivity index for models and represents the influence of aquifer thickness. The $S_{CS}$ is the sensitivity index for climate scenarios. The bottom x-axis of (b)-(g) represents the simulated time steps, and the upper x-axis of (b)-(g) represents the local time.

**Figure 6.** Estimated sensitivities for the spatially averaged evapotranspiration (*ET*) at 6 time points (simulation times = 1428 hours (Day 60, 12:00), 1440 hours (Day 60, 24:00), 2868 hours (Day 120, 12:00), 2880 hours (Day 120, 24:00), 4308 hours (Day 180, 12:00), and 4320 hours (Day 180, 24:00)). $S_{PR}$ is the sensitivity index for the parameters. $S_{NM}$ is the sensitivity index for the numerical models, and $S_{CS}$ is the sensitivity index for the climate scenarios.

**Figure 7.** Maps of parametric ($S_{PR}$), numerical model ($S_{NM}$), and climate scenario ($S_{CS}$) sensitivity index values for time-averaged evapotranspiration (*ET*) predictions.

**Figure 8.** Estimated sensitivities for the spatially averaged groundwater contribution to stream flow ($Q_G$) at whole time steps (a). We chose six periods at three-day intervals to display the sensitivity index values in detail. The bottom six figures exhibit the sensitivity index results for 241-312 hours (b), 961-1032 hours (c), 1681-1752 hours (d), 2401-2472 hours (e), 3121-3192 hours (f), and 3841-3912 hours (g). $S_{PR}$ is the sensitivity index for parameters. $S_{NM}$ is the sensitivity index for models and represents the influence of aquifer thickness. The $S_{CS}$ is the sensitivity index for climate scenarios. The bottom x-axis of (b)-(g) represents the simulated time steps, and the upper x-axis of (b)-(g) represents the local time.

**Figure 9.** Maps of parametric sensitivity indices ($S_{PR}$), numerical model sensitivity indices ($S_{NM}$), and climate scenario sensitivity indices ($S_{CS}$) for the time-averaged groundwater contribution to stream flow ($Q_G$) predictions.

**Figure 10.** Frequency histograms of subdivided parametric sensitivity indices for spatially averaged results over all 4320 time steps. The results for evapotranspiration (*ET*) as our output are depicted in (a), and the results for groundwater contribution to stream flow ($Q_G$) as our output are depicted in (b). $PR_{VDZ}$ represents the vadose zone parameters. $PR_{GW}$ represents the groundwater parameters. $PR_{OVN}$ represents the overland flow parameter.

**Figure 11.** Maps of vadose zone parameter sensitivity indices ($S_{PR_{VDZ}}$), groundwater parameter sensitivity indices ($S_{PR_{GW}}$) and overland flow parameter sensitivity indices ($S_{PR_{OVN}}$) for time-averaged evapotranspiration (*ET*) (a) and groundwater contribution to stream flow ($Q_G$) (b) predictions.

**Figure 12. Patterns of $S_{PR}$ , $S_{NM}$ , and $S_{CS}$ for space-averaged evapotranspiration (*ET*) and space-averaged groundwater contribution to stream flow ($Q_G$) with changes in the prior weights of numerical model $NM_1$, climate scenario $CS_1$ and climate scenario $CS_6$ at the time step of 4308 hours (at 12:00).**

**Figure C1. Maps of six parameter sensitivity indices for groundwater contribution to stream flow ($Q_G$) predictions.**

**Figures**

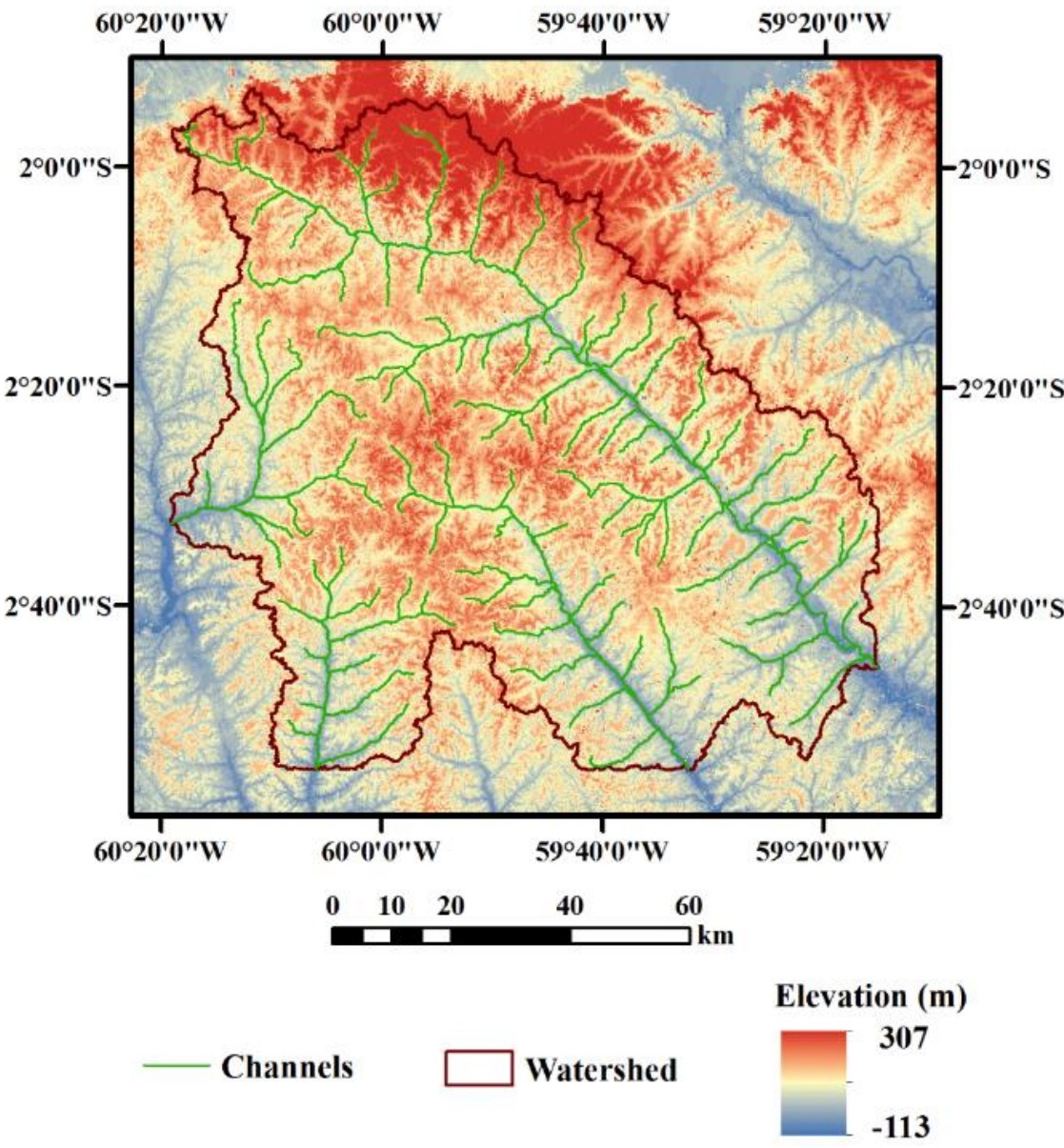

**Figure 1**

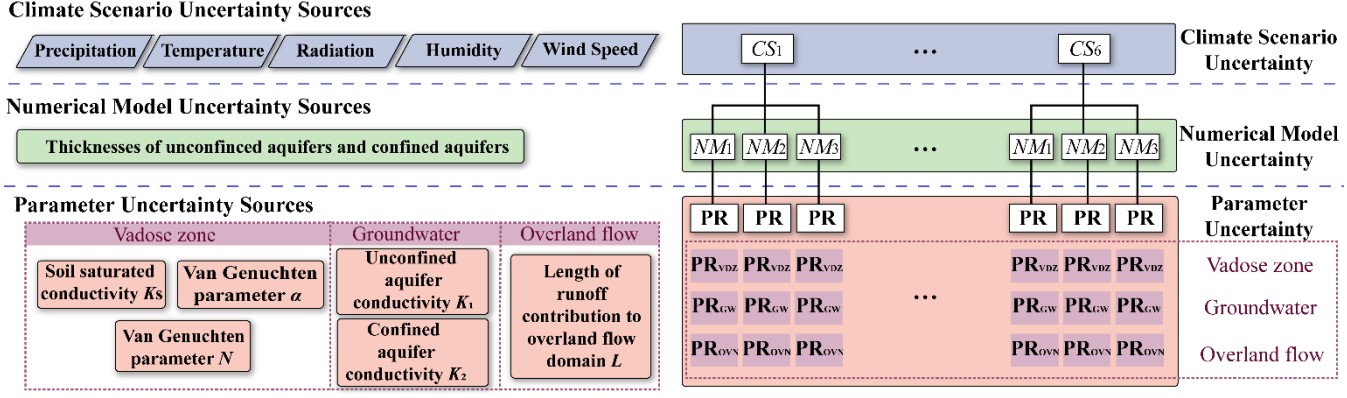

**Figure 2**

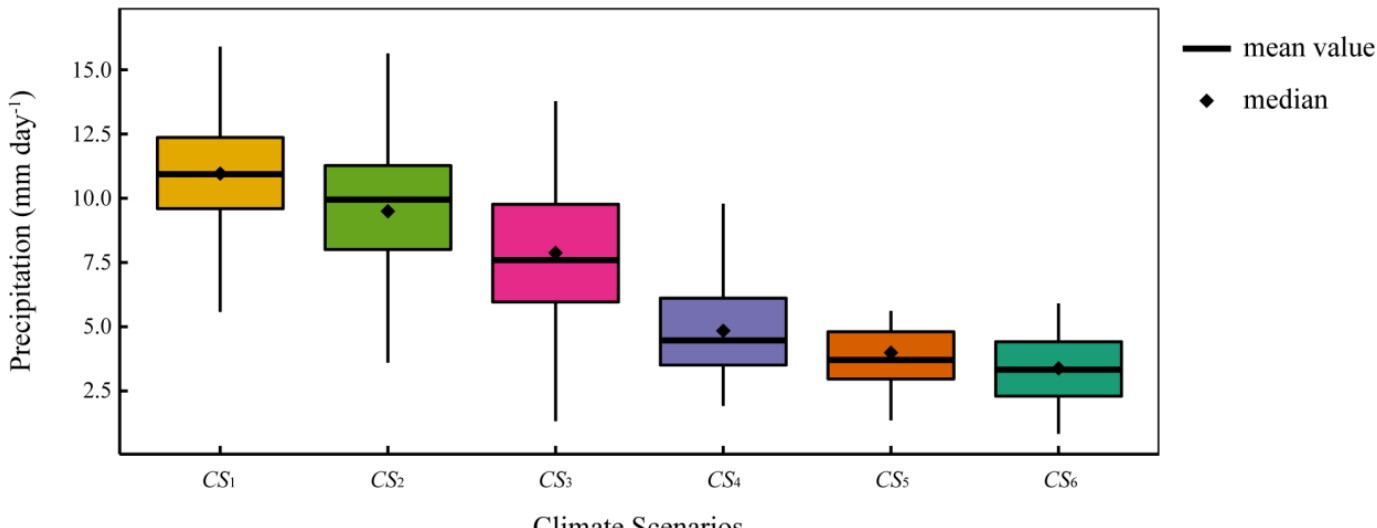

**Figure 3**

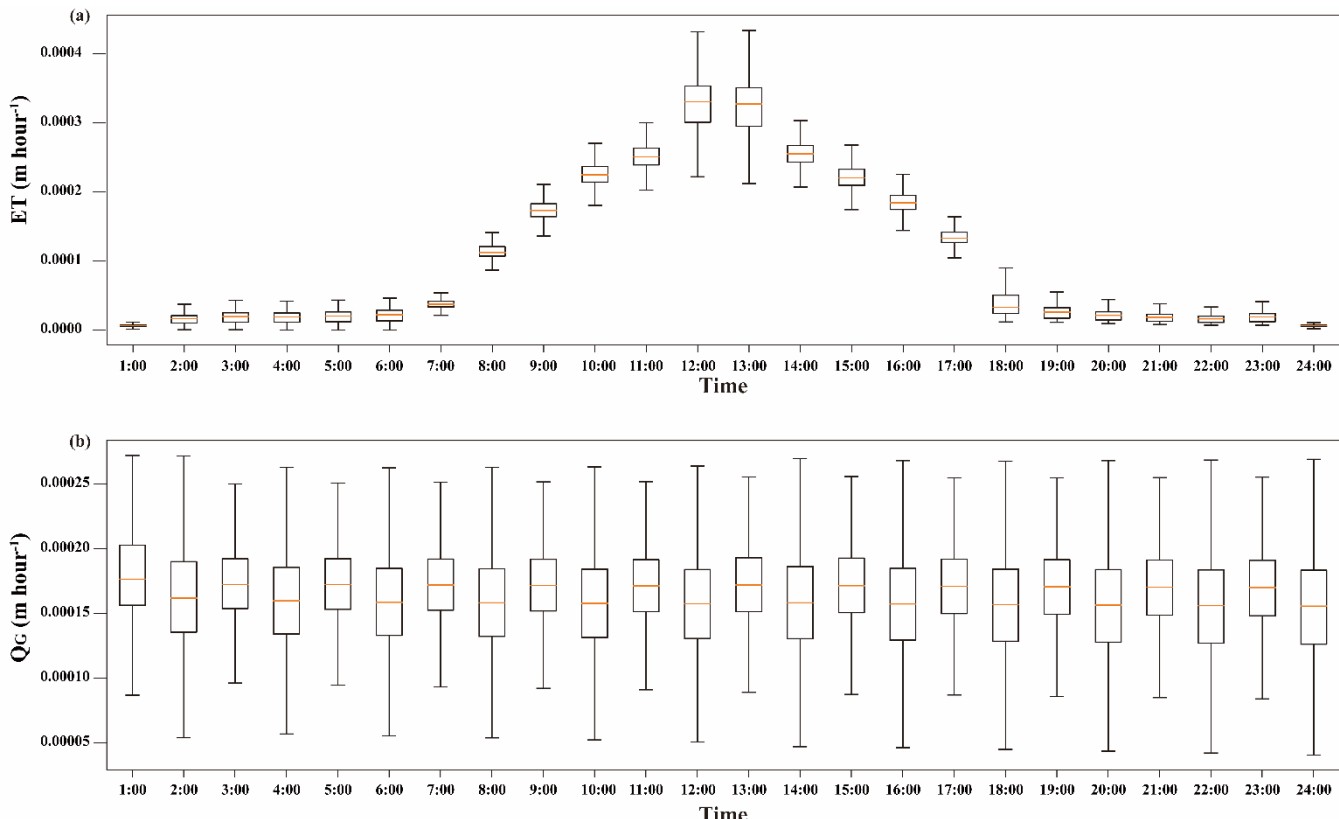

**Figure 4**

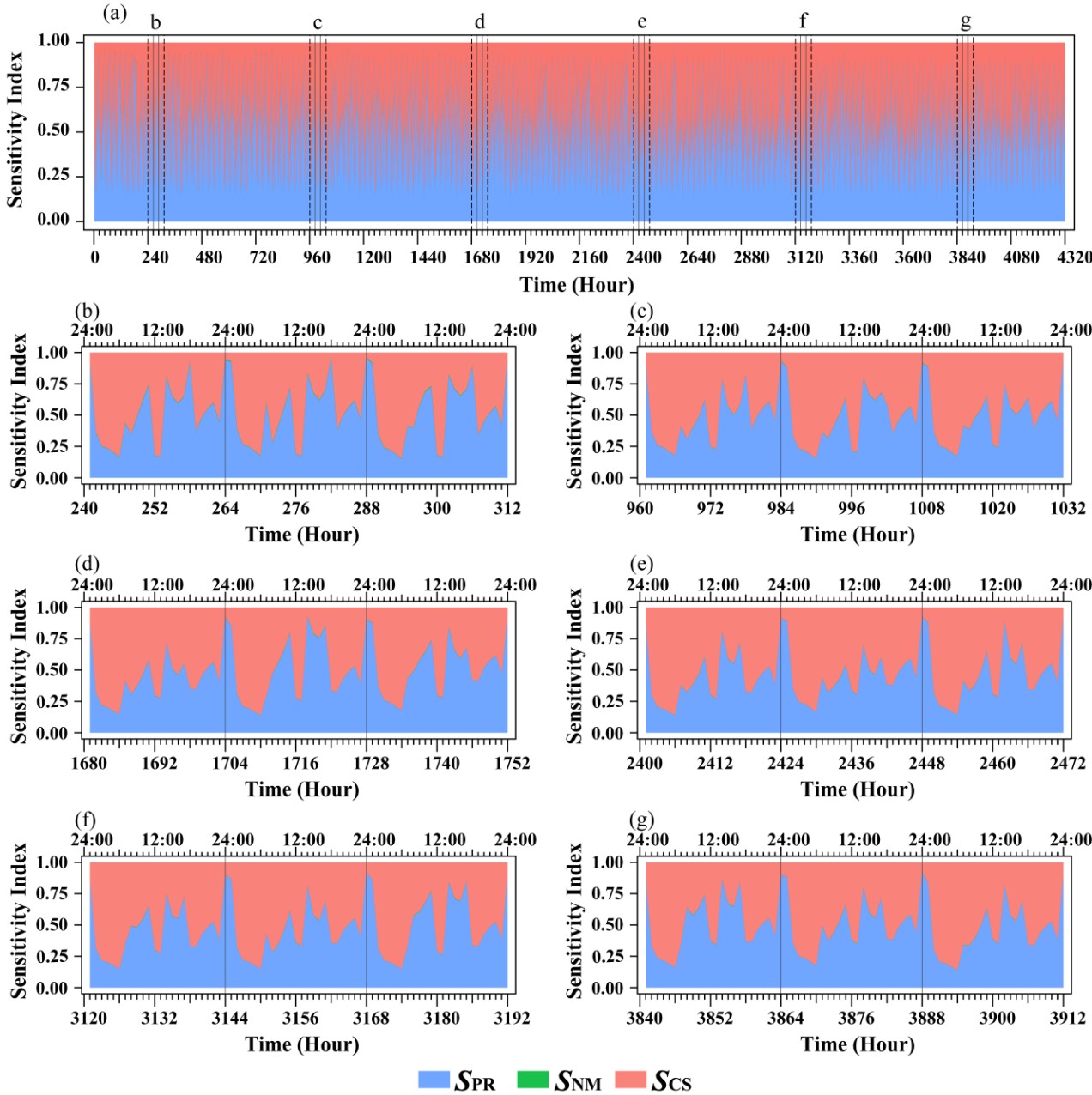

**Figure 5**

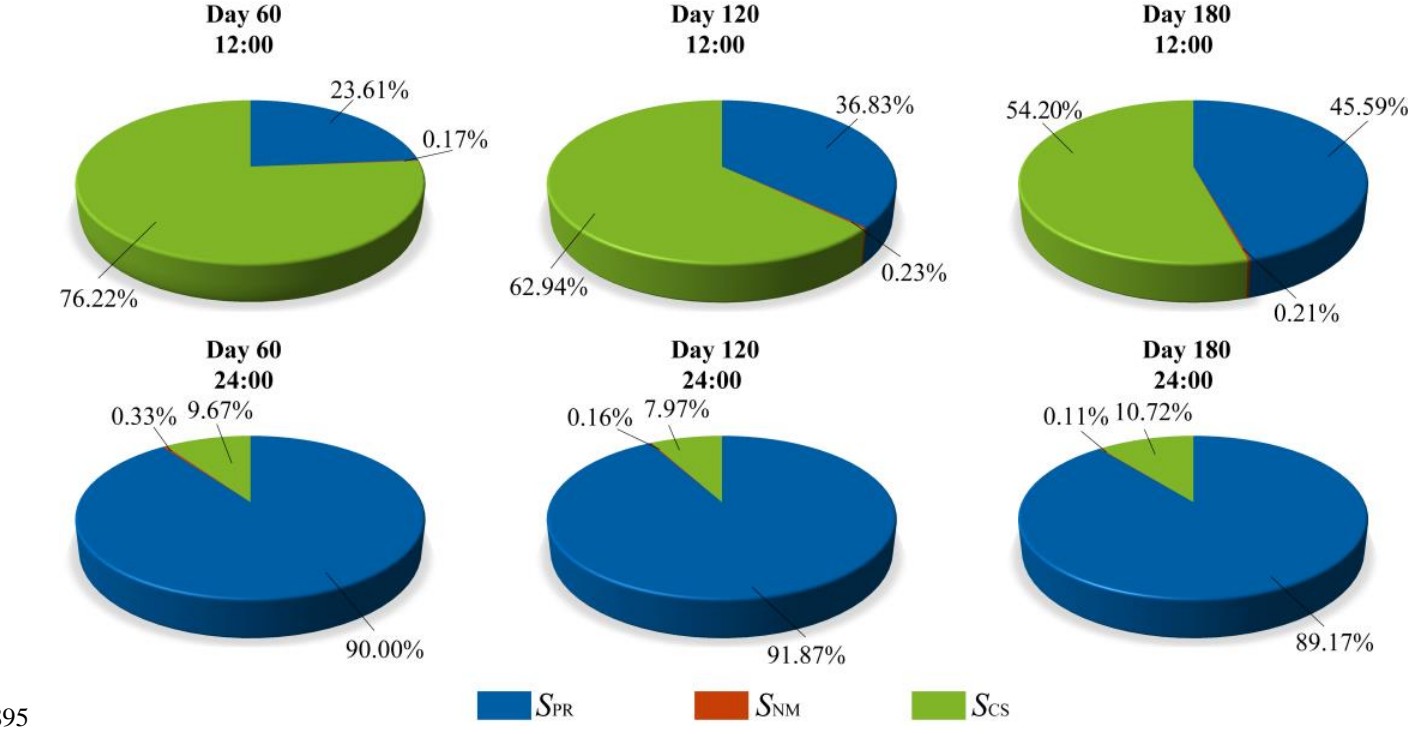

**Figure 6**

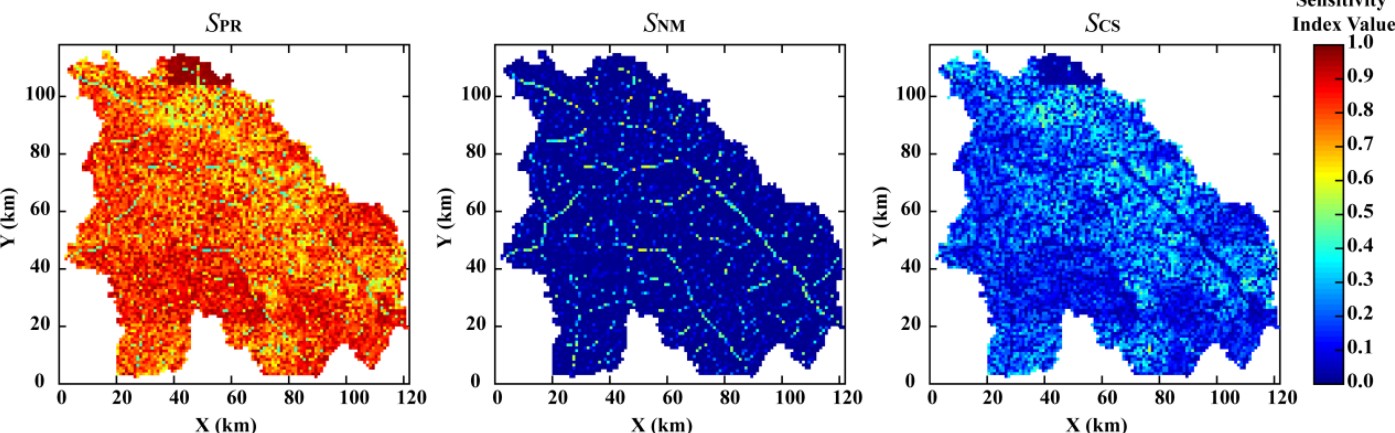

**Figure 7**

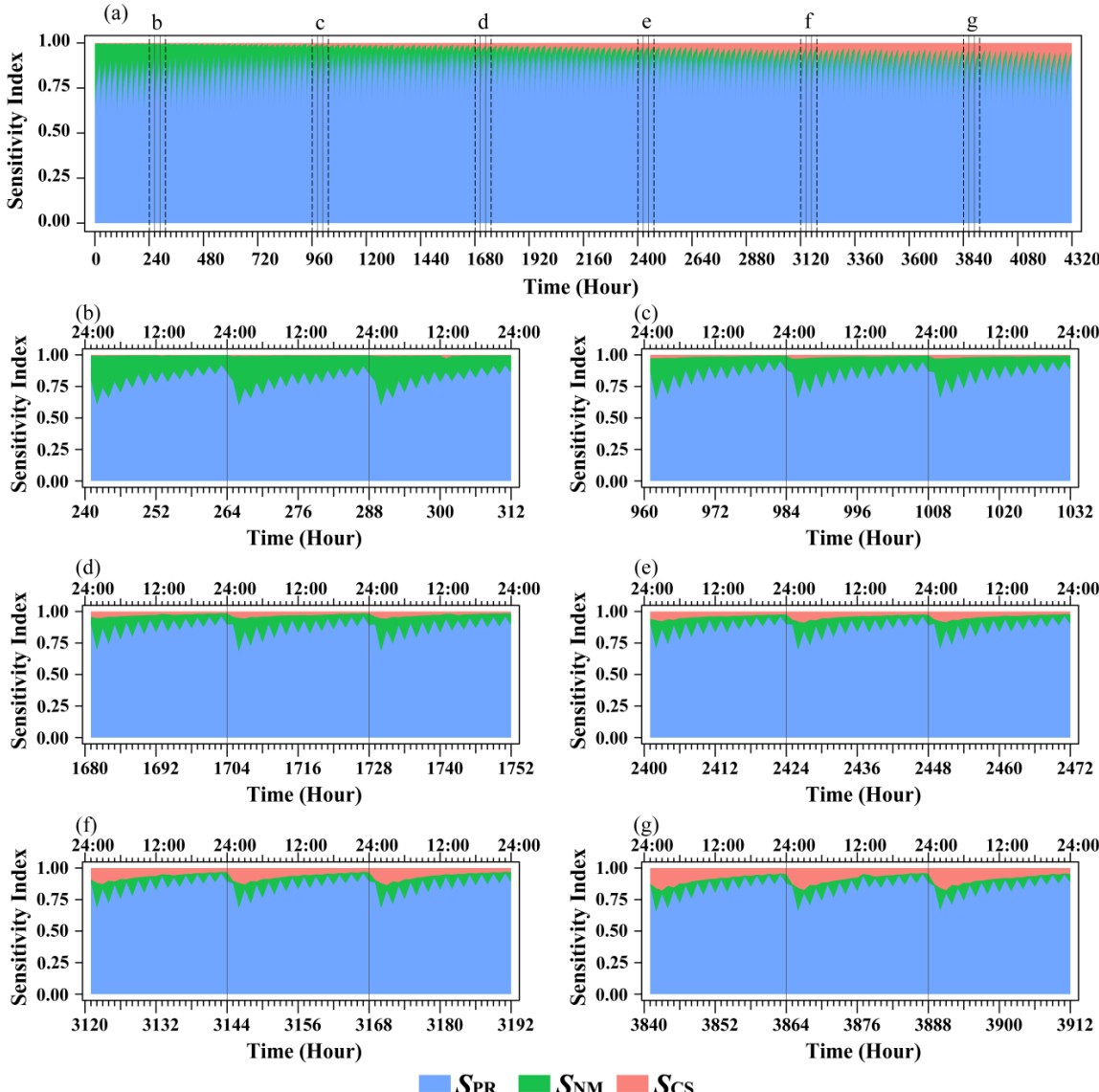

**Figure 8**

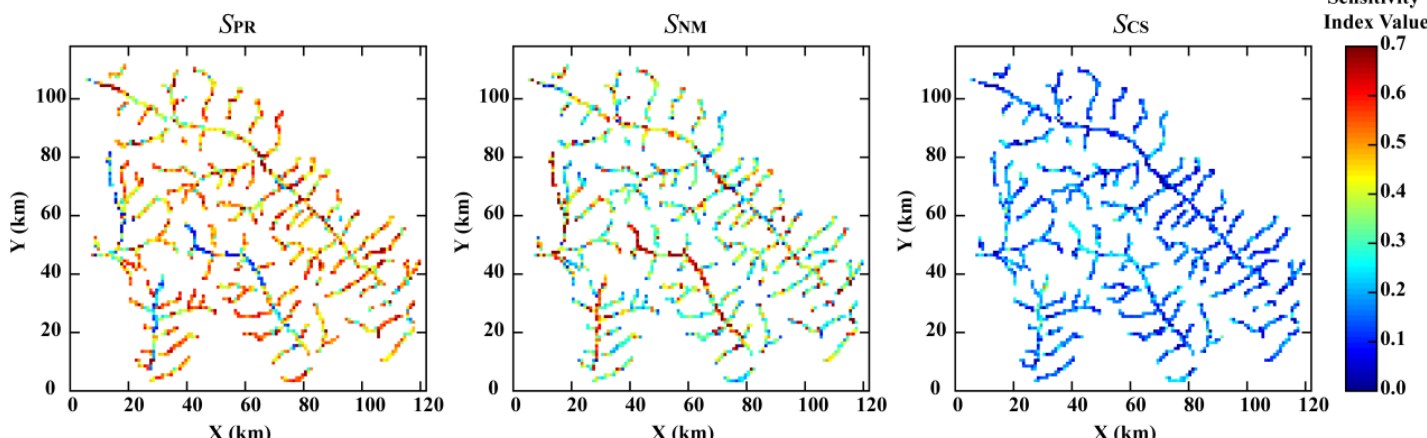

**Figure 9**

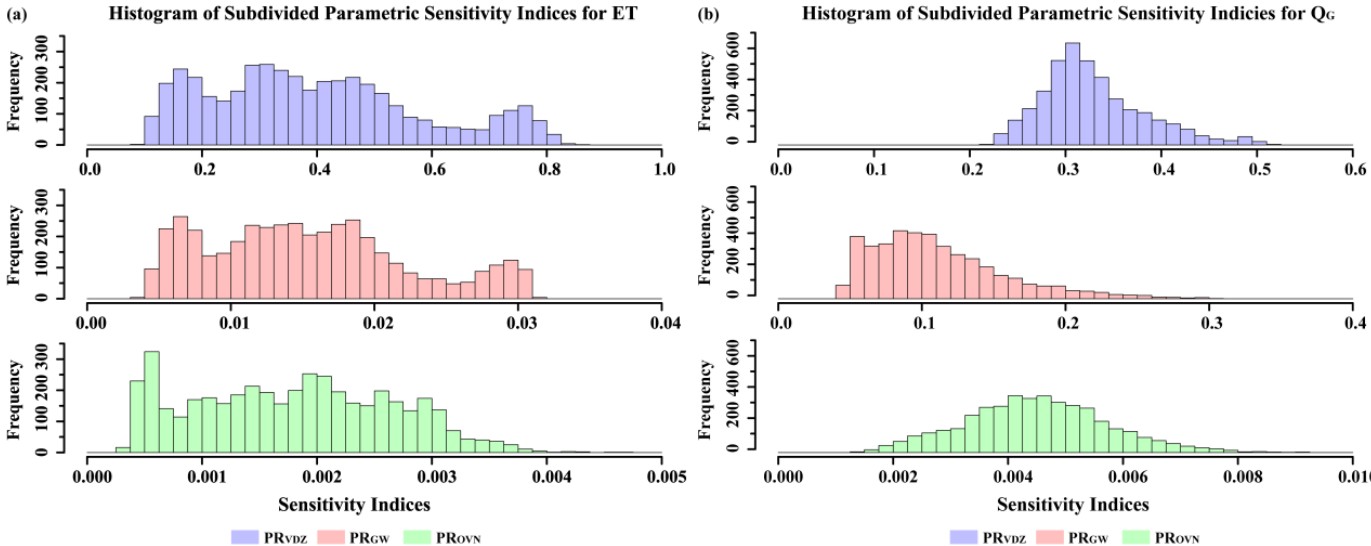

**Figure 10**

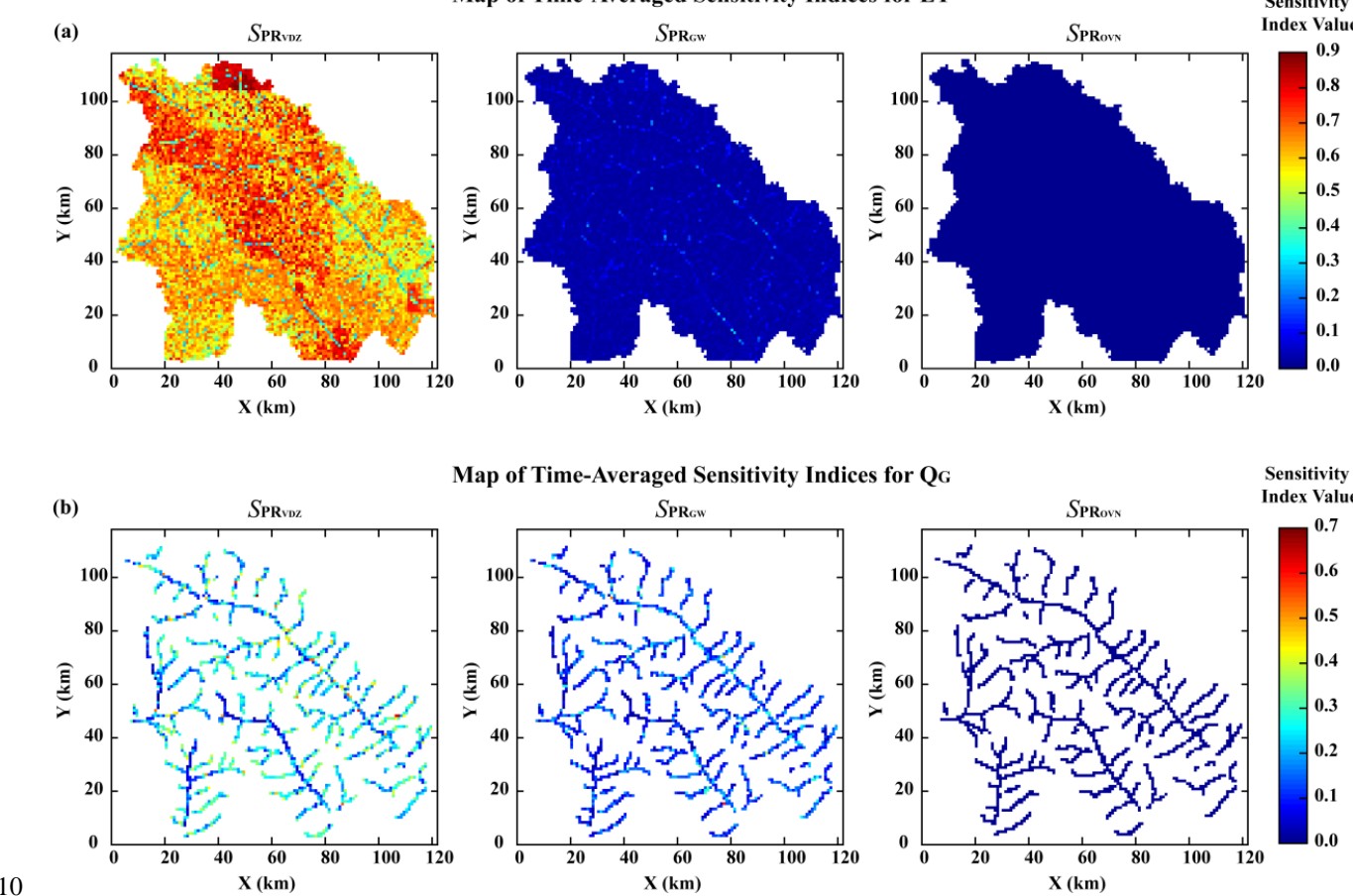

**Figure 11**

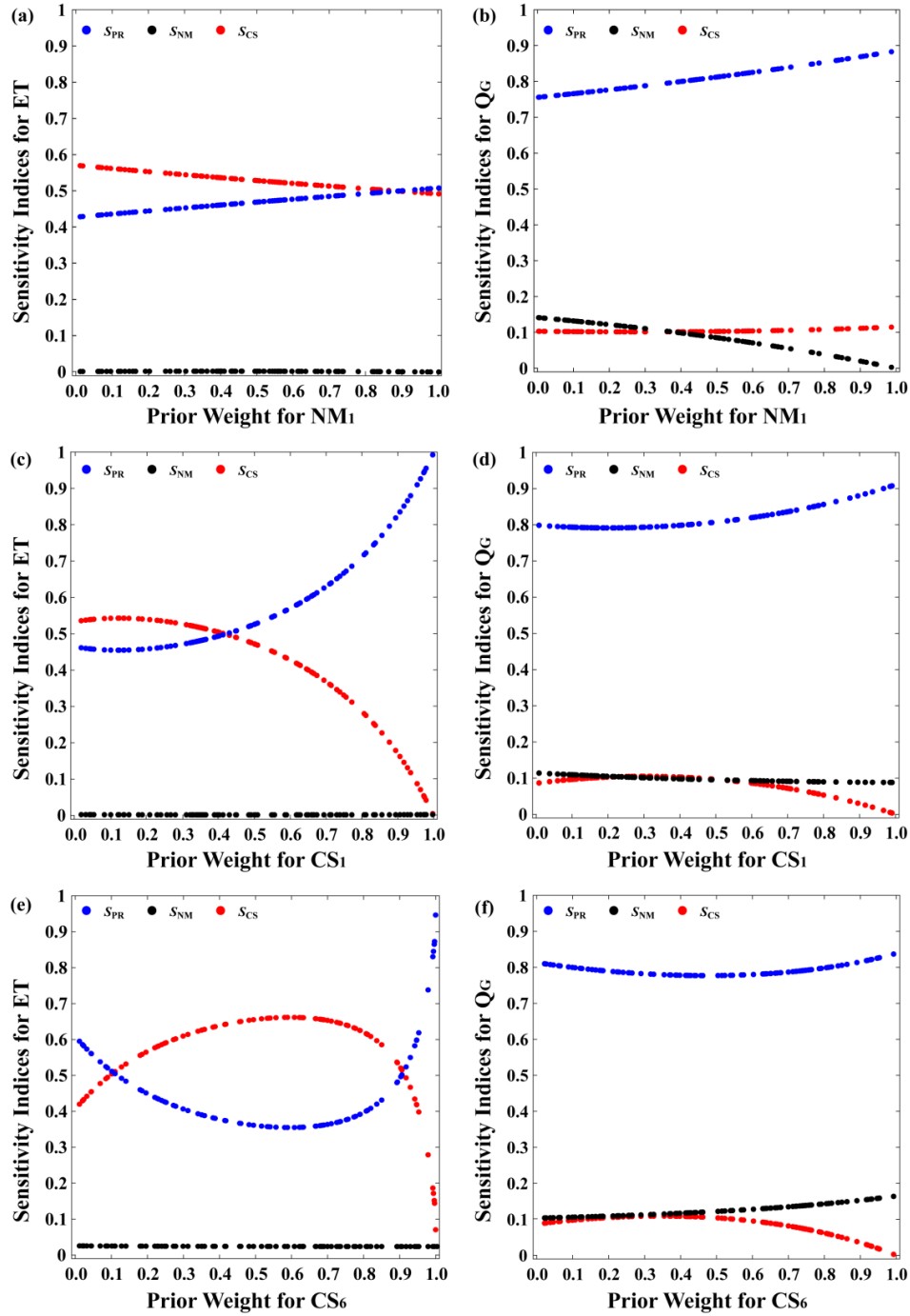

**Figure 12**

**Map of Six Parameters Time-Averaged Sensitivity Indices for Q$_G$**

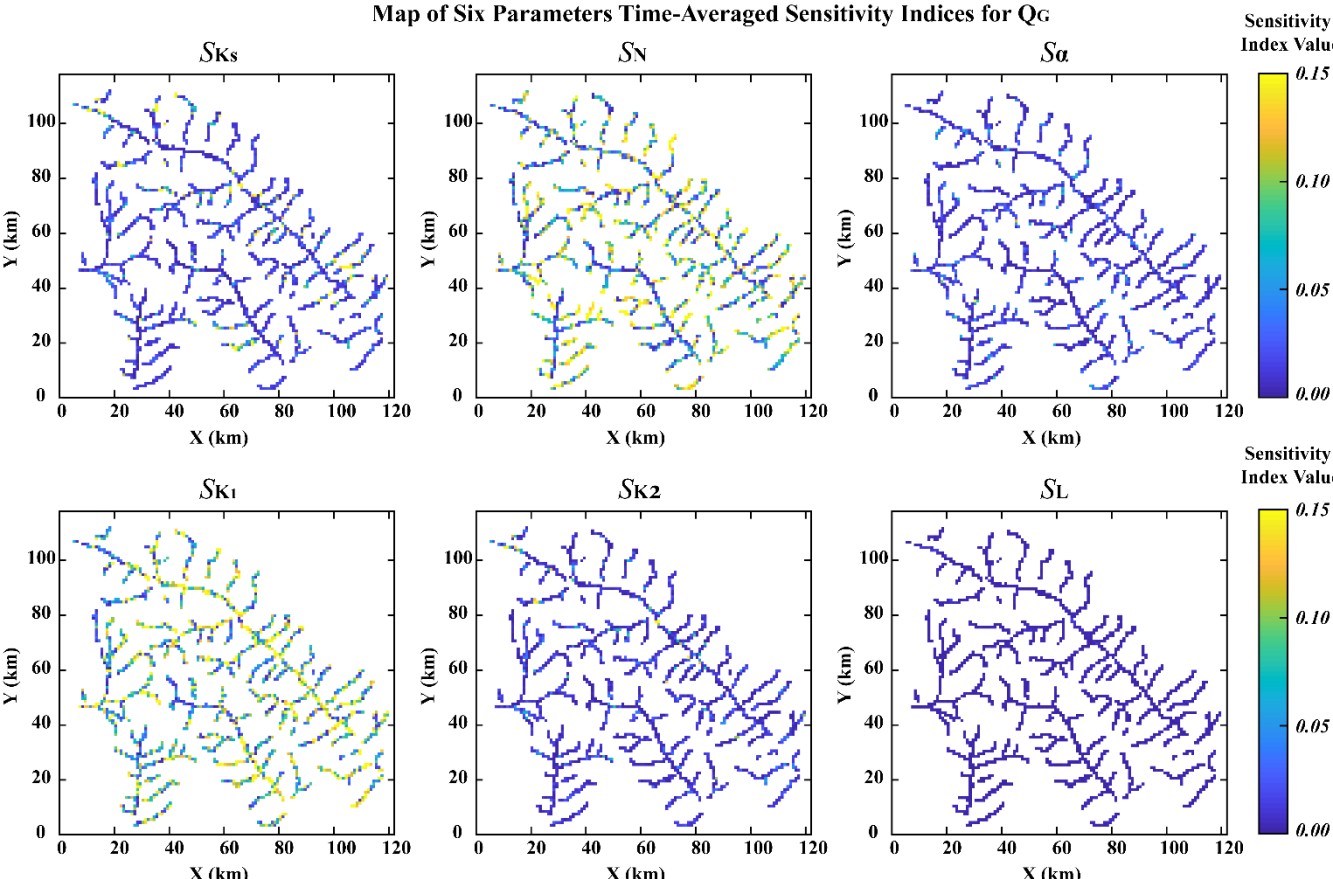