# Peer review of "Hierarchical <mark>s</mark>ensitivity <mark>a</mark>nalysis for <mark>a l</mark>arge-scale <mark>p</mark>rocess-based hydrological model applied to an Amazonian watershed"

_Hydrology and Earth System Sciences, 2020_

## Referee Comment (RC1) · Anonymous Referee #1 · 5 Apr 2020

General comments: 1. The authors did a lot of work, but I think the writing needs a great improvement to highlight their work. The current writing reads more like a report rather than a scientific paper. The scientific motivation is not very clear to me. 2. The authors said the aim of this work is to provide a pilot example of comprehensive global sensitivity analysis for large-scale PBHMs, then what lessons can the audience learn from this pilot example? Please provide a detailed discussion. 3. In the introduction, please highlight the objective, contribution and novelty of this work, and justify its significance. 4. I think the authors need pay more attention to the writing. The logic is not very clear and sometimes the conclusive sentences pop out without justification. Specific comments: 1. The grammar of the title is not right. 2. I think the abstract

needs to be rewritten. Right now, it reads like a report instead of a scientific paper. I did not see a scientific motivation but a description of what the authors did. 3. I think the logic of the introduction needs an improvement. 4. Line 47-50, I found this last sentence is confusing. SA "becomes" important? Limited resources? 5. Line 59, this sentence is confusing. "Using" large-scale PBHMs? Why the computational cost is high? 6. Please justify why the authors chose the PAWS model as the pilot example. 7. It seems that the authors have some methodology development on the basis of their previous work. Please highlight these contributions and novelty, and justify that this new development is necessary for the complex and large-scale model sensitivity analysis. 8. Line 221, why 600 samples? 9. Line 223-224, in what sense the LHS greatly reduces the required sample size compared to MC sampling? To achieve the same estimation accuracy? Please provide evidence or reference. 10. Line 265, when does the binning method not work? Please comment. 11. Line 410, how do the authors justify the results accuracy? 12. Line 417-419, the sentence is confusion. 13. Line 419-420, the sentence comes out of nowhere. 14. Line 430, "all" is a strong work, be careful of using it. 15. Line 434, the method can largely reduce the computational cost associated with complex, large-scale hydrological models. From which aspects to reduce the cost? Does it reduce the forward simulation time? Please be specific here. 16. The words in Figure 2 are hard to read.

---

## Referee Comment (RC2) · Anonymous Referee #2 · 14 Apr 2020

General Comments: Sensitivity analysis is an effective tool for identifying important uncertainty sources and improving model calibration and predictions. This study used an advanced hierarchical global sensitivity analysis framework to investgate the uncertainty sources of a three-dimensional, process-based hydrologic model in Amazon catchment. Three uncertainty sources are considered including model parameters, model structure and climate scenarios.

I think this research topic is meaningful in hydrology community, especially for a large scale catchment study. This paper is well organized and easy to read. The conclusions are well supported by the results and data. However, some important problems

are not clear, which should be addressed before publication. Please see the specific comments.

Line 55: Please give a brief description to other global sensitivity analysis methods, e.g., the sensitivity analysis based on information entropy.

Line 290: Are the three aquifer models sufficient to investigate the sensitivity of the model output to aquifer thickness? Please justify it.

Line 293: Are these model weights used for model averaging or model combination?

Line 297: Parameters' ranges have influence on the results of sensitivity analysis, please explain the allowable ranges of these parameters in Table 2.

Line 346: It is good to investgate the contribution of groundwater system to stream-flow, and this research find that the thickness of an aquifer will greatly influence the water redistribution process in the aquifer. However, I want to see the influences of aquifers' thicknesses on streamflow in more detail, such as, the unconfined aquifer and the confined aquifer, It is expected that the unconfined aquifer has more influence on streamflow, because it has stronger interaction with surface flow.

Line 383: what's the meaning of prior weights here, will they used for Bayesian model averaging?

---

## Referee Comment (RC3) · Anonymous Referee #3 · 3 May 2020

This study offers a fine-grained analysis of sensitivities of simulated evapotranspiration and groundwater contribution to climate, model parameters and subsurface stratigraphy (what the authors called model uncertainty). A lot of work is involved here and rich results have been presented. I think overall the results are interesting and offer a great deal of insight about how these models function, which also are hypotheses about how the natural systems function (note: hypotheses not conclusions). These sensitivities could guide future research and model development, e.g., previous studies have rarely studied the impacts of soil thickness. The authors presented analysis of sensitivities at hourly level, at spatially-distributed gridpoint scale, and river reach scales, which I have not seen elsewhere. This rich set of analysis gave me some

useful things to think about, e.g., how climate has delayed sensitivity on the groundwater, and how vertical and horizontal parameters exert different controls baseflow. I think this paper will be a valuable contribution to HESS. However I would like to point out a few issues that should be addressed: Major issues1. The overall motivation should be improved. It should not be "there lacked of researchÂăutlilizingÂăquantitative and representative global sensitlvityÂăanalysis" (line 59-61). I mean yes this is a gap but the primary objective should be to understand uncertainty sources and provide insights to physical processes that control ET and baseflow. 2. Section 2.2 should be greatly shortened, or moved to Supporting Information. It's way too long right now.3. Conversely, some figures deserve more discussion, e.g., Figure 7 – I do not think it is the thickness right under the river cells, it's about overall thickness and how much water from the watershed concentrates to the channels. This hypothesis can be tested;Âă Figure 9 – headwater vs stem river cells. Figure 11 – not really discussed much.3. How the authors came up with the six climate scenarios are not described. How could you have these different scenarios? 4. Figure 4 is a big mess. A cleaner representation such as a boxplot is required.Âă 5. The authors need to tone down the description of the hourly sensitivity especially around night. There may be many assumptions baked into how daily precipitation is disaggregated into hourly which influenced these results. I doubt how robust this is. 6. Although not required, it will be nice to demonstrate the results for a year rather than 180 days. The annual cycle tells us more things. 7. the soil thickness should not be called "numerical model" uncertainty, but "subsurface stratetigraphy".

Some minor points:line 82, Michigan state –> Michigan or the state of MichiganA relevant citation for this paragraph: Ji et al., 2019, 10.1029/2018WR023897line 126. the model tool –> the modeling toolline 122, drainage network was formed –> formedline 287-292 paragraph— Brunke et al.Âă 2016 is relevant to discuss 10.1175/JCLI-D-15-0307.1line 307– on what machine did you run these many simulations and how much time did it take?line 323 –> has little influence on spatially-averaged ET. (I believe for different cells it has a more prominent impact)line 326 –> temporally dependent –>

time-dependentline 331 "greatly decreasing"?? awkward phrasing.line 356 – accumulation over time? Maybe also due to seasonality? we don't know for sureline 366, remove "through this investigation".line 354 – river flow always occurs hours later than the rainfall process — what if the rainfall isn't large enough to trigger a response?line 347-348– circular logic and tautology

———————————————

---

## Author Comment (AC1) · 2 Jun 2020

We appreciate the reviewer for his/her valuable comments and feedback on our study. We believe these comments will greatly improve this research.

General Evaluation 1

The authors did a lot of work, but I think the writing needs a great improvement to highlight their work. The current writing reads more like a report rather than a scientific paper. The scientific motivation is not very clear to me.

Response

[Figure]

We appreciate the reviewer's evaluation of this manuscript and constructive comments. We will totally revise this paper, especially the abstract, introduction, discussions and conclusions sections. The contents of these sections will be reorganized and partially rewritten in order to highlight and emphasize our research motivation and goals: to develop a new tool and demonstrate its implementation to a pilot example for comprehensive global sensitivity analysis of large-scale hydrological modelling.

General Evaluation 2

The authors said the aim of this work is to provide a pilot example of comprehensive global sensitivity analysis for large-scale PBHMs, then what lessons can the audience learn from this pilot example? Please provide a detailed discussion.

Response

We will add more discussions about our findings for this comprehensive global sensitivity analysis in the revised manuscript. In general, we would like to discuss three key insights we learnt from this study results. First, it is necessary to implement such a comprehensive global sensitivity analysis method which considers more than parametric uncertainty for the large-scale PBHMs since the sensitivity analysis results showed that other sources of uncertainty (e.g., climate scenario and model uncertainties) are essential as well for model predictions. Second, using the new improved hierarchical sensitivity analysis method with the capability of flexibly combining different uncertain factors together is a computational affordable and useful way to identify the most important physical process for the large-scale complex PBHMs. By categorizing and combining these uncertainty sources into different processes and placing them in a proper layer of a hierarchical uncertainty framework, this advanced hierarchical sensitivity analysis method can largely reduce the initially unaffordable computational cost of global sensitivity analysis of PBHMs and useful sensitivity indices can be provided to measure the importance of different model uncertain inputs in the physical process viewpoint. And we also discovered certain patterns for the sensitivity analysis results

of this study case. Although these patterns may not be universally correct, they still can provide useful insights for other modelers with similar cases and models. We will add more contents related to these points in the discussions and conclusions sections.

General Evaluation 3

In the introduction, please highlight the objective, contribution and novelty of this work, and justify its significance.

Response

We will totally rewrite the introduction section with different logical flow and new content. After modification, the purpose, contribution and novelty of this research will be more highlighted. We will point out the main purpose and contribution of this work is to develop and demonstrate a new comprehensive global sensitivity analysis method for large-scale complex hydrological models with considering various types of uncertainty sources and physical processes. The novelty of our work includes the implementation of comprehensive global sensitivity analysis for the large-scale complex PBHMs using affordable computational cost. And the new flexible hierarchical uncertainty framework and sensitivity index system also provide modelers novel capability of analysing sensitivity in the physical process viewpoint and estimating accurate importance for further subdivided parametric uncertainty.

General Evaluation 4

I think the authors need pay more attention to the writing. The logic is not very clear and sometimes the conclusive sentences pop out without justification.

Response

To improve the presentation of this study, we will revise the manuscript substantially to make sure that the logic flows smoothly and also avoid making any conclusions without any justification. The abstract, introduction, and conclusions sections will be totally rewritten. And we will hire professional editor for revising the grammar.

Comment 1

The grammar of the title is not right.

Response

We will revise the title following professional English language editor's suggestions.

Comment 2

I think the abstract needs to be rewritten. Right now, it reads like a report instead of a scientific paper. I did not see a scientific motivation but a description of what the authors did.

Response

We will rewrite the abstract. The basic scientific motivation of this work will be highlighted as to develop a new computational affordable tool for comprehensive global sensitivity analysis of large-scale complex PBHMs with considering various uncertainty sources and physical processes.

Comment 3

I think the logic of the introduction needs an improvement.

Response

We will rewrite the introduction and the basic logic of this part has been rearranged as: 1. Introduce the background of the PBHM model and the importance of sensitivity analysis of the PBHM model. 2. Present an overview of traditional sensitivity analysis methods. 3. Present the challenges encountered in the analysis of PBHMs using traditional sensitivity analysis methods. 4. State the purpose and contribution of this research with highlighting the novelty of our method. 5. Provide an overview of the structure of this paper.

Comment 4
Line 47-50, I found this last sentence is confusing. SA "becomes" important? Limited resources?

Response

We understand the reviewer's confusion. We will revise this sentence and add more references. The sentence will be revised as: "Uncertainty is inevitable and important in numerical modelling (Saltelli and Sobol, 1995; Neuman, 2003; Saltelli et al., 2000, 2010; Neuman, 2003; Lu et al., 2012; Song et al., 2015; Razavi and Gupta, 2015, 2016; Neuman, 2003; Rojas et al., 2010) , especially for the highly complex PBHMs considering various physical processes. Therefor the sensitivity analysis which can identify the most influential sources of uncertainty is a useful tool for both modelers and managers. The resources can be preferentially used to reduce the most important uncertainties (e.g., obtaining new data for more accurate boundary conditions) and thus most efficiently improve the model calibration and prediction processes.

Comment 5

Line 59, this sentence is confusing. "Using" large-scale PBHMs? Why the computational cost is high?

Response

We will rewrite this sentence as "there is still a lack of research of quantitative and comprehensive global sensitivity analysis for large-scale PBHMs". There are generally two main reasons for the high computational cost of global sensitivity analysis of a large-scale PBHM: the high complexity of model itself and the high standard of global sensitivity analysis. A PBHM usually has a very large number of parameters and multiple high-order nonlinear governing equations. These facts combining with a large-scale model domain cause the running of a PBHM itself is already very computationally expensive. For the sensitivity analysis, comparing with the local sensitivity analysis which can only provide results valid in certain range of parameter values (e.g., the derivative

of model prediction with respect to parameter A at certain value point can be a measurement of A's local sensitivity at this point), the global sensitivity analysis is more comprehensive because its results are valid for the whole range of parameter value. To achieve this goal, the methods of global sensitivity analysis are all relatively computationally expensive (e.g., the Morris method and variance-based method both use complex sampling techniques) and their computational cost grows exponentially with number of parameters. Therefore, the implementation of global sensitivity analysis for a PBHM leads to extremely high computational cost considering we have to run a large number of simulations for a complex PBHM using different parameter samples. We will state these clearly in the revised manuscript.

Comment 6

Please justify why the authors chose the PAWS model as the pilot example.

Response

The main reason for choosing PAWS as the pilot example of PBHMs is that, compared with other PBHMs, PAWS is a comprehensive and representative large-scale hydrological model which can be applied to large catchments and long-time frames by efficiently coupling both surface and subsurface hydrological processes (Shen and Phanikumar, 2010). And the complexity and parameter dimensionality of PAWS are high enough to test and demonstrate our new global sensitivity analysis method. Furthermore, PAWS has been applied to the studying watershed previously and it was capable of simulating multiple key variables of hydrological states and fluxes at different spatiotemporal scales and presented good model performance comparing to various ground and satellite observation data (Niu et al., 2017). This previous model application provides solid base for our uncertainty identification and sensitivity analysis study. We will add these reasons for choosing PAWS in the introduction section.

Comment 7

It seems that the authors have some methodology development on the basis of their previous work. Please highlight these contributions and novelty, and justify that this new development is necessary for the complex and large-scale model sensitivity analysis.

Response

We do have a new methodology development based on the previous work: the further decomposition and quantification for the subdivided parametric uncertainties. This new technique defines a new set of sensitivity indices and allows modelers to analysing the importance of a physical process only involves partial model parameters. The implement of this new method is necessary because the main weakness of our previous methodology is it considers all parameters together as one single process. This simple strategy may be adequate for a groundwater modelling case, but it cannot provide detailed information for a PBHM which includes multiple hydrological processes. We will revise the introduction to highlight this contribution and novelty, then justify the necessity of this new development.

Comment 8

Line 221, why 600 samples?

Response

We understand the confusion of reviewer. First, based on the experiences of previous research cases (Emery et al., 2016; Dai et al., 2019), we believe 600 is an adequate parameter sample size for our test case considering the model domain and number of uncertain parameters. Second, because of the high computational cost, 600 parameter samples are the maximum size we can afford. After combining model uncertainty and climate scenario uncertainty, we need to conduct a total of $600 \times 3 \times 6 = 10,800$ simulations for the PBHM. The pure simulation time without analysing data is already very time consuming even using the best high-performance computing (HPC) platform we have.

Comment 9

Line 223-224, in what sense the LHS greatly reduces the required sample size compared to MC sampling? To achieve the same estimation accuracy? Please provide evidence or reference.

Response

The LHS has been one popular sampling technique to reduce computational cost and it has been proven to be more effective than conventional MC sampling method. We will provide more references to explain and exhibit the effectiveness of LHS method.

Comment 10

Line 265, when does the binning method not work? Please comment.

Response

The binning method is a rigorously derived mathematical technique designed to separating and estimating the partial variances contributed from different parameters of one LHS method sampled parameter set. Because the mathematical equations are general and rigorous, this method can be applied to any modelling case with LHS parameter samplings. However, when the samplings for different parameters are totally random and unrelated like conventional Monte Carlo simulation, the binning method is not applicable. We will add these comments in the revised manuscript.

Comment 11

Line 410, how do the authors justify the results accuracy?

Response

We intended to express that this hierarchical sensitivity analysis method is capable of estimating accurate sensitivity measurements (sensitivity index) for the importance of certain uncertainty sources. Comparing with other sensitivity analysis methods which

only provide qualitative results (e.g., ranks of importance), this variance-based quantitative global sensitivity analysis method is more accurate. It is more appropriate to replace the word "accurate" with "quantitative". We will revise this sentence to "This study has shown that the improved hierarchical sensitivity analysis method is capable of providing quantitative and comprehensive assessment of the importance of uncertain inputs through variance decomposition..."

Comment 12

Line 417-419, the sentence is confusion.

Response

We will revise this sentence and collaborate with the introduction section to explain the importance of further decomposing of subdivided parametric uncertainties.

Comment 13

Line 419-420, the sentence comes out of nowhere.

Response

We will remove this sentence.

Comment 14

Line 430, "all" is a strong work, be careful of using it.

Response

We will replace "all" with "three types of common", which is more accurate.

Comment 15

Line 434, the method can largely reduce the computational cost associated with complex, large-scale hydrological models. From which aspects to reduce the cost? Does it reduce the forward simulation time? Please be specific here.

Response

We understand the confusion caused by this sentence. What we want to express here is that the improved hierarchical sensitivity analysis method largely reduces the computational cost of global sensitivity analysis for PBHMs, not the computational cost of PBHM simulation itself. In order to express more accurately, we will modify the sentence in the original text as: "this advanced hierarchical sensitivity analysis method can largely reduce the computational cost for sensitivity analysis associated with complex, large-scale hydrological models by categorizing multiple uncertainties into processes and placing them into a proper layer in a hierarchical framework" in the revised manuscript.

Comment 16

The words in Figure 2 are hard to read.

Response

We will update this figure using a higher resolution and a larger font.

References Dai, H., Chen, X., Ye, M., Song, X., Hammond, G., Hu, B., and Zachara, J. M.: Using Bayesian Networks for Sensitivity Analysis of Complex Biogeochemical Models, Water Resour Res, 55, 3541-3555, 2019. Emery, C. M., Biancamaria, S., Boone, A., Garambois, P. A., Ricci, S., Rochoux, M. C., and Decharme, B.: Temporal Variance-Based Sensitivity Analysis of the River-Routing Component of the Large-Scale Hydrological Model ISBA-TRIP: Application on the Amazon Basin, J Hydrometeorol, 17, 3007-3027, 2016. Niu, J., Shen, C. P., Chambers, J. Q., Melack, J. M., and Riley, W. J.: Interannual Variation in Hydrologic Budgets in an Amazonian Watershed with a Coupled Subsurface-Land Surface Process Model, J Hydrometeorol, 18, 2597-2617, 10.1175/Jhm-D-17-0108.1, 2017. Shen, C. P., and Phanikumar, M. S.: A process-based, distributed hydrologic model based on a large-scale method for surface-subsurface coupling, Adv Water Resour, 33, 1524-1541,

10.1016/j.advwatres.2010.09.002, 2010.

Please also note the supplement to this comment:
https://www.hydrol-earth-syst-sci-discuss.net/hess-2020-87/hess-2020-87-AC1-supplement.pdf

---

## Author Comment (AC2) · 2 Jun 2020

We greatly appreciate the reviewer for his/her valuable comments and feedback on our study. We believe these comments will greatly improve this manuscript.

General Evaluation

Sensitivity analysis is an effective tool for identifying important uncertainty sources and improving model calibration and predictions. This study used an advanced hierarchical global sensitivity analysis framework to investigate the uncertainty sources of a three-dimensional, process-based hydrologic model in Amazon catchment. Three

uncertainty sources are considered including model parameters, model structure and climate scenarios. I think this research topic is meaningful in hydrology community, especially for a large scale catchment study. This paper is well organized and easy to read. The conclusions are well supported by the results and data. However, some important problems are not clear, which should be addressed before publication. Please see the specific comments.

Response

We thank the reviewer for the positive evaluation and constructive comments of the manuscript. We will substantially revise the manuscript following reviewer's comments.

Comment 1

Line 55: Please give a brief description to other global sensitivity analysis methods, e.g., the sensitivity analysis based on information entropy.

Response

We will add more descriptions and references to other global sensitivity analysis methods in the introduction, for example, screening method, regression-based method, variance-based method, meta-model method, and information-entropy-based method.

Comment 2

Line 290: Are the three aquifer models sufficient to investigate the sensitivity of the model output to aquifer thickness? Please justify it.

Response

We chose three aquifer models here because (1) the sediments throughout the Amazon Basin are relatively old, so the stratification of soil and aquifer is relatively stable and the thickness does not change much (Do Rosario et. al., 2016); (2) the lack of actual measurements prevents us to determine the boundary between unconfined and confined aquifers (Fan et. al., 2010); (3) these three models represent three typical

cases that (i) the unconfined aquifer is very thick, (ii) the thickness of the unconfined aquifer is similar to that of the confined aquifer, and (iii) the confined aquifer is very thick. Based on the above three reasons, we chose these three aquifer situations to investigate the sensitivity of the model output to aquifer thickness. We will add more detailed discussion in the revised manuscript.

Comment 3

Line 293: Are these model weights used for model averaging or model combination?

Response

We are indeed using these weights for the model averaging process involved in the variance calculations of sensitivity analysis. The hierarchical sensitivity analysis method requires the weight of model NMk under scenario CSl satisfying , and the weight of scenarios satisfying . In this study, we assume that different climate scenarios have the same weight. We also assume that the weights of the different models under each climate scenario are the same. In other words, for 6 climate scenarios, the weight of each climate scenario is 1/6; for each of the 3 models under a certain climate scenario, the weight of each model is 1/3. The results from Section 3.1 to Section 3.4 are exhibited under these assumptions. However, we changed the weights for NM1, CS1, and CS6 within the range of (0, 1) in Section 3.5, respectively. If the weight for NM1, CS1, or CS6 is p, then the weight of the remaining uncertain factors of the same kind will be assumed as (1-p)/n, where n is the number of the remaining same-kind factors.

Comment 4

Line 297: Parameters' ranges have influence on the results of sensitivity analysis, please explain the allowable ranges of these parameters in Table 2.

Response

We will add explanations to allowable ranges of these parameters in the revised manuscript.

Comment 5

Line 346: It is good to investigate the contribution of groundwater system to streamflow, and this research find that the thickness of an aquifer will greatly influence the water redistribution process in the aquifer. However, I want to see the influences of aquifers' thicknesses on streamflow in more detail, such as, the unconfined aquifer and the confined aquifer, It is expected that the unconfined aquifer has more influence on streamflow, because it has stronger interaction with surface flow.

Response

We thank the reviewer for the suggestion. And we think the comparison for the importance of unconfined and confined aquifers is indeed worth to investigate. We will add new results for defining and estimating new sensitivity indices of these two different types of aquifers.

Comment 6

Line 383: what's the meaning of prior weights here, will they used for Bayesian model averaging?

Response

We understand the confusion of the reviewer. We called this weight prior weight is because it is currently arbitrary value (i.e., we give equal value to every plausible model in this study) and the prior is indeed relative to the posterior model weight considering data in the Bayesian model viewpoint. We used the term "prior" to imply the posterior model weight can also be used in our sensitivity analysis method. However, the estimation of posterior model weight is complex and not the focus of this sensitivity analysis research. And we found out the weight values have little influence on the sensitivity analysis results in this study. The integration of the posterior weight and sensitivity analysis is an important goal for our future research.

References

Dai, H., X. Chen, M. Ye, X. Song, and J. M. Zachara: A geostatisticsinformed hierarchical sensitivity analysis method for complex groundwater flow and transport modeling, Water Resour. Res., 53, doi: 10.1002/2016WR019756, 2017. Do Rosario, F. F., Custodio, E., & Da Silva, G. C. J.: Hydrogeology of the western amazon aquifer system (waas), Journal of South American earth ences, 72(dec.), 375-386, 2016. Fan, Y., and Miguez-Macho, G.: Potential groundwater contribution to Amazon evapotranspiration, Hydrol. Earth Syst. Sci., 14, 2039-2056, 10.5194/hess-14-2039-2010, 2010.

Please also note the supplement to this comment:
https://www.hydrol-earth-syst-sci-discuss.net/hess-2020-87/hess-2020-87-AC2-supplement.pdf
* * *

---

## Author Comment (AC3) · 2 Jun 2020

We greatly appreciate the reviewer for his/her positive evaluation and valuable feedback on our study. We believe these comments will greatly improve this research.

Major Comment 1

The overall motivation should be improved. It should not be "there lacked of research utlilizing quantitative and representative global sensitivity analysis" (line 59-61). I mean yes this is a gap but the primary objective should be to understand uncertainty sources and provide insights to physical processes that control ET and baseflow.

Response

We thank the reviewer for the valuable insights on the purpose of the study. We will improve the motivation such as "to discover and understand the different types of uncertainty sources of PBHMs and to further provide modelers insights of dominant physical processes that control hydrologic fluxes such as ET and baseflow etc.".

Major Comment 2

Section 2.2 should be greatly shortened, or moved to Supporting Information. It's way too long right now.

Response

We will shorten the Section 2.2 to highlight methodological improvements of the hierarchical sensitivity analysis in this study. And we are going to move the main equations of the hierarchical sensitivity analysis method to the appendix.

Major Comment 3

Conversely, some figures deserve more discussion, e.g., Figure 7 – I do not think it is the thickness right under the river cells, it's about overall thickness and how much water from the watershed concentrates to the channels. This hypothesis can be tested; Figure 9 – headwater vs stem river cells. Figure 11 – not really discussed much. How the authors came up with the six climate scenarios are not described. How could you have these different scenarios?

Response

We will add more discussions about figures, especially Figures 7, 9, 11. We agree with the reviewer that in Figure 7, the sensitivity index for aquifer thickness is about the average aquifer thickness for the whole watershed, rather than the thickness 'right' under the river cells. This is because the volume of groundwater to the river takes into account the contribution of overall watershed grids. We will replace the relevant expres-

sions in the manuscript. As for the Figure 9, we will add more discussions about the difference between headwater and stem river cells. In general, according to this figure, in headwater cells, aquifer thicknesses are usually the most important uncertainties; in stem river cells, model parameters are usually the most important uncertainties. We will expand our discussion of Figure 11 to specifically analyse the effects of three sub-divided groups of parameters on ET and QG. We present the generation of six climate scenarios in Section 2.4 (Page 11, Line 276-282). In general, we generated six climate scenarios based on statistical information from real climate data. The generation procedure for the new scenarios can be described as follows: the annual weather data from 1998 to 2013 were first collected and divided into multiple dry and wet seasons. Then, we sorted the different wet and dry seasons according to their total precipitation values during the whole season. Next, we divided these wet and dry seasons into three different groups representing six climate scenarios from wet to dry. The mean and standard deviation of the values of the different climate variables (e.g., precipitation, maximum temperature) for each group were calculated using the daily data. Finally, we generated random daily weather data for each climate scenario based on these mean and standard deviation data using a normal distribution. We will add these details in the revised manuscript.

Major Comment 4

Figure 4 is a big mess. A cleaner representation such as a boxplot is required.

Response

We will replace this figure to better exhibit the great uncertainty of the model simulation results.

Major Comment 5

The authors need to tone down the description of the hourly sensitivity especially around night. There may be many assumptions baked into how daily precipitation

is disaggregated into hourly which influenced these results. I doubt how robust this is.

Response

We understand that assumptions about averaging daily precipitation data to hourly data may affect the results. We will pay more attention to the interpretation when describing the hourly results.

Major Comment 6

Although not required, it will be nice to demonstrate the results for a year rather than 180 days. The annual cycle tells us more things.

Response

We agree with the reviewer on this point. However, we believe it may be unnecessary because we focused on the wet and dry seasons in the Amazon area for the climate conditions. And different climate scenarios with length of a half year were generated to represent alternative situations of these two seasons. Furthermore, the computational cost is too high for us if we used one year for simulation time. One annual cycle would be a good choice for this type of hydrological model cases, we will try to construct different annual climate scenarios in the future study.

Major Comment 7

The soil thickness should not be called "numerical model" uncertainty, but "subsurface stratigraphy".

Response

We believe the reviewer is talking about different confined and unconfined aquifer thicknesses. We understand the confusion of reviewer, but we believe the different thicknesses of distinct types of aquifers indeed lead to different conceptual hydrological models and represent a form of model uncertainty. We have used similar concept (different thicknesses for material A and B underground) for the model uncertainty in the

previous work (Dai et al., 2017). We will create other forms of model uncertainty especially using different mathematical models (i.e. different governing equations) in our future research.

Comment 1

Line 82, Michigan state –> Michigan or the state of Michigan. A relevant citation for this paragraph: Ji et al., 2019, 10.1029/2018WR023897

Response

We appreciate the reviewer to point this out and provide this reference. We would modify "Michigan state" to "Michigan" and include this reference in our manuscript.

Comment 2

Line 126. the model tool –> the modeling tool.

Response

Yes, we will revise this phrase.

Comment 3

Line 122, drainage network was formed –> formed.

Response

Yes, we will revise this phrase.

Comment 3

Line 287-292 paragraph-Brunke et al. 2016 is relevant to discuss 10.1175/JCLI-D-15-0307.1.

Response

We thank the reviewer for this reference. We will add this reference into our manuscript.

Comment 4

Line 307– on what machine did you run these many simulations and how much time did it take?

Response

In the case of parallel calculations using High Performance Computing (13 cores of Xeon 2.8G CPU), the average time spent on a single simulation was 2.8 minutes. 10,800 simulations took 3 weeks. We will add related descriptions in the revised manuscript.

Comment 5

Line 323 –> has little influence on spatially-averaged ET. (I believe for different cells it has a more prominent impact).

Response

Yes. The time-averaged ET result shows that the thickness of aquifers has different effects on different grid cells. We will revise these sentences to more accurately describe the results.

Comment 6

Line 326 –> temporally dependent –>time-dependent.

Response

Yes, we will revise this.

Comment 7

Line 331 "greatly decreasing"?? awkward phrasing.

Response

We will revise this phrase to "significantly decreased".

Comment 8

Line 356 – accumulation over time? Maybe also due to seasonality? We don't know for sure.

Response

Yes, there may not be only one explanation for this result. We will pay attention to our tone, and thus elaborate on the various possibilities for producing this result.

Comment 9

Line 354 – river flow always occurs hours later than the rainfall process — what if the rainfall isn't large enough to trigger a response?

Response

The original text refers to that considering the QG results, the sensitivity indices of the climate scenario always reach their maximum values around 1:00 am. We think that the reason of this pattern may be the exchange process between groundwater and river water always happens several hours later than the rainfall process, and the exchange volume always reaches its peak at around 1:00 am in the night. We believe if there was not enough rainfall to trigger the exchange of groundwater and rivers, this pattern of sensitivity analysis results would cease to exist. However, we believe that since the Amazon region receives a lot of rainfall, and we considered a total of six climate scenarios in this study, there is always more than one climate scenarios in which rainfall can trigger the exchange of groundwater and rivers.

Comment 10

Line 347-348– circular logic and tautology.

Response

We will revise this sentence as: "Groundwater has been demonstrated by previous research to be crucial for soil moisture in the Amazon region (Miguez-Macho and Fan, 2012b). Meanwhile, it also exerts a significant buffering effect on maintaining evapotranspiration during dry seasons (Miguez-Macho and Fan, 2012a; Pokhrel et al., 2014). The model PAWS uses the output QG to quantify the variation of groundwater volumes and measure the interaction process between groundwater and rivers. It is essential to implement sensitivity analysis to investigate which factor is most influential to this groundwater exchange process." in the revised manuscript. We will add some descriptions in the method section to explain why we chose QG as the output of interest.

References

Dai, H., X. Chen, M. Ye, X. Song, and J. M. Zachara: A geostatisticsinformed hierarchical sensitivity analysis method for complex groundwater flow and transport modeling, Water Resour. Res., 53, doi:10.1002/2016WR019756, 2017. Miguez-Macho, G., and Fan, Y.: The role of groundwater in the Amazon water cycle: 1. Influence on seasonal streamflow, flooding and wetlands, Journal of Geophysical Research: Atmospheres, 117, 10.1029/2012jd017539, 2012a. Miguez-Macho, G., and Fan, Y.: The role of groundwater in the Amazon water cycle: 2. Influence on seasonal soil moisture and evapotranspiration, Journal of Geophysical Research: Atmospheres, 117, 10.1029/2012jd017540, 2012b. Pokhrel, Y. N., Fan, Y., and Miguez-Macho, G.: Potential hydrologic changes in the Amazon by the end of the 21st century and the groundwater buffer, Environ Res Lett, 9, 084004, 10.1088/1748-9326/9/8/084004, 2014.

Please also note the supplement to this comment:
https://www.hydrol-earth-syst-sci-discuss.net/hess-2020-87/hess-2020-87-AC3-supplement.pdf

---

## Author Comment (AC4) · 2 Jun 2020

We greatly appreciate the editor for evaluating and processing our manuscript. We will substantially revise the manuscript following the valuable comments and suggestions provided by the three reviewers. The point to point responses for every comment have been arranged in the response letters. We summarize the planned major revisions for the manuscript as follows:

(1) Both reviewer #1 and reviewer #3 made comments about the purpose or motivation of this study needs to be more highlighted. To improve this weakness, we will rewrite the abstract and introduction sections to highlight and emphasize the motivation and

goals of this paper: to develop a new tool and demonstrate its implementation to a pilot example for comprehensive global sensitivity analysis of large-scale hydrological modelling. We believe that this work would be helpful to discover and understand the different types of uncertainty sources of PBHMs and to further provide modelers insights of dominant physical processes that control hydrologic fluxes such as ET and baseflow etc.

(2) Reviewer #1 made a major comment about the writing of this work needs to be improved, especially the logic. We will substantially revise this manuscript, to make sure that the logic flows smoothly and avoid making any conclusions without any justification. Besides, we will hire the professional English language editors to polish the language of this manuscript.

(3) Both reviewer #1 and reviewer #3 made comments about a few figures in manuscript need revisions and more discussions. Based on the comment of reviewer #1, we will update Figure 2 using a higher resolution and a larger font. Following the suggestion of reviewer #3, we will replace Figure 4 to find a better way to exhibit the great uncertainty of the model simulation results. We will also add more discussions of some figures, e.g., in Figure 7, the sensitivity index for aquifer thickness is about the average aquifer thickness for the whole watershed, rather than the thickness 'right' under the river cells. As for the Figure 9, we will add more discussions about the difference between headwater and stem river cells. Moreover, we will expand our discussions of Figure 11 to further analyse the effects of three subdivided groups of parameters on ET and QG.

(4) Reviewer #1 suggests us to highlight the contribution and novelty of this new method, thereby distinguishing it from previous work. And reviewer #3 thinks we should shorten Section 2.2 or move this section to appendix. To address these comments, we are going to move the main equations of the hierarchical sensitivity analysis method to appendix. And we will focus on the improvements we made to the previous hierarchical sensitivity in the method section.

(5) Reviewer #1 poses questions about the efficiency of the LHS method and the applicability of the binning method. We provide preliminary responses to Comments 9 and 10 in the letter to reviewer #1. We will present and explain more details about these two methods in the revised manuscript.

(6) All three reviewers made a suggestion for adding additional discussions of results in the manuscript. Based on the comment of reviewer #1, we will add discussions of insights learned from this pilot example. As suggested by the reviewer #2, we will expand the results and discussions on the relative importance of unconfined and confined aquifers.

Please also note the supplement to this comment:
https://www.hydrol-earth-syst-sci-discuss.net/hess-2020-87/hess-2020-87-AC4-supplement.pdf

---

## Author Response (AR1)

**Reply to Reviewers' Comments on Manuscript HESS-2020-87**

We thank the Editor, Associate Editor and the anonymous reviewers for their thoughtful and insightful review comments. The manuscript has been substantially revised, and an item-by-item response to the review comments is provided below.

**Responses to Associate Editor's Evaluation:**

**General Evaluation**

The totality of the revisions requested by the Reviewers who served during the review process paint a very constructive picture, with reference to both methodological, result, and presentation aspects. On the basis of the Authors' replies, I would be willing to give the Authors a possibility to revise their work in the context of a set of major revisions. The manuscript will then be subject to an additional round of reviews, possibly by the most critical reviewers who served during the current stage, in case they are still available. It has to be clear that in case the Reviewers' comments and concerns are not unambiguously satisfied, the manuscript will finally be released.

**Response**

We greatly appreciate the opportunity to revise the manuscript. We have substantially revised the manuscript following the valuable comments and suggestions provided by the three reviewers. The point-to-point responses for every comment have been arranged in the response letters. We summarize the major revisions of the manuscript as follows:

(1) Both reviewer #1 and reviewer #3 commented that the research purpose or scientific motivation of this study needs to be better highlighted. Reviewer #1 further commented that the objective, contribution and novelty of this work need to be improved. To address these comments, we have completely rewritten the abstract and introduction sections to highlight and emphasize the motivation and goals of this paper. The contributions, novelty, and significance of this work are also emphasized in the new introduction section.

(2) Reviewer #1 commented that the writing of this work needed to be improved, especially the logical flow of the introduction. To address these comments, we have substantially revised this manuscript to ensure that the logic flows smoothly, and we avoid making any conclusions without any justification. Both the introduction and conclusion sections have been completely rewritten with a rearranged logical flow. Furthermore, we have hired professional English language editors from Springer Nature to edit the language of this manuscript.

(3) Reviewer #1 and reviewer #2 commented that additional discussions of the results should be added to the manuscript. To address the comment of reviewer #1, we added a new discussion section (section 3.6) to discuss the insights learned from the results of this pilot global sensitivity analysis example. As suggested by reviewer #2, we have expanded the results and discussions on the relative importance of unconfined and confined aquifers.

(4) Both reviewer #1 and reviewer #3 commented that some figures in the manuscript required revisions and further discussions. To address the comment of reviewer #1, we updated Figure 2 using a higher resolution and a larger font. Following the suggestion of reviewer #3, we replaced Figure 4 with a new box plot to exhibit the great uncertainty of the model simulation results. We also added more discussions regarding Figure 7, Figure 9, and Figure 11.

(5) Reviewer #1 suggested highlighting the contribution and novelty of this new method, thereby distinguishing it from previous work. Reviewer #3 suggested that we shorten Section 2.2 or move this section to the Appendix. To address these comments, we moved the main equations of the hierarchical sensitivity analysis method to Appendix B. We focus on the improvements we made to the previous hierarchical sensitivity in the methods section. The new equations with implementation of binning method have been totally revised.

(6) Reviewer #2 commented on the comparison of parameter sensitivity analysis results for unconfined and confined aquifers. To address this comment, we have

added Appendix C and Figure C.1 to demonstrate the detailed sensitivity indices for all six parameters.

**Responses to Referee #1:**

We appreciate the reviewer for his/her valuable comments and feedback on our study. We believe these comments will greatly improve this research.

**General Evaluation 1**

The authors did a lot of work, but I think the writing needs a great improvement to highlight their work. The current writing reads more like a report rather than a scientific paper. The scientific motivation is not very clear to me.

**Response**

We appreciate the reviewer's evaluation of this manuscript and constructive comments. We have totally revised this paper, especially the abstract, introduction, discussions and conclusions sections. The contents of these sections have been reorganized and rewritten in order to highlight and emphasize our research motivation and goals: to develop a new tool and demonstrate its implementation to a pilot example for comprehensive global sensitivity analysis of large-scale hydrological modelling.

Some important revisions about our scientific motivations are summarized as follows:

In the abstract part, we introduce the motivation of this study as follows: *"Therefore, a global sensitivity analysis method that is capable of simultaneously analyzing multiple uncertainty sources of PBHMs and providing quantitative sensitivity analysis results is still lacking. In an effort to develop a new tool for overcoming these weaknesses, we improved the hierarchical sensitivity analysis method by defining a new set of sensitivity indices for subdivided parameters. A new binning method and Latin hypercube sampling (LHS) were implemented for estimating these new sensitivity indices."* (Line 17-22).

In the introduction part, the goal of this research is summarized as follows: *"By developing the new hierarchical sensitivity analysis method and implementing it in this test case, we aim to (1) provide a new tool and pilot example of comprehensive global sensitivity analysis for the PBHMs; (2) identify the most important uncertainty sources for modeling hydrological processes in the Amazon; and (3) investigate the possible*

*patterns for sensitivity analysis results of PBHMs.”* (Line 113-116).

**General Evaluation 2**

The authors said the aim of this work is to provide a pilot example of comprehensive global sensitivity analysis for large-scale PBHMs, then what lessons can the audience learn from this pilot example? Please provide a detailed discussion.

**Response**

We have added our insights for this comprehensive global sensitivity analysis in the new section of Section 3.6. We discussed the patterns found in the sensitivity analysis results of this pilot example. The following discussions have been added in the Section 3.6:

*“The results from this case study exhibit the importance of parameters, especially the vadose zone parameters, for ET and $Q_G$ predictions. Furthermore, according to the space-accumulative results, the climate scenario is also an important uncertainty source for ET predictions, especially at 12:00. Meanwhile, the thickness of the aquifer has a nonignorable influence on the $Q_G$ predictions on the groundwater grid cells. Moreover, according to the results of adjusting the climate scenario and model weights, the change in model (aquifer thickness) weights only has a small impact on the importance of different uncertainties. When the probability of occurrence of the extreme humid season is high, the importance of the parameters increases significantly. However, when the probability of occurrence of the extreme dry season is high, the main factors affecting ET predation are still the climate scenario unless the probability of occurrence of CS is greater than 90%. Although these patterns of sensitivity analysis results may not be universally correct, they can still provide useful insights for other modelers with similar cases and models.*

*In addition to the specific results, we also have some new insights into the general patterns of sensitivity analysis for the PBHMs provided by this pilot case. For instance, first, the ranks of importance of uncertain inputs are totally different for different model outputs, e.g., CS have a large impact on ET predictions but a small impact on $Q_G$ predictions. There is no one set of results that are valid for all different model outputs.*

*Second, the sensitivity analysis results of ET and $Q_G$ predictions show that the uncertainty has high temporal and spatial variability, which reflects that for very complex hydrological models, such as PBHMs, it is incorrect to generalize the sensitivity analysis results of a grid or a timestep to the entire watershed or the entire simulation cycle. Third, it is necessary to implement such a comprehensive global sensitivity analysis method that considers more than parametric uncertainty for the large-scale PBHMs since the sensitivity analysis results showed that other sources of uncertainty (e.g., climate scenario and model uncertainties) are essential as well for model predictions. Finally, evaluating the sensitivity of the parameters in detail is essential for PBHMs. For such a complex surface-subsurface coupling model, the new sensitivity analysis method can efficiently identify the uncertain inputs that have the greatest impact on the model outputs. This process can greatly improve our understanding of the complex model system and save time that is normally spent calibrating the model."* (Line 471-493).

**General Evaluation 3**

In the introduction, please highlight the objective, contribution and novelty of this work, and justify its significance.

**Response**

We have totally rewritten the introduction section with different logical flow and new content. After revision, the objectives of this study are highlighted as

*"By developing the new hierarchical sensitivity analysis method and implementing it in this test case, we aim to (1) provide a new tool and pilot example of comprehensive global sensitivity analysis for the PBHMs; (2) identify the most important uncertainty sources for modeling hydrological processes in the Amazon; and (3) investigate the possible patterns for sensitivity analysis results of PBHMs."* (Line 113-116).

We have highlighted the contribution and novelty of this work in the introduction part as:

*"This research presents a new tool of the improved hierarchical sensitivity analysis*

*method and demonstrates its implementation to a pilot example for comprehensive global sensitivity analysis of large-scale PBHMs. A new set of subdivided parametric sensitivity indices was defined to quantify the importance of a physical process involving only partial model parameters. A new binning method was implemented with the Latin hypercube sampling (LHS) method to estimate these subdivided parameter sensitivity indices. The LHS method also makes the assessment of hierarchical sensitivity analysis for large-scale PBHMs more computationally affordable compared with the original Monte Carlo method. This new and flexible hierarchical sensitivity analysis method provides modelers with the novel capability of analyzing sensitivity from the physical process viewpoint and estimating accurate importance for further subdivided parameter groups."* (Line 91-98).

We have totally revised the introduction section to further justify the significance of our research. In addition to the novelty and contributions of our work described above, we have pointed out the lack of research for the comprehensive global sensitivity analysis of PBHMs and the two main obstacles researchers are facing. Furthermore, we have stated our previous hierarchical sensitivity analysis cannot provide detailed parametric sensitivity index and the new improved hierarchical method developed in this research can be implemented to overcome these problems (the revised texts are in Line 62-90): "*To date, considerable research has been conducted to reduce the uncertainties in hydrological models by using local or global sensitivity analysis methods (e.g., Nijssen et al., 2001; Chávarri et al., 2013; de Paiva et al., 2013). However, conducting a comprehensive global sensitivity analysis, especially variance-based sensitivity analysis on PBHMs, remains a challenge, and there are two main obstacles. The first obstacle is the high computational cost rising from two sources: the high complexity of the model itself and the method requirement of variance-based global sensitivity analysis. A PBHM usually has a very large number of parameters and multiple high-order nonlinear governing equations. These facts combined with a large-scale model domain cause the running of a PBHM itself to be very computationally expensive. For the sensitivity analysis method, compared with the local sensitivity analysis, which can only provide results valid in a certain range of parameter values (e.g., the derivative of the model prediction with respect to parameter A at a certain value point can be a measurement of A's local sensitivity at this point), the global sensitivity analysis is more comprehensive because its results are valid for the whole range of parameter values.*

*To achieve this goal, the methods of global sensitivity analysis are all relatively computationally expensive, especially for the variance-based method, which uses complex sampling techniques, and its computational cost grows exponentially with the number of parameters (Saltelli et al., 2000, 2010). Therefore, the implementation of a global sensitivity analysis for a PBHM leads to an extremely high computational cost considering that we have to run a large number of simulations for a complex PBHM using different parameter samples.*

*The second obstacle of implementing the global sensitivity analysis method in PBHMs is the variant uncertainty sources included in the model. Conventional global sensitivity analysis generally considers only uncertainty from model parameters and ignores other important hydrological model uncertainties. However, for PBHMs, uncertainties usually arise from three different sources, including parametric uncertainty, model structural uncertainty (induced through multiple different plausible conceptual or mathematical models), and scenario uncertainty (caused by alternative unpredictable future climate conditions) (Ye et al., 2005; Makler-Pick et al., 2011; Neumann, 2012; Dai and Ye, 2015; Song et al., 2015; Dai et al., 2017a, 2017b; Zeng et al., 2018; Pan et al., 2020). To overcome these two obstacles, Dai et al. (2017a) developed a new hierarchical sensitivity analysis method that integrates the variance-based method and hierarchical uncertainty framework. By combining uncertain inputs based on their characteristics and dependencies, hierarchical sensitivity analysis can quantify the sensitivity of different sources of uncertainty involved in hydrological models (e.g., parameters, models, and climate scenarios) and dramatically reduce the computational cost. However, the original hierarchical sensitivity analysis method is limited to considering parameters as a whole, and the sensitivity indices of different parameters cannot be defined or estimated. This simple strategy may be adequate for a groundwater modeling case, but it cannot provide detailed information for a PBHM that includes multiple hydrological processes.*".

**General Evaluation 4**

I think the authors need pay more attention to the writing. The logic is not very clear and sometimes the conclusive sentences pop out without justification.

**Response**

We have revised the manuscript substantially to make sure that the logic flows smoothly and avoid making any conclusions without any justification. The abstract, introduction, and conclusions sections have been totally rewritten. And we have hired professional editor for revising the grammar.

**Comment 1**

The grammar of the title is not right.

**Response**

We have revised the title as *"Hierarchical Sensitivity Analysis for A Large-scale Process-based Hydrological Model Applied in an Amazonian Watershed".*

**Comment 2**

I think the abstract needs to be rewritten. Right now, it reads like a report instead of a scientific paper. I did not see a scientific motivation but a description of what the authors did.

**Response**

We have totally rewritten the abstract. The basic scientific motivation of this work has been highlighted as to develop a new computational affordable tool for comprehensive global sensitivity analysis of large-scale complex PBHMs with considering various uncertainty sources and physical processes. And we have rewritten the scientific motivation of this research as:

"*Therefore, a global sensitivity analysis method that is capable of simultaneously analyzing multiple uncertainty sources of PBHMs and providing quantitative sensitivity analysis results is still lacking. In an effort to develop a new tool for overcoming these weaknesses, we improved the hierarchical sensitivity analysis method by defining a new set of sensitivity indices for subdivided parameters. A new binning method and Latin hypercube sampling (LHS) were implemented for estimating these new sensitivity indices.*" (Line 18-20).

**Comment 3**

I think the logic of the introduction needs an improvement.

**Response**

We have totally rewritten the introduction section and the basic logic of this part has been rearranged as:

1. Introduce the background of the PBHM model and the importance of sensitivity analysis of the PBHM model.
2. Present an overview of traditional sensitivity analysis methods.
3. Present the challenges encountered in the analysis of PBHMs using traditional sensitivity analysis methods.
4. Introduce the advantages and disadvantages of conventional hierarchical sensitivity analysis method.
5. State the purpose and contribution of this research with highlighting the novelty of our method.
6. Provide an overview of the structure of this paper.

**Comment 4**

Line 47-50, I found this last sentence is confusing. SA "becomes" important? Limited resources?

**Response**

We understand the reviewer's confusion. The SA becomes important because the uncertainty of model predictions becomes larger and more important with model complexity increases. The limited resources refer to the limited funding and manpower which can be used to reduce the uncertainty of inputs through obtaining more data and calibrating the input values. The sensitivity analysis can save these resources by identifying the most important uncertain inputs. We have revised this sentence and added more references. These sentences have been revised as:

*"However, these complex processes and governing equations embedded in the PBHMs*

*inevitably induce large uncertainties in the modeling predictions (Neuman, 2003; Rojas et al., 2010; Lu et al., 2012; Shen et al., 2014; Razavi and Gupta, 2015, 2016; Qiu et al., 2019). How to efficiently decrease these large uncertainties becomes an essential problem for modelers. Sensitivity analysis aims to identify the most influential sources of uncertainty and is therefore an important tool (Saltelli and Sobol, 1995; Saltelli et al., 2000, 2010; Song et al., 2015). The sensitivity analysis results assist modelers and managers in focusing on observing and calibrating the uncertain inputs that have the greatest influences on model outputs. Thus, the sensitivity analysis process saves resources (e.g., funding and manpower) used for calibration and significantly improves the efficiency of reducing the uncertainty of PBHM predictions.*" (Line 42-50).

**Comment 5**

Line 59, this sentence is confusing. "Using" large-scale PBHMs? Why the computational cost is high?

**Response**

We understand the confusion caused by this term, and we have rewritten this sentence as:

"*However, conducting a comprehensive global sensitivity analysis, especially variance-based sensitivity analysis on PBHMs, remains a challenge, and there are two main obstacles.*" (Line 63-65).

There are generally two main reasons for the high computational cost of global sensitivity analysis of a large-scale PBHM: the high complexity of model itself and the high standard of global sensitivity analysis. A PBHM usually has a very large number of parameters and multiple high-order nonlinear governing equations. These facts combining with a large-scale model domain cause the running of a PBHM itself is already very computationally expensive. For the sensitivity analysis, comparing with the local sensitivity analysis which can only provide results valid in certain range of parameter values (e.g., the derivative of model prediction with respect to parameter A at certain value point can be a measurement of A's local sensitivity at this point), the global sensitivity analysis is more comprehensive because its results are valid for the

whole range of parameter value. To achieve this goal, the methods of global sensitivity analysis are all relatively computationally expensive and their computational cost grows exponentially with number of parameters. Therefore, the implementation of global sensitivity analysis for a PBHM leads to extremely high computational cost considering we have to run a large number of simulations for a complex PBHM using different parameter samples.

We have explained the reasons for this question in the rewritten introduction section:

"*The first obstacle is the high computational cost rising from two sources: the high complexity of the model itself and the method requirement of variance-based global sensitivity analysis. A PBHM usually has a very large number of parameters and multiple high-order nonlinear governing equations. These facts combined with a large-scale model domain cause the running of a PBHM itself to be very computationally expensive. For the sensitivity analysis method, compared with the local sensitivity analysis, which can only provide results valid in a certain range of parameter values (e.g., the derivative of the model prediction with respect to parameter A at a certain value point can be a measurement of A's local sensitivity at this point), the global sensitivity analysis is more comprehensive because its results are valid for the whole range of parameter values. To achieve this goal, the methods of global sensitivity analysis are all relatively computationally expensive, especially for the variance-based method, which uses complex sampling techniques, and its computational cost grows exponentially with the number of parameters (Saltelli et al., 2000, 2010). Therefore, the implementation of a global sensitivity analysis for a PBHM leads to an extremely high computational cost considering that we have to run a large number of simulations for a complex PBHM using different parameter samples.*" (Line 65-76).

**Comment 6**

Please justify why the authors chose the PAWS model as the pilot example.

**Response**

We have chosen the PAWS mainly for its high efficiency, great performance, and complex uncertain inputs. We have explained these details of choosing PAWS in the

introduction and methodology sections:

*"The process-based adaptive watershed simulator (PAWS) model was first developed in Shen and Phanikumar (2010); the PAWS is capable of simulating large catchments and long-term frames by efficiently coupling surface and subsurface hydrological processes. Coupling the PAWS with the CLM (Community Land Model) can enable the model to describe vegetation respiration and evapotranspiration in a physics-based manner (Shen et al., 2014; Niu et al., 2017). The model has been applied extensively in many watersheds, e.g., the large-scale watersheds in Michigan, U.S. (Shen et al., 2013, 2014, 2016; Niu et al., 2014, 2017; Ji et al., 2015; Qiu et al., 2019) and the watershed in the Amazon basin (Niu et al., 2017), and the model has presented good performances in these watersheds. The PAWS can also estimate multiple key variables of hydrological states and fluxes at different spatiotemporal scales. The high efficiency, great performance, and complex variables all make PAWS an excellent model choice for PBHMs to evaluate and demonstrate the sensitivity analysis method."* (Line 99-107).

*"The modeling tool used in this study is the PAWS model (Shen and Phanikumar, 2010; Shen et al., 2014; Niu et al., 2017). The main reason for choosing the PAWS as the pilot example of PBHMs is that compared with other PBHMs, the PAWS is a comprehensive and representative large-scale hydrological model that can be applied to large catchments and long-term frames by efficiently coupling both surface and subsurface hydrological processes (Shen and Phanikumar, 2010). The complexity and parameter dimensionality of the PAWS are high enough to test and demonstrate our new global sensitivity analysis method. Furthermore, the PAWS was previously applied to the studied watershed, and it was capable of simulating multiple key variables of hydrological states and fluxes at different spatiotemporal scales and presented good model performance validated by various ground and satellite observation data (Niu et al., 2017). This previous model application provides a solid basis for our uncertainty identification and sensitivity analysis study."* (Line 129-137).

**Comment 7**

It seems that the authors have some methodology development on the basis of their previous work. Please highlight these contributions and novelty, and justify that this new development is necessary for the complex and large-scale model sensitivity

analysis.

**Response**

We do have some new methodology developments based on the previous work: the further quantification for the subdivided parametric uncertainties and the implementation of LHS method and binning method. We have defined a new set of sensitivity indices for subdivided parameter groups and allows modelers to analyse the importance of a physical process only involves partial model parameters. These new sensitivity indices are necessary because the main weakness of our previous methodology is it considers all parameters together as one single process. This simple strategy may be adequate for a groundwater modelling case, but it cannot provide detailed information for a PBHM which includes multiple hydrological processes. The implementation of LHS method and binning method further reduce the computational cost and make this comprehensive global sensitivity analysis of large-scale complex PBHM plausible.

We have described the new methodology developments in the rewritten introduction section as:

 "*A new set of subdivided parametric sensitivity indices was defined to quantify the importance of a physical process involving only partial model parameters. A new binning method was implemented with the Latin hypercube sampling (LHS) method to estimate these subdivided parameter sensitivity indices. The LHS method also makes the assessment of hierarchical sensitivity analysis for large-scale PBHMs more computationally affordable compared with the original Monte Carlo method. This new and flexible hierarchical sensitivity analysis method provides modelers with the novel capability of analyzing sensitivity from the physical process viewpoint and estimating accurate importance for further subdivided parameter groups.*" (Line 92-98).

**Comment 8**

Line 221, why 600 samples?

**Response**

We understand the confusion of reviewer. First, based on the experiences of previous research cases, we believe 600 is an adequate parameter sample size for our test case considering the model domain and number of uncertain parameters. Second, because of the high computational cost, 600 parameter samples are the maximum size we can afford. After combining model uncertainty and climate scenario uncertainty, we need to conduct a total of 600×3×6=10,800 simulations for the PBHM. The pure simulation time without analysing data is already very time consuming even using the best high-performance computing (HPC) platform we have.

We have explained these reasons in the section 2.4 as:

"*The reasons for using 600 parameter samples in this study are because, first, based on the experiences of previous research cases (Emery et al., 2016; Dai et al., 2019), 600 is an adequate parameter sample size for this research considering the model domain and number of uncertain parameters; and second, considering the computational cost, 600 parameter samples are an appropriate sample size for this study. By combining model uncertainty and climate scenario uncertainty, there are 600 × 3 × 6 = 10,800 simulations in total. The pure simulation time without analyzing data is already very time consuming even when using the best high-performance computing (HPC) platform we have.*" (Line 313-318).

**Comment 9**

Line 223-224, in what sense the LHS greatly reduces the required sample size compared to MC sampling? To achieve the same estimation accuracy? Please provide evidence or reference.

**Response**

The LHS has been one popular sampling technique to reduce computational cost and it has been proven to be more effective than the conventional MC sampling method. We have provided more references to explain and exhibit the effectiveness of LHS method in the section 2.3 as follows:

"*Compared with the conventional Monte Carlo method, the LHS method can guarantee*

*space-filling and noncollapsing of parameter samples (Grosso et al., 2009; Crombecq et al., 2011; Husslage et al., 2011; Damblin et al., 2013; Ba et al., 2015; Qian, 2012), which means that the sampling points can be evenly distributed throughout the sampling region and that there are no two sampling points with the same value. Thus, LHS is a sampling method with higher sampling efficiency (Helton and Davis, 2003). The convergence rate of the conventional Monte Carlo method is $O(N^{-1/2})$, where N is the sample size (Caflisch, 1998). However, for a system where the parameters are simply distributed (e.g., uniformly distributed), the convergence rate of LHS can reach $O(N^{-3})$ (Iman and Conover, 1980). The LHS method has been one popular sampling technique used to reduce computational cost."* (Line 222-229).

**Comment 10**

Line 265, when does the binning method not work? Please comment.

**Response**

The binning method is a rigorously derived mathematical technique designed to separating and estimating the partial variances contributed from different parameters of one LHS method sampled parameter set. Because the mathematical equations are general and rigorous, this method can be applied to any modelling case with LHS parameter samplings. However, when the samplings for different parameters are totally random and unrelated like conventional Monte Carlo simulation, the binning method is not applicable.

We have added the explanations of this question in Section 2.3 as follows:

*"The binning method is a rigorously derived mathematical technique designed to separate and estimate the partial variances contributed from different parameters of one LHS method sampled parameter set. Because the mathematical equations are general and rigorous, this method can be applied to any modeling case with LHS parameter samplings. However, when the samplings for different parameters are totally random and unrelated, such as the conventional Monte Carlo simulation, the binning method is not applicable."* (Line 274-278).

**Comment 11**

Line 410, how do the authors justify the results accuracy?

**Response**

We intended to express that this hierarchical variance-based global sensitivity analysis method is capable of estimating accurate sensitivity measurements (sensitivity index) for the importance of certain uncertainty sources. Comparing with other sensitivity analysis methods which only provide qualitative results (e.g., ranks of importance), this variance-based quantitative global sensitivity analysis method is more accurate. It is more appropriate to replace the word "accurate" with "quantitative". We have removed this sentence to avoid confusion and the conclusions section has been rewritten to focus on the new methodology development and insights found in this pilot example.

**Comment 12**

Line 417-419, the sentence is confusion.

**Response**

We have rewritten the conclusions section and this sentence has been removed. A new sentence collaborates with the introduction section to explain the further decomposing of subdivided parametric uncertainties is added as follows:

"*A new set of sensitivity indices of subdivided parameters was defined to quantify the importance of processes that only involve partial parameters.*" (Line 498-499).

**Comment 13**

Line 419-420, the sentence comes out of nowhere.

**Response**

We have rewritten the conclusions section and removed this sentence.

**Comment 14**

Line 430, "all" is a strong work, be careful of using it.

**Response**

We have replaced "all" with "Three common groups of uncertainty sources", which is more accurate.

**Comment 15**

Line 434, the method can largely reduce the computational cost associated with complex, large-scale hydrological models. From which aspects to reduce the cost? Does it reduce the forward simulation time? Please be specific here.

**Response**

We understand the confusion caused by this sentence. What we want to express here is that the improved hierarchical sensitivity analysis method largely reduces the computational cost of global sensitivity analysis for PBHMs, not the computational cost of PBHM simulation itself. In order to express more accurately, we have removed this sentence and added a description for the improved methodology in the revised manuscript as follows:

"*The highly efficient sampling algorithm of the LHS and binning method were implemented for the estimation of sensitivity indices to reduce computational cost.*" (Line 499-500).

**Comment 16**

The words in Figure 2 are hard to read.

**Response**

We have updated this figure using a higher resolution and a larger font.

**Climate Scenario Uncertainty Sources**

| Precipitation | Temperature | Radiation | Humidity | Wind Speed |

**Numerical Model Uncertainty Sources**

| Thicknesses of unconfinced aquifers and confined aquifers |

**Parameter Uncertainty Sources**

| Vadose zone | Groundwater | Overland flow |
| --- | --- | --- |
| Soil saturated conductivity $Ks$ | Unconfined aquifer conductivity $K_1$ | Length of runoff contribution to overland flow domain $L$ |
| Van Genuchten parameter $\alpha$ | | |
| Van Genuchten parameter $N$ | Confined aquifer conductivity $K_2$ | |

[Figure]

Climate Scenario Uncertainty

Numerical Model Uncertainty

Parameter Uncertainty

Vadose zone

Groundwater

Overland flow

**Responses to Referee #2:**

We greatly appreciate the reviewer for his/her valuable comments and feedback on our study. We believe these comments will greatly improve this manuscript.

**Comment 1**

Line 55: Please give a brief description to other global sensitivity analysis methods, e.g., the sensitivity analysis based on information entropy.

**Response**

We have added more descriptions to other global sensitivity analysis method in the introduction as follows:

"*Common global sensitivity analysis methods include screening methods, regression-based methods, variance-based methods, meta-model methods (Song et al., 2013), and information-entropy-based methods (Zeng et al., 2012).*" (Line 57-58).

**Comment 2**

Line 290: Are the three aquifer models sufficient to investigate the sensitivity of the model output to aquifer thickness? Please justify it.

**Response**

We chose three aquifer models here because (1) the sediments throughout the Amazon Basin are relatively old, so the stratification of soil and aquifer is relatively stable and the thickness does not change much (Do Rosario et. al., 2016); (2) the lack of actual measurements prevents us to determine the boundary between unconfined and confined aquifers; (3) the thickness of the unconfined aquifer in the central Amazon is larger than 50 m, and the depth of the bedrock is very deep. These three aquifer models are: (1) 100 m and 200 m ($NM_1$), (2) 50 m and 250 m ($NM_2$), and (3) 250 m and 50 m ($NM_3$), respectively. These three models respectively represent the situations of (i) thick unconfined aquifer, large bedrock depth, (ii) similar thickness of the unconfined and the confined aquifer, medium bedrock depth, and (iii) thick confined aquifer, low

bedrock depth. And for the sensitivity analysis, the three different models are sufficient enough to generate a stable variance and sensitivity index.

We have justified the reasons that we chose these three aquifer models in the revised manuscript as follows:

*"Considering that (1) the stratification of the soil and aquifer is relatively stable, and the thickness does not change much (do Rosário et al., 2016), (2) there is a lack of actual measurements in this area to determine the stratification of unconfined aquifers and confined aquifers, and (3) according to Pelletier et al. (2016), the thickness of the unconfined aquifer in the central Amazon is larger than 50 m, and the depth of the bedrock is very deep, as three aquifer models involving different thicknesses of the unconfined and confined aquifers were generated to investigate the sensitivity of the model outputs to aquifer thickness. These three aquifer models involving different thicknesses of the unconfined and confined aquifers are (1) 100 m and 200 m (NM1), (2) 50 m and 250 m (NM2), and (3) 250 m and 50 m (NM3), respectively. These three models represent the situations of (i) similar thickness of the unconfined and confined aquifer, medium bedrock depth, (ii) thick confined aquifer, low bedrock depth, and (iii) thick unconfined aquifer, large bedrock depth."* (Line 303-311).

**Comment 3**

Line 293: Are these model weights used for model averaging or model combination?

**Response**

We are indeed using these weights for the model averaging process involved in the variance calculations of sensitivity analysis. The hierarchical sensitivity analysis method requires the weight of model $NM_k$ under scenario $CS_l$ satisfying $\sum_k P(NM_k|CS_l) = 1$, and the weight of scenarios satisfying $\sum_l P(CS_l) = 1$. In this study, we assume that different climate scenarios have the same weight. We also assume that the weights of the different models under each climate scenario are the same. In other words, for 6 climate scenarios, the weight of each climate scenario is 1/6; for each of the 3 models under a certain climate scenario, the weight of each model is 1/3.

**Comment 4**

Line 297: Parameters' ranges have influence on the results of sensitivity analysis, please explain the allowable ranges of these parameters in Table 2.

**Response**

We chose those allowable ranges for parameters mainly because of the strata characteristics of the study site. We have added explanations to allowable ranges of these parameters in the revised manuscript as follows:

"*According to the study of Cuartas et al. (2012), the clay content in the northwestern part of Manaus is very high (65-90%). Considering the difference in regional soil texture (Fisher et al., 2008; Teixeira et al., 2014), the allowable range of soil saturated conductivity Ks selected in this study is between 0-10 m day$^{-1}$. The ranges of unconfined aquifer conductivity, $K_1$, and confined aquifer conductivity, $K_2$, are chosen as 0-10 m day$^{-1}$ and 0-60 m day$^{-1}$, respectively. The results of the model calibration in Niu et al. (2017), which are related to the characteristics of the soil and groundwater layers in the watershed (Oleson et al., 2008; Christoffersen et al., 2014), are used to define the mean values of distributions used for these uncertain parameters. The soil saturated conductivity, $K_s$, unconfined aquifer conductivity, $K_1$, and confined aquifer conductivity, $K_2$, were assumed to follow lognormal distributions (log-N (1.6094, 0.4214$^2$), log-N (3.4012, 0.4214$^2$), and log-N (1.6094, 0.4214$^2$), respectively). The remaining three parameters (α, N, and L) were assumed to follow a uniform distribution: U (0.1, 4), U (1.03, 5), and U (20, 700). The allowable ranges of these six parameters are listed in Table 2.*" (Line 320-329).

**Comment 5**

Line 346: It is good to investigate the contribution of groundwater system to stream-flow, and this research find that the thickness of an aquifer will greatly influence the water redistribution process in the aquifer. However, I want to see the influences of aquifers' thicknesses on streamflow in more detail, such as, the unconfined aquifer and the confined aquifer, It is expected that the unconfined aquifer has more influence on streamflow, because it has stronger interaction with surface flow.

**Response**

We thank the reviewer for the suggestion. And we think the comparison for the importance of unconfined and confined aquifers is indeed worth to investigate. However, limited by the scope of this study, we compared the hydraulic conductivity of the unconfined aquifer with that of the confined aquifer, rather than the thicknesses of aquifers. We think this also explains to a certain extent which aquifer has a greater influence on $Q_G$. In this regard, we have added Appendix C and Fig. (C.1) to display the results. The results confirmed that the unconfined aquifer has more influence on streamflow.

[Figure]

Figure C.1. Maps of six parameter sensitivity indices for groundwater contribution to stream flow ($Q_G$) predictions.

**Comment 6**

Line 383: what's the meaning of prior weights here, will they used for Bayesian model averaging?

**Response**

We understand the confusion of the reviewer. We called this weight prior weight is because it is currently arbitrary value (i.e., we give equal value to every plausible model in this study) and the prior is indeed relative to the posterior model weight considering data in the Bayesian model viewpoint. We used the term "prior" to imply the posterior model weight can also be used in our sensitivity analysis method. However, the estimation of posterior model weight is complex and not the focus of this sensitivity analysis research. And we found out the weight values have little influence on the sensitivity analysis results in this study. The integration of the posterior weight and sensitivity analysis is an important goal for our future research.

**Responses to Referee #3:**

We greatly appreciate the reviewer for his/her positive evaluation and valuable feedback on our study. We believe these comments will greatly improve this research.

**Major Comment 1**

The overall motivation should be improved. It should not be "there lacked of research utlilizing quantitative and representative global sensitivity analysis" (line 59-61). I mean yes this is a gap but the primary objective should be to understand uncertainty sources and provide insights to physical processes that control ET and baseflow.

**Response**

We thank the reviewer for the valuable insights on the purpose of the study. We have totally rewritten the introduction section and improved the motivation as follows:

"*This research presents a new tool of the improved hierarchical sensitivity analysis method and demonstrates its implementation to a pilot example for comprehensive global sensitivity analysis of large-scale PBHMs.*" (Line 91-92).

**Major Comment 2**

Section 2.2 should be greatly shortened, or moved to Supporting Information. It's way too long right now.

**Response**

We have shortened the Section 2.2 to highlight methodological improvements made in this research for the original hierarchical sensitivity analysis. And we have moved the main equations of the original hierarchical sensitivity analysis method to the Appendix B.

**Major Comment 3**

Conversely, some figures deserve more discussion, e.g., Figure 7 – I do not think it is the thickness right under the river cells, it's about overall thickness and how much water

from the watershed concentrates to the channels. This hypothesis can be tested; Figure 9 – headwater vs stem river cells. Figure 11 – not really discussed much.

How the authors came up with the six climate scenarios are not described. How could you have these different scenarios?

**Response**

We agree with the reviewer that in Figure 7, the sensitivity index for aquifer thickness is about the average aquifer thickness for the whole watershed, rather than the thickness 'right' under the river cells. This is because the volume of groundwater to the river considers the contribution of overall watershed grids. We have replaced the relevant expressions in the manuscript.

As for the Figure 9, we have added more discussions about the difference between groundwater and stem river cells as follows:

"*Because groundwater exchange with stream flow occurs only at grid cells along the streams, the sensitivity indices only have valid values in those stream grid cells (Fig. 9). Our results indicate that considering most grid cells, the parameters are the most important contributor to the uncertainty of time-accumulative $Q_G$ predictions, and the second most important factor is aquifer thickness. However, if we divide the grid cells into groundwater and stem river grid cells based on their location relative to the river and aquifer type, the sensitivity analysis results are totally different in these two types of grid cells. The model parameter uncertainty is usually the most important in stem river grid cells; in contrast, the aquifer thicknesses contribute the largest portion of the uncertainty in groundwater grid cells. This pattern of results may be caused by the unconfined aquifer and river being unconnected in the stem river grid cells, and there is an unsaturated zone between two of them. Therefore, the soil parameters affect $Q_G$ predictions in stem river grid cells. Moreover, the groundwater table is relatively high, and the groundwater is directly connected with rivers in the groundwater grid cells. Thus, the aquifer thicknesses are more important under this condition.*" (Line 404-414).

We have expanded our discussion of Fig. 11 to specifically analyse the effects of three subdivided groups of parameters on *ET* and $Q_G$ as follows:

*"We plotted the time-accumulative subdivided parametric sensitivity indices for ET in Fig. 11(a) and for $Q_G$ in Fig. 11(b). Considering ET as our output, for most grids, the vadose zone parameters are the most important contributor to parametric uncertainties. Compared with that on other grids, the influence of groundwater parameters on the river grids is more significant (Fig. 11(a)). For the $Q_G$ results, the vadose zone parameters generally dominate the parametric sensitivities for most grids (Fig. 11(b)). However, if considering different types of grid cells, we find that the vadose zone parameters mainly affect the stem river grid cells and have a relatively small influence on the groundwater grid cells. This pattern coincides with our hypotheses that there is an unsaturated zone between the stem rivers and groundwater."* (Line 434-440).

We present the generation of six climate scenarios in Section 2.4 (Line 290-296). In general, we generated six climate scenarios based on statistical information from real climate data. The generation procedure for the new scenarios can be described as follows: the annual weather data from 1998 to 2013 were first collected and divided into multiple dry and wet seasons. Then, we sorted the different wet and dry seasons according to their total precipitation values during the whole season. Next, we divided these wet and dry seasons into three different groups representing six climate scenarios from wet to dry. The mean and standard deviation of the values of the different climate variables (e.g., precipitation, maximum temperature) for each group were calculated using the daily data. Finally, we generated random daily weather data for each climate scenario based on these mean and standard deviation data using a normal distribution.

**Major Comment 4**

Figure 4 is a big mess. A cleaner representation such as a boxplot is required.

**Response**

We have replaced this figure to exhibit the great uncertainty of the model simulation results.

[Figure]

*Figure 4. The spatial-accumulative outputs for evapotranspiration (ET) (a) and groundwater contribution to stream flow ($Q_G$) (b) at all time steps and considering all of three uncertainties. All the time steps are divided into different groups based on local time. Different groups represent different hours of the day.*

**Major Comment 5**

The authors need to tone down the description of the hourly sensitivity especially around night. There may be many assumptions baked into how daily precipitation is disaggregated into hourly which influenced these results. I doubt how robust this is.

**Response**

We understand that assumptions about averaging daily precipitation data to hourly data may affect the results. We have adjusted our tones to describe results. For the reason of small sensitivity indices of the climate scenarios at night for *ET* are described as:

"*A possible explanation for this result might be that precipitation and radiation forcing all decrease to zero during this period, leading to a decrease in the sensitivity indices for the climate scenarios ($S_{CS}$).*" (Line 367-369).

For the reason of large sensitivity indices of the climate scenarios and models at 1:00

are described as:

"*In terms of $S_{CS}$, this may be because the exchange between groundwater and river flow occurs hours later than the rainfall process, and the amount of water during the exchange process always reaches its peak at night, at approximately 1:00. The $S_{NM}$ might be because the thickness of aquifers will greatly influence the water redistribution process in the aquifer.*" (Line 398-401).

**Major Comment 6**

Although not required, it will be nice to demonstrate the results for a year rather than 180 days. The annual cycle tells us more things.

**Response**

We agree with the reviewer on this point. However, we believe it may be unnecessary because we focused on the wet and dry seasons in the Amazon area for the climate conditions. And different climate scenarios with length of a half year were generated to represent alternative situations of these two seasons. Furthermore, the computational cost is too high for us if we used one year for simulation time. However, we totally agree with the reviewer that one annual cycle would be a better choice for this type of hydrological model cases, we will try to construct different annual climate scenarios in the future study.

**Major Comment 7**

The soil thickness should not be called "numerical model" uncertainty, but "subsurface stratigraphy".

**Response**

We believe the reviewer is talking about different confined and unconfined aquifer thicknesses. We understand the confusion of reviewer, but we believe the different thicknesses of distinct types of aquifers indeed lead to different conceptual hydrological models and represent a form of model uncertainty. We have used similar concept (different thicknesses for material A and B underground) for the model uncertainty in

the previous work (Dai et al., 2017). We will create other forms of model uncertainty especially using different mathematical models (i.e. different governing equations) in our future research.

**Comment 1**

Line 82, Michigan state –> Michigan or the state of Michigan. A relevant citation for this paragraph: Ji et al., 2019, 10.1029/2018WR023897

**Response**

We appreciate the reviewer to point this out and provide this reference. We have modified "Michigan state" to "Michigan" (Line 103).

**Comment 2**

Line 126. the model tool –> the modeling tool.

**Response**

Yes, we have revised this phrase in Line 129.

**Comment 3**

Line 122, drainage network was formed –> formed.

**Response**

Yes, we have revised this phrase.

**Comment 3**

Line 287-292 paragraph-Brunke et al. 2016 is relevant to discuss 10.1175/JCLI-D-15-0307.1.

**Response**

We thank the reviewer for this reference. We have added this reference into our manuscript as follows:

"*For the model uncertainty, the research of Brunke et al. (2016) shows that the shallow bedrock depth or deep bedrock depth has a great influence on surface runoff and base flow in CLM.*" (Line 300-301).

**Comment 4**

Line 307– on what machine did you run these many simulations and how much time did it take?

**Response**

We have added descriptions for this question in the revised manuscript as follows:

"*We used the parallel computing technique for running these simulations through the HPC platform (13 cores of Xeon 2.8G CPU). The average time spent on a single simulation was 2.8 minutes, and a total of 10,800 simulations were run for 3 weeks.*" (Line 339-341).

**Comment 5**

Line 323 –> has little influence on spatially-averaged ET. (I believe for different cells it has a more prominent impact).

**Response**

Yes, the time-accumulative ET result shows that the thickness of aquifers has different effects on different grid cells. Such as the result we described as:

"*However, for stream grid cells, the importance of aquifer thicknesses increases. Therefore, the parameters and aquifer thicknesses are both important.*" (Line 380-382).

**Comment 6**

Line 326 –> temporally dependent –>time-dependent.

**Response**

Yes, we have revised this term.

**Comment 7**

Line 331 "greatly decreasing"?? awkward phrasing.

**Response**

We have revised this phrase as:

"*leading to a decrease in the sensitivity indices for the climate scenarios ($S_{CS}$).*" (Line 368-369).

**Comment 8**

Line 356 – accumulation over time? Maybe also due to seasonality? We don't know for sure.

**Response**

Yes, these sentences are confusing and there may be more than one explanation for this pattern. We have changed these sentences as follows:

"*Another pattern demonstrated in Fig. 8 is that the values of $S_{CS}$ generally increase with time. This trend may be caused by the seasonality effect of CS or the long-term cumulative influence of CS on the groundwater flow.*" (Line 385-388).

**Comment 9**

Line 354 – river flow always occurs hours later than the rainfall process — what if the rainfall isn't large enough to trigger a response?

**Response**

The original text refers to that considering the $Q_G$ results, the sensitivity indices of the

climate scenario always reach their maximum values around 1:00 am. We think that the reason of this pattern may be the exchange process between groundwater and river water always happens several hours later than the rainfall process, and the exchange volume always reaches its peak at around 1:00 am in the night. We believe if there was not enough rainfall to trigger the exchange of groundwater and rivers, this pattern of sensitivity analysis results would cease to exist.

However, we believe that since the Amazon region receives a lot of rainfall, and we considered a total of six climate scenarios in this study, there is always more than one climate scenarios in which rainfall can trigger the exchange of groundwater and rivers.

**Comment 8**

Line 347-348– circular logic and tautology.

**Response**

We have revised this sentence as follows:

[revised manuscript text omitted]

\right\}\\[6pt]
&= \sum_{l}\sum_{k}\left\{
\begin{aligned}
&\frac{1}{W}\sum_{w}\left(\frac{1}{U}\sum_{u}\Delta\left((\mathbf{PR}_{\mathrm{GW}},\mathbf{PR}_{\mathrm{OVN}})_{u}\,|\,\mathbf{PR}_{\mathrm{VDZ}}^{\mathrm{bin}_{w}},NM_{k},CS_{l}\right)\right)^{2}\\
&-\left(\frac{1}{WU}\sum_{w}\sum_{u}\Delta\left((\mathbf{PR}_{\mathrm{GW}},\mathbf{PR}_{\mathrm{OVN}})_{u}\,|\,\mathbf{PR}_{\mathrm{VDZ}}^{\mathrm{bin}_{w}},NM_{k},CS_{l}\right)\right)^{2}
\end{aligned}
\right\}P\left(NM_{k}\,|\,CS_{l}\right)P\left(CS_{l}\right)
\end{aligned}
\tag{18}
$$

where the symbol $U$ refers to the number of combinations of $\mathbf{PR}_{\mathrm{GW}}$ and $\mathbf{PR}_{\mathrm{OVN}}$ in bin $\mathbf{PR}_{\mathrm{VDZ}}^{\mathrm{bin}_{w}}$, i.e., the size of the parameter set in bin $\mathbf{PR}_{\mathrm{VDZ}}^{\mathrm{bin}_{w}}$, and the symbol $u$ is the index for these combinations. $w$ represents the index for the bins of vadose zone parameters, and $W$ is the total number of bins. After applying the LHS sampling method and the same binning method, the partial variance for $\mathbf{PR}_{\mathrm{GW}}$ and $\mathbf{PR}_{\mathrm{OVN}}$ can be estimated as follows:

$$
\begin{aligned}
V\left(\mathbf{PR}_{\mathrm{GW}}\right) &= E_{\mathbf{CS}}E_{\mathbf{NM}|\mathbf{CS}}V_{\mathbf{PR}_{\mathrm{GW}}|\mathbf{NM},\mathbf{CS}}E_{\mathbf{PR}_{\mathrm{VDZ}},\mathbf{PR}_{\mathrm{OVN}}|\mathbf{PR}_{\mathrm{GW}},\mathbf{NM},\mathbf{CS}}\left(\Delta\,|\,\mathbf{PR}_{\mathrm{GW}},\mathbf{NM},\mathbf{CS}\right)\\[6pt]
&= \sum_{l}\sum_{k}\left\{
\begin{aligned}
&\frac{1}{W}\sum_{w}\left(\frac{1}{U}\sum_{u}\Delta\left((\mathbf{PR}_{\mathrm{VDZ}},\mathbf{PR}_{\mathrm{OVN}})_{u}\,|\,\mathbf{PR}_{\mathrm{GW}}^{\mathrm{bin}_{w}},NM_{k},CS_{l}\right)\right)^{2}\\
&-\left(\frac{1}{WU}\sum_{w}\sum_{u}\Delta\left((\mathbf{PR}_{\mathrm{VDZ}},\mathbf{PR}_{\mathrm{OVN}})_{u}\,|\,\mathbf{PR}_{\mathrm{GW}}^{\mathrm{bin}_{w}},NM_{k},CS_{l}\right)\right)^{2}
\end{aligned}
\right\}P\left(NM_{k}\,|\,CS_{l}\right)P\left(CS_{l}\right),
\end{aligned}
\tag{19}
$$

$$
\begin{aligned}
V\left(\mathbf{PR}_{\mathrm{OVN}}\right) &= E_{\mathbf{CS}}E_{\mathbf{NM}|\mathbf{CS}}V_{\mathbf{PR}_{\mathrm{OVN}}|\mathbf{NM},\mathbf{CS}}E_{\mathbf{PR}_{\mathrm{VDZ}},\mathbf{PR}_{\mathrm{GW}}|\mathbf{PR}_{\mathrm{OVN}},\mathbf{NM},\mathbf{CS}}\left(\Delta\,|\,\mathbf{PR}_{\mathrm{OVN}},\mathbf{NM},\mathbf{CS}\right)\\[6pt]
&= \sum_{l}\sum_{k}\left\{
\begin{aligned}

[revised manuscript text omitted]
$_{\text{VDZ}}$ represents the vadose zone parameters. PR$_{\text{GW}}$ represents the groundwater parameters. PR$_{\text{OVN}}$ represents the overland flow parameter.

**Figure 11.** Maps of vadose zone parameter sensitivity indices ($S_{\text{PR}_{\text{VDZ}}}$), groundwater parameter sensitivity indices ($S_{\text{PR}_{\text{GW}}}$) and overland flow parameter sensitivity indices ($S_{\text{PR}_{\text{OVN}}}$) for time-averaged evapotranspiration (*ET*) (a) and groundwater contribution to stream flow (*Q$_G$*) (b) predictions.

**Figure 12. Patterns of $S_{PR}$ , $S_{NM}$ , and $S_{CS}$ for space-averaged evapotranspiration ($ET$) and space-averaged groundwater contribution to stream flow ($Q_G$) with changes in the prior weights of numerical model $NM_1$, climate scenario $CS_1$ and climate scenario $CS_6$ at the time step of 4308 hours (at 12:00).**

**Figure C1. Maps of six parameter sensitivity indices for groundwater contribution to stream flow ($Q_G$) predictions.**

**Figures**

[Figure]

880

**Figure 1**

[Figure]

**Figure 2**

885

[Figure]

**Figure 3**

[Figure]

890    Figure 4

[Figure]

**Figure 5**

[Figure]

895

**Figure 6**

[Figure]

**Figure 7**

900

[Figure]

**Figure 8**

[Figure]

905    **Figure 9**

[Figure]

**Figure 10**

[Figure]

910

**Figure 11**

[Figure]

**Figure 12**

**Map of Six Parameters Time-Averaged Sensitivity Indices for Q$_G$**

[Figure]

Figure C.1